# Mutations in ammonia transporter RhBG that impair NH$_3$/NH$_4^+$ transport in patients with chronic kidney disease

He Zhou[1,2], Solange Abdulnour-Nakhoul[1,2], L. Lee Hamm[1] and Nazih L. Nakhoul[1,2]

[1]*Department of Medicine, Tulane Medical School, New Orleans, LA, USA*
[2]*Department of Physiology, Tulane Medical School, New Orleans, LA, USA*

Handling Editors: Peying Fong & Matthew Bailey

The peer review history is available in the Supporting Information section of this article (https://doi.org/10.1113/JP288958#support-information-section).

**Abstract figure legend** Overview of the study design and workflow. Left: screening for RhBG mutations associated with chronic kidney disease. Right: functional validation of identified mutations using *Xenopus* oocyte electrophysiology with two-electrode voltage clamp technique and intracellular pH measurements, showing impairment of NH$_3$/NH$_4^+$ transport.

**Abstract** Chronic kidney disease (CKD) imposes a substantial health burden globally, with emerging evidence pointing to the significance of metabolic acidosis and low urinary NH$_4^+$

**He Zhou** is a Postdoctoral Researcher in the Department of Internal Medicine-Nephrology at Tulane University School of Medicine. She earned her PhD in Biomedical Sciences (2024) and MPH in Epidemiology from Tulane University, and her Bachelor of Medicine from Qingdao University. Her work bridges basic science and clinical practice by investigating renal membrane transporters in chronic kidney disease. She combines electrophysiology with genetic epidemiology to understand how protein variants affect kidney function, aiming to improve CKD patient outcomes through personalized acid–base management. She has accomplished multiple peer-reviewed publications and presentations at national conferences, and several awards from the American Society of Nephrology (ASN) and the American Physiological Society (APS).

The Journal of Physiology

excretion resulting in poor CKD outcomes. The present study aims to identify in CKD patients, loss of function mutations in RhBG, one of the $NH_3/NH_4^+$ transporters in the collecting duct, and to show that $NH_3/NH_4^+$ transport is impaired by these mutations. Single nucleotide polymorphisms of RhBG associated with CKD occurrence were identified using ancestry-stratified data from the Chronic Renal Insufficiency Cohort (CRIC) study. Functional analysis of $NH_3/NH_4^+$ transport was conducted in *Xenopus* oocytes expressing RhBG protein or mutants. $NH_3$ and $NH_4^+$ transport was evaluated by electrophysiological measurements, including whole cell current, surface pH and intra-cellular pH. Our study identified six critical RhBG mutations associated with CKD. G86S and G86C inhibited the transport of $NH_3$; mutations G148R and G148W completely blocked transport of $NH_3$ and $NH_4^+$, whereas T250A and T250S only inhibited $NH_3$ transport. Mutation T250M completely inhibited transport of both $NH_3$ and $NH_4^+$. Our study identified critical rare non-synonymous single nucleotide polymorphisms in RhBG associated with CKD and elucidated the impact of these variants on $NH_3/NH_4^+$ transport. These data are crucial to our understanding of how mutations can disrupt $NH_3/NH_4^+$ transport, potentially affecting kidney function in CKD patients susceptible to acidosis.

(Received 27 March 2025; accepted after revision 11 December 2025; first published online 11 February 2026)

**Corresponding author** N. L. Nakhoul: Section of Nephrology, SL-45, Department of Medicine, Tulane Medical School, 1415 Tulane Avenue, Suite 400A, New Orleans, LA 70112, USA.     Email: nakhoul@tulane.edu

**Key points**

- Acidosis and low urinary ammonium excretion contribute to poor outcomes in chronic kidney disease (CKD).
- This study investigates how the function of an ammonia transporter in renal collecting duct (RhBG) may contribute to CKD.
- Here, we report six rare RhBG mutations associated with CKD, identified using data from the Chronic Renal Insufficiency Cohort (CRIC) study.
- Using electrophysiological measurements, functional analysis in *Xenopus* oocytes showed that these RhBG mutations disrupt ammonia transport, with some mutations affecting only $NH_3$ transport, whereas others affect both $NH_3$ and $NH_4^+$ transport.
- The results suggest that impaired ammonia transport by RhBG contributes to CKD, highlighting the need to understand mechanisms that link function ($NH_3/NH_4^+$ and acid-base regulation) and genetic predisposition to CKD.

## Introduction

Chronic kidney disease (CKD) involves a loss of renal function over time (Maringhini & Zoccali, 2024). As CKD progresses, the ability of the kidneys to excrete the daily load of acids, predominately generated from metabolism, is impaired (Nagami & Hamm, 2017; Nagami & Kraut, 2022). This leads to the retention of hydrogen ions ($H^+$) and eventually an associated decline in plasma pH, namely metabolic acidosis.

The kidneys respond to acidosis by increasing urinary excretion of total ammonia ($NH_3$ and $NH_4^+$) and titratable acids. The excretory capacity for ammonia (50–67%) accounts for the major urinary acid excretion compared to titratable acids (33–50%) (Hamm et al., 2015). In the human kidney, total ammonia is generated predominantly in the proximal tubule by deamination of glutamine and secreted into the lumen, reabsorbed in the thick ascending limb of Henle's loop, and then secreted into the collecting duct. In the collecting duct, $NH_3/NH_4^+$ is transported by two $NH_3/NH_4^+$ transporters, RhBG and RhCG (Han et al., 2013; Quentin et al., 2003). RhBG, that is, Rhesus (Rh) glycoprotein type B, transports both forms of ammonia ($NH_3$ and $NH_4^+$) (Nakhoul & Lee Hamm, 2013; Nakhoul et al., 2005, 2006; Nakhoul, Abdulnour-Nakhoul, Boulpaep, et al., 2010; Nakhoul, Abdulnour-Nakhoul, Schmidt, et al., 2010) at the basolateral membrane of $\alpha$-intercalated cells and non-A, non-B cells (Han et al., 2013).

In the early stage of CKD, $H^+$ can accumulate in such organs as kidneys and bones, even without affecting serum bicarbonate levels; this has been referred to as

eubicarbonatemic metabolic acidosis (Nagami & Kraut, 2022). As CKD progresses, the increase in ammonium excretion in response to acidosis is impaired, and therefore the prevalence of overt metabolic acidosis increases (Kim, 2021). Recent studies suggest that the reduced ability of the kidney to excrete ammonium is associated with detrimental outcomes in CKD (such as death and cardio-vascular events) and faster progression towards end-stage renal disease (ESRD), measured as a decline in glomerular filtration rate (GFR) over time (Kim, 2021; Raphael et al., 2017; Rehman et al., 2023; Vallet et al., 2015; Wesson et al., 2020). This association between lower urinary total ammonia excretion and CKD progression (ESRD or faster GFR decline) has been found even without overt metabolic acidosis (Madias, 2021; Tyson et al., 2021; Vallet et al., 2015) independently of measured GFR and other factors (Madias, 2021). These findings suggest that lower urinary ammonium excretion can serve as an early indicator of higher CKD risk, even earlier than serum total $CO_2$ level (Raphael et al., 2017). Lower ammonium excretion and more retention of acid may be an important pathogenic factor, not just a marker, in CKD progression as well.

Ammoniagenesis is enhanced to increase bicarbonate production to counteract acidosis (DuBose, 2017; Madias, 2021; Wesson et al., 2020). The enhanced ammoniagenesis probably also exists in early-stage CKD during preclinical acidosis given that dietary acid load, in those patients, increased ammonia production without discernible systemic acidosis (DuBose, 2017). Tissue-specific knockout studies in mice have reported a correlation between Rhbg (homologue of RhBG) deletion and exacerbated metabolic acidosis with acid challenge (Bishop et al., 2010; Lee et al., 2014). A higher risk of CKD progression has been observed in human subjects with early-stage CKD subjected to a high-protein diet as acid loading. Acid loading was determined by measurement of urinary net acid excretion (Banerjee et al., 2015; Scialla et al., 2012, 2017). Those important findings suggest a critical role of ammonia excretion, possibly mediated by RhBG.

Because current evidence supporting the link between low urinary ammonium excretion and poor CKD outcomes is heavily based on observational studies, there is a critical need to provide biomedical evidence to validate the importance of clinically measuring urinary ammonium in patients. In the present study, our hypothesis is that impaired transport of $NH_3/NH_4^+$ by RhBG is a factor that contributes to CKD. Although other processes and transporters can affect ammonium excretion, the focus here is on RhBG.

The aims of the present study are to: (i) identify human RhBG non-synonymous single nucleotide polymorphisms (nsSNPs) that are possibly linked to CKD and (ii) validate the functional impacts of such SNPs on total ammonia transport. To do so, we first conducted association analyses using whole-exome sequencing (WES) data collected from participants in the Chronic Renal Insufficiency Cohort (CRIC) study and identified SNPs with significant associations with CKD outcome. We then performed functional studies to investigate the biological significance of the identified SNPs using electrophysiological techniques. Our results show that: (i) six mutations (G86S, G86C, G148R, G148W, T250A and T250S) are associated with CKD in the study cohort; (ii) two mutations associated with CKD progression, G86S and G86C, partially inhibited transport, with G86C being more deleterious than G86S in its inhibitory effect on $NH_3/NH_4^+$ transport; (iii) mutations G148R, G148W and T250M completely inhibited the transport of $NH_3$ and $NH_4^+$, demonstrating that they are critical sites for RhBG protein function.

## Methods

### Ethical approval

The protocols for housing and handling of *Xenopus laevis* frogs and the experimental protocols were reviewed and approved by the Institutional Animal Care and Use Committee at Tulane University (approval ID number 1802).

### Animal care

Frogs were housed in a 100 gallon water tank, typically half-filled, that held a minimum of six and up to a maximum of 20 frogs at a time. The tank water was dechlorinated and circulated through a charcoal-filtered power pump. The size of the tank and volume of water allowed frogs to swim freely. The tank was supplied with plastic pipes for activity enrichment. The frogs were fed three times per week with adult *Xenopus* diet (Nasco Frog Brittle; Nasco Education, Fort Atkinson, WI, USA) sprinkled directly into the tank. After each feeding, the frogs were removed to a spare similar tank and the first tank was cleaned and refilled with water.

### Animal procedures

To extract oocytes, *Xenopus* frogs were anaesthetized via the transdermal route by immersion in filtered water containing 0.2% tricaine (ethyl 3-aminobenzoate methanesulfonate; Sigma, St Louis, MO, USA) for 15 min. Adequate anaesthetic depth was confirmed by the absence of withdrawal reflexes upon application of mechanical stimulus to the hindlimb. Oocytes were then extracted as described below (see section below, 'Oocyte isolation'). Following oocyte removal, the frogs were monitored for

30–60 min to ensure complete recovery from anaesthesia before being transferred to individual recovery bins overnight without feeding. Frogs were then returned to standard housing conditions. Our protocol allows six survival surgeries, after which the frog was killed by deep anaesthesia (immersion in 0.2% tricaine for 2 h) followed by cardiac excision.

### Sex as a biological variable

Our studies on association tests in humans included data on both males and females. These data obtained from public databases included both sexes, but sex was not considered a biological variable as a result of an insufficient sample size for stratification by sex. Only female *Xenopus* frogs were used in this study to extract oocytes.

### Identification of RhBG nsSNPs associated with CKD

**Study population.** We obtained the WES dataset of the CKD patients in CRIC study and selected 507 participants of African ancestry and 1147 of European ancestry. All selected cases had eGFR <60 mL min$^{-1}$ per 1.73 m$^2$ or spot urine albumin-to-creatinine ratio ≥30 µg mg$^{-1}$ (Pan et al., 2023). Cases were compared with healthy controls selected from the Atherosclerosis Risk in Communities (ARIC) study that included 6044 subjects of European ancestry and 2056 of African ancestry. All selected healthy controls had eGFR ≥90 mL min$^{-1}$ per 1.73 m$^2$.

**Association analysis.** Ancestry-stratified nsSNP association tests were performed using RStudio (R version 4.3.1; R Foundation, Vienna, Austria). nsSNPs of RhBG present in the CRIC WES dataset with a minor allele frequency (MAF) $\leq$ 1% were tested individually for association between cases and controls. Rare variants (MAF $\leq$ 1%) are prioritized because they more probably have larger effect sizes on the function. MAF values were calculated using standard allele frequency methodology (MAF $= \frac{\text{copies of the alternate allele}}{\text{total copies of all alleles}}$), computed separately for each ancestry group in CRIC and ARIC cohorts. Pearson's chi-squared test with Yates' continuity correction was chosen for simplicity and its non-parametric nature. The odds ratio (OR) was also calculated to measure the effect size using Fisher's exact test given our focus on rare variants. For certain nsSNPs with MAF of 0 in our sample, risk difference (RD) instead of OR was reported along with the *P* value of Fisher's exact test. All *P* values were adjusted with the Bonferroni correction method by pooling the two ancestry groups to reach maximum robustness. *P* < 0.05 was considered statistically significant. Findings with significant OR or RD after correction (*P* < 0.05) were

used for functional testing. This study did not examine common variants (MAF > 1%).

**Selection of potentially critical nsSNPs.** The number of RhBG nsSNPs in our study population that met selection criteria was small ($N = 10$) (Table 1) and only four reached statistical significance (Table 2). Among these four nsSNPs, rs150963900 (F81L) in the European group showed a protective effect (OR < 1) which is outside our study scope. Therefore, we excluded it from the functional test. nsSNP rs200320178 (T250M) causes a missense mutation of the same amino acid as rs760016272 (T250A/S). To explore the potential effect of this critical amino acid, we included rs200320178 (T250M) in downstream functional analyses.

### Functional tests of selected RhBG nsSNPs

Functional testing of RhBG mutations was conducted by expressing RhBG mutants in *Xenopus* oocytes and measuring NH₃/NH₄⁺ transport. NH₃/NH₄⁺ transport was assessed by measurements of NH₃/NH₄⁺-induced currents, intracellular pH (pH$_i$) and surface pH (pH$_s$), as described below.

**Site-directed mutagenesis.** Human RhBG clone was purchased from GeneScript and subcloned into pGH19 plasmid and purified by QIAprep Miniprep kit (Qiagen, Santa Clarita, CA, USA).

*Mutagenic primer design.* Complementary (100% overlap) mutagenic oligonucleotide primers were designed individually according to the desired mutation (Table 3). Each pair of mutagenic primers was between 25 and 45 bases in length with a high annealing temperature (≥78°C) and a minimum GC content of 40% and terminated in one or more C or G bases. The desired mutation was flanked by at least 10 bases of unmodified sequence on both sides. To increase the success rate of getting the desired mutations, we conducted two-stage PCR amplification (Wang & Malcolm, 2002). Purified forward and reverse primers were added to two thin-walled PCR tubes separately. After adding the circular plasmid DNA template and other PCR reagents (buffer, dNTPs, ddH₂O, dimethyl sulfoxide, PfuTurbo polymerase) (#600600; Agilent, Santa Clara, CA, USA) to each tube (50 µL reaction volume), we performed PCR following a regular three-step PCR protocol. After five cycles at the annealing stage, we paused the program, took a half-volume of PCR mix (25 µL) from each tube containing forward or reverse primer to combine in a new tube and allow the combined tube to continue PCR for another 16 cycles followed by the extension and cooling stages.

**Table 1. List of rare single nucleotide variants (SNVs) from the CRIC study**

| Variant.ID | Amino acid substitution | African ancestry (N = 2563) | | European ancestry (N = 7191) | |
| | | Case MAF (N = 507) | Control MAF (N = 2056) | Case MAF (N = 1147) | Control MAF (N = 6044) |
| --- | --- | --- | --- | --- | --- |
| rs202003473 | Y36C | 0 | 0 | 0.001689 | 0.000437 |
| rs370957772 | L82R | 0 | 0 | 0.001689 | 0.000146 |
| rs200069134 | G86S/C | 0 | 0 | 0.005068 | 0.001091 |
| rs575652633 | G148R/W | 0 | 0 | 0.001689 | 0 |
| rs190623502 | R211C | 0 | 0 | 0.001689 | 0.000509 |
| rs760016272 | T250A/S | 0 | 0 | 0.001689 | 0 |
| rs200320178 | T250M | 0 | 0.000632 | 0.003378 | 0.003420 |
| rs201484566 | A286G | 0.001684 | 0.000632 | 0 | 0 |
| rs376569860 | L358P | 0.001684 | 0.001476 | 0 | 0 |
| rs150963900 | F81L | 0.003367 | 0.001896 | 0.013510 | 0.022700 |

Selected SNVs from the WES data of the subgroup of CRIC study with MAF falling in the interval (0, 1%]. Only 10 SNVs meet the criteria. Controls are from the ARIC WES dataset. *N* indicates the sample size of participants in each subgroup based on disease and ancestry. Allele counts are not shown. ARIC, Atherosclerosis Risk in Communities; CRIC, Chronic Renal Insufficiency Cohort; MAF, minor allele frequency; WES, whole-exome sequencing.

*Transformation.* PCR product was treated with *Dpn*I restriction enzyme (New England Biolabs, Ipswich, MA, USA) and incubated at 37°C for 1 h to digest the template DNA. *Dpn*I treated PCR product was then transformed into XL10 gold Ultracompetent cells (#200517-4; Agilent) for amplification, antibiotic selection, plasmid isolation and purification. We purified plasmid DNA from selected colonies and confirmed the mutation by Sanger sequencing (GENEWIZ; Azenta Life Science, Burlington, MA, USA).

**cRNA preparation.** The purified pGH19 plasmid containing RhBG cDNA [wild-type (WT) or with mutations] was digested with *Nhe*1 enzyme (New England Biolabs) to create a linear template, followed by digestion with proteinase K. The DNA was then extracted using phenol-chloroform extraction and ethanol precipitation. Capped RNA (cRNA) transcripts were synthesized *in vitro* using the mMachine T7 transcription kit (Ambion, Austin, TX, USA). The concentration of cRNA was measured using a Nanodrop spectrophotometer (Thermo Fisher Scientific, Waltham, MA, USA) and its quality was evaluated using formaldehyde/Mops/1% agarose gel electrophoresis.

**Oocyte isolation.** All experiments were conducted on *X. laevis* oocytes. Isolation of oocytes followed standard protocols as reported in previous study (Nakhoul et al., 2005). Briefly, the frog was anesthetized by immersion in water containing 0.2% tricaine. A 1 cm incision was made in the abdominal wall to access a portion of the ovary, which was then cut. The wound was closed with stitches in

the muscular layer using 5-0 chromic gut and, in the skin, using 6-0 silk. The excised ovary piece containing oocytes was rinsed with $Ca^{2+}$-free ND96 solution, then agitated in $Ca^{2+}$-free ND96 solution containing collagenase type IV (0.3 mg mL$^{-1}$). After 30–40 min, free oocytes were rinsed with sterile OR3 medium, pH 7.5 and osmolality of 200 mOsm kg$^{-1}$ L-15 medium [(L-4386; Sigma), 5x penicillin-streptomycin (P0781; Sigma) and 5.5 mM Hepes (H4034; Sigma)], and then sorted and stored at 18°C overnight before injection.

**Oocyte injection.** Each oocyte from the same batch was injected with 50 nL of WT-RhBG or mutant cRNA (0.2 μg μL$^{-1}$, totalling 10 ng of RNA) for protein expression or double-distilled $H_2O$ (Thermo Fisher Scientific). Water-injected oocytes served as controls to account for endogenous $NH_4^+$-permeable channel activity. The injections were made using sterile pipettes with tip diameters of 20–30 μm, connected to a Nanoject motor-driven pipette (Drummond Scientific, Broomall, PA, USA). Injected oocytes were observed 3 days post-cRNA injection and used for the following 4–5 days. Injection for all experimental and control groups was repeated in three or more batches of oocytes.

**Electrophysiological measurements.** All measurements were performed on oocytes expressing WT-RhBG or mutants. Oocytes were placed in a perfusion chamber (Warner Instruments, Holliston, MA, USA) designed to allow continuous fast perfusion of different solutions. Perfusion rate was 10 mL min$^{-1}$ and complete change of solutions in the chamber was accomplished in 6 s. Voltage

**Table 2. Results of ancestry-stratified single-variant association tests**

| Variant.ID | Amino acid substitution | Ancestry | Chi-squared | Chi-square P value | OR | RD | Fisher's exact P value | P value corrected | P value significance |
|---|---|---|---|---|---|---|---|---|---|
| rs200320178 | T250M | African | 0.018353 | 0.8922 | | −0.1979309 | 1 | 1 | |
| rs201484566 | A286G | African | 0.32965 | 0.5659 | 2.706786 | | 0.2581265 | 1 | |
| rs376569860 | L358P | African | 1.00E-27 | 1 | 1.158403 | | 0.6947681 | 1 | |
| rs150963900 | F81L | African | 8.91E-03 | 0.9248 | 1.355423 | | 0.7145403 | 1 | |
| rs202003473 | Y36C | European | 3.5427 | 0.05981 | 4.225876 | | 0.04139785 | 0.4553763 | |
| rs370957772 | L82R | European | 8.0594 | 0.004527 | 10.56771 | | 0.00734661 | 0.08081271 | |
| rs200069134 | G865/C | European | 16.967 | 3.80E-05 | 4.902102 | | 0.0001724 | 0.001896389 | ** |
| rs575652633 | G148R/W | European | 15.307 | 9.14E-05 | | 0.8409628 | 0.00064208 | 0.007062902 | ** |
| rs190623502 | R211C | European | 2.7158 | 0.09936 | 3.521293 | | 0.06049482 | 0.665443 | |
| rs760016272 | T250A/S | European | 15.307 | 9.14E-05 | | 0.8409628 | 0.00064208 | 0.007062902 | ** |
| rs200320178 | T250M | European | 1.69E-26 | 1 | 1.027837 | | 0.8471217 | 1 | |
| rs150963900 | F81L | European | 7.6372 | 0.005718 | 0.5849269 | | 0.00346187 | 0.0380806 | * |

Chi-squared was calculated with Pearson's chi-squared test with Yates' continuity correction, degree of freedom = 1. The odds ratio was calculated with Fisher's exact test. Allele counts are not shown. Risk difference (RD) was calculated for variants with one MAF of 0 with Fisher's exact *P* value reported. Fisher's exact *P* values are adjusted to account for multiple comparisons with Bonferroni method. Corrected $P < 0.05$ is considered statistically significant. OR, odds ratio; RD, risk difference.

clamp experiments, surface pH ($pH_s$) and intracellular pH ($pH_i$) measurements were extensively described previously (Geyer et al., 2013; Nakhoul et al., 2005, 2006). Measurements on each group were repeated in at least three batches of oocytes, with similar group sizes maintained across batches to ensure balanced representation.

**Perfusion solutions.** Control solution (referred to as 'Hepes' in this paper) was made from ND96 medium containing 96 mM NaCl, 2 mM KCl, 1 mM $MgCl_2$ and 1.8 mM $CaCl_2$ and buffered with 5 mM Hepes to pH 7.5. Two types of testing solution were prepared: (i) $NH_3/NH_4^+$ solution was prepared by replacing NaCl with $NH_4Cl$ resulting in 5 mM $NH_4Cl$ and (ii) methylamine/methylammonium ($MA/MA^+$) solution was made by replacing NaCl with methylamine hydrochloride resulting in 5 mM $MA/MA^+$. All solutions had osmolarities of $200 \pm 5$ mOsm $L^{-1}$. Unless otherwise noted, all chemicals and reagents used in this study were purchased from Sigma-Aldrich (St Louis, MO, USA).

**Two-electrode voltage clamp.** Whole-cell currents were measured using a two-electrode voltage clamp system (OC-725; Warner Instruments). Borosilicate glass capillaries (1.5 mm outer diameter; Warner Instruments) were pulled using a horizontal puller (model P-97; Sutter Instruments, Novato, CA, USA) to create electrodes with resistances of 1–4 MΩ, filled with 3 M KCl solution. The external reference electrodes were a long glass micropipette with a tip of ∼10 μm filled with 3 M KCl. A Ag/AgCl wire, dipped in the 3 M KCl filling solution provided electrical connection to the electrometer. It was directly immersed in the chamber. Oocytes were clamped at −60 mV, and continuous current readings were sampled at a rate of once per second. Inward flow of cations was detected as inward current (negative current).

**Measurement of surface pH ($pH_s$).** Single-barreled alumino-silicate glass capillaries (1.5 mm outer diameter, 0.86 mm inner diameter; Frederick Haer, Brunswick, MD, USA) were pulled to form microelectrodes with a tip of ∼12 μm, then dried in an oven at 200°C for 2 h. They were then silanized with bis(dimethylamino)-dimethyl silane vapor in a closed vessel (300 mL). The tip of each microelectrode was filled with $H^+$ ionophore cocktail B (#95293; Fluka Chemical Corp., Ronkonkoma, NY, USA) to form a pH microelectrode. The remaining space of the microelectrode was backfilled with a buffer solution (Ammann et al., 1981; Nakhoul, Abdulnour-Nakhoul, Boulpaep, et al., 2010). The electrode signal was measured with a FD223 electrometer (World Precision Instruments, Sarasota, FL, USA). The external reference electrode was prepared in the same way as for the two-electrode voltage

**Table 3. List of SNPs included in functional tests and primers used for mutagenesis**

| Variant.ID | Amino acid sub-stitution | Nucleotide substitution | Primers for SDM |
|---|---|---|---|
| rs200069134 | G86S | G>A | [R] 5′-cacgctgctgaagctgtaacgctgcagga-3′<br>[F] 5′-tcctgcagcgttacagcttcagcagcgtg-3′ |
| rs200069135 | G86C | G>T | [R] 5′-cacgctgctgaagcagtaacgctgcagga-3′<br>[F] 5′-tcctgcagcgttactgcttcagcagcgtg-3′ |
| rs575652633 | G148R | G>C | [R] 5′-gctgggtaggccgggtcttgcccag-3′<br>[F] 5′-ctgggcaagacccggcctacccagc-3′ |
| rs575652634 | G148W | G>T | [R] 5′-gctgggtaggccaggtcttgcccag-3′<br>[F] 5′-ctgggcaagacctggcctacccagc-3′ |
| rs760016272 | T250A | A>G | [R] 5′-tgttgagggccgcccgatgctgccc-3′<br>[F] 5′-gggcagcatcgggcggccctcaaca-3′ |
| rs760016273 | T250S | A>T | [R] 5′-tgttgagggccgaccgatgctgccc-3′<br>[F] 5′-gggcagcatcggtcggccctcaaca-3′ |
| rs200320179 | T250M | C>T | [R] 5′-ttgagggccatccgatgctgcccagcc-3′<br>[F] 5′-ggctgggcagcatcggatggccctcaa-3′ |

Abbreviations: F, forward; R, reverse; SDM, site-directed mutagenesis.

clamp. During the experiment, the $pH_s$ electrode was placed in direct contact with the oocyte surface after the calibration in bath. The electrode in-position does not impale the oocyte but should cause a visible slight dimple on the oocyte membrane.

**Measurement of intracellular pH ($pH_i$).** Microelectrodes for measuring $pH_i$ were prepared in a similar way to $pH_s$ microelectrode except that they were pulled to a smaller tip (<0.2 μm). Ling–Gerard (voltage) micro-electrodes were pulled from 1.5 mm (outer diameter) borosilicate glass capillaries with an inner filament and filled with 3 M KCl. An Ag/AgCl wire was inserted in the 3 M KCl filling solution. The external reference electrode was prepared in the same way as above. During the experiment, the oocyte was impaled with both a $pH_i$ electrode and a voltage electrode for sensing pH and membrane potential simultaneously. The outputs of the FD223 electrometer (both $V_m$ and pH) were fed to an electronic subtraction box (Yale, New Haven, CT, USA) to obtain net $pH_i$ voltage. Electrical outputs were digitized and custom-made software displayed net pH and $V_m$ readings online.

**Immunofluorescence to test expression.** The preparation was described in detail previously (Abdulnour-Nakhoul et al., 2016). Briefly, oocytes injected with cRNA (WT or mutant) or water (as the control) were snap-frozen in OCT compound over dry ice. Cryo-sections (5 μm) were fixed, rehydrated, and incubated with human RhBG antibody (dilution 1:500; #AAS03495C; Antibody Verify, Las Vegas, NV, USA), followed by goat anti-rabbit Alexa Fluor 488 secondary antibodies (dilution 1:500; #A-11008;

Invitrogen, Waltham, MA, USA). Negative controls were included. Micrographs were captured using an Eclipse 80i microscope (Nikon, Tokyo, Japan) with a SPOT RT digital camera (SPOT Imaging, Sterling Heights, MI, USA). All immunofluorescence micrographs were captured under identical settings using standardized exposure and acquisition parameters.

**Statistical analysis**

All variables in electrophysiological measurements were compared using mixed-effects models to account for oocyte clustering within injected batches. Outliers were identified using 'above Q3 + 1.5 × interquartile range (IQR) or below Q1 − 1.5 × IQR' method and removed prior to all analyses. To assess the consistency of electro-physiological measurements across oocyte batches, intra-class correlation coefficient (ICC) scores were calculated using mixed-effects models. For each parameter, a null model (including only the random effect of batch) and a full model (including both fixed and random effects) were fitted using lmer() function in package 'lme4' (Bates et al., 2015). ICC values were computed in R using the performance::icc() function in package 'performance' (Lüdecke et al., 2021), which estimates adjusted ICC based on experimental oocyte groups (WT, mutants, water-injected control). Adjusted ICC values reflect the proportion of total variance attributable to batch effects after accounting for fixed effects. ICC scores were interpreted as follows: values <0.2 indicate minimal batch-related variance, whereas values between 0.2 and 0.5 suggest moderate batch-related variance. Higher ICC

values indicate greater inter-batch variability, supporting the need for mixed-effects modelling.

Linear mixed-effects models were fitted with experimental oocyte groups (WT, mutants, water-injected control) as fixed effects and batch as a random effect using lmer() function in package 'lme4'. Linearity of the relationship between experimental groups and electrophysiological measurements was assessed using the Ramsey RESET test in package 'lmtest' (Zeileis et al., 2002), which indicated no significant deviation from linearity, supporting the use of a linear model. Estimated marginal means (EMMs) were computed using package 'emmeans' (DOI:10.32614/CRAN.package.emmeans) and custom contrasts were defined to compare each mutant group and the water-injected group against the RhBG control. *P* values of EMM contrasts were adjusted for multiple comparisons using the false discovery rate method to control for type I error. Complete mixed-effects model outputs are provided in the Appendix (Tables A1 and A2) for methodological transparency.

Descriptive statistics (mean ± SD) represent observed (measured) values and were reported for biological interpretation and data transparency. Observed means of different groups were also analysed with one-way ANOVA followed by appropriate *post hoc* tests; for methodological comparison and transparency, see Appendix Methods and Table A3. All statistical results were generated in RStudio (R version 4.4.3) and Excel (Microsoft Corp., Redmond, WA, USA), unless otherwise noted.

## Results

### CKD-related RhBG rare variants

Our study population from the CRIC database included a European ancestry group (73%) and an African ancestry group (27%) (Table 1). There were 10 rare variants (MAF < 1%) identified from the CRIC study WES dataset. Six of these variants were only detected in the group of European ancestry. Only two were in the African group and only one was in both ancestry groups. The results of association tests (chi-squared test) and effect size (OR or RD) based on Fisher's exact test were corrected with different methods (Table 2). After correction, four SNPs reached statistical significance in both the chi-squared test and Fisher's exact test (*P* < 0.05). Variant rs150963900 (F81L) has an OR less than one (OR = 0.58), which indicates a protective effect on the outcome (presence of CKD) and was not included in functional experiments. The three other variants (rs200069134, rs575652633 and rs760016272) cause six types of amino acid substitution (G86S, G86C, G148R, G148W, T250A and T250S) (Table 3). Variant rs200320178 (T250M) did not reach statistical significance but it causes a missense mutation at the same amino acid as SNP rs760016272 (T250A/S).

The identified variants that are significantly associated with CKD and their primer designs for site-directed mutagenesis (SDM) are listed in Table 3.

### Effects of expressing human RhBG on current, pH$_i$ and pH$_s$ changes

In initial experiments, we documented that human RhBG transports NH₃/NH₄⁺ and MA/MA⁺ similar to the well-studied mouse homologues. NH₄⁺ (or MA⁺) transport will cause an inward current and depolarization of the oocyte membrane, and NH₃ and MA transport will cause a decrease in pH$_s$. Our results show direct measurements of currents and intracellular and surface pH in oocytes expressing the human clone of RhBG exposed to NH₃/NH₄⁺ or MA/MA⁺ (Fig. 1). In RhBG-expressing oocytes, the inward current induced by 5 mm NH₄⁺ (Fig. 1*A*) was significantly different (−35.7 ± 13.3 nA, *P* < 0.001) compared to H₂O-injected control oocytes (−15.6 ± 7.7 nA). MA⁺ at a concentration of 5 mm induced a significant inward current (−27.1 ± 9.88 nA, *P* < 0.001) in RhBG-expressing oocytes. This inward current was not observed in any H₂O-injected control oocytes exposed to MA⁺, which is consistent with previous studies (Ludewig, 2004; Nakhoul et al., 2005; Nakhoul, Abdulnour-Nakhoul, Boulpaep, et al., 2010). The results of these measurements suggested that the electrogenic transport of NH₄⁺ and MA⁺ by human RhBG follows a similar pattern to its mouse homologue (Rhbg) as shown in our previous studies (Nakhoul et al., 2005, 2006; Nakhoul, Abdulnour-Nakhoul, Boulpaep, et al., 2010). Moreover, 5 mm NH₄⁺ induced a pH$_i$ decrease and depolarization in both RhBG-expressing and H₂O-injected oocytes (Fig. 1*B*) as reported previously in studies using mouse Rhbg (Nakhoul et al., 2005, 2006; Nakhoul, Abdulnour-Nakhoul, Boulpaep, et al., 2010). These changes are consistent with influx of NH₄⁺. A concentration of 5 mm NH₄⁺ induced a faster pH$_i$ decrease (steeper slope, *P* < 0.001) and larger depolarization of $V_m$ (*P* < 0.001) in RhBG-expressing oocytes compared to H₂O-injected control oocytes. The effects of MA/MA⁺ on pH$_i$ and $V_m$ caused an increase in pH$_i$ (not a decrease) but still caused a depolarization of $V_m$ (Fig. 1*B*, black tracing) but had no effect on H₂O-injected oocytes (Fig. 1*B*, grey tracing). These changes resemble those in oocytes expressing mouse Rhbg (Nakhoul, Abdulnour-Nakhoul, Boulpaep, et al., 2010).

We also measured the effects of NH₃/NH₄⁺ and MA/MA⁺ on surface pH (pH$_s$). As shown in Fig. 1*C*, NH₃ reduced pH$_s$ in RhBG-expressing oocytes by 0.201 ± 0.053 pH units. We also found that MA induced a decrease in pH$_s$ (−0.081 ± 0.030 pH units) as it did in previous Rhbg studies (Caner et al., 2015). The effects of NH₃ entry on pH$_s$ in RhBG-expressing

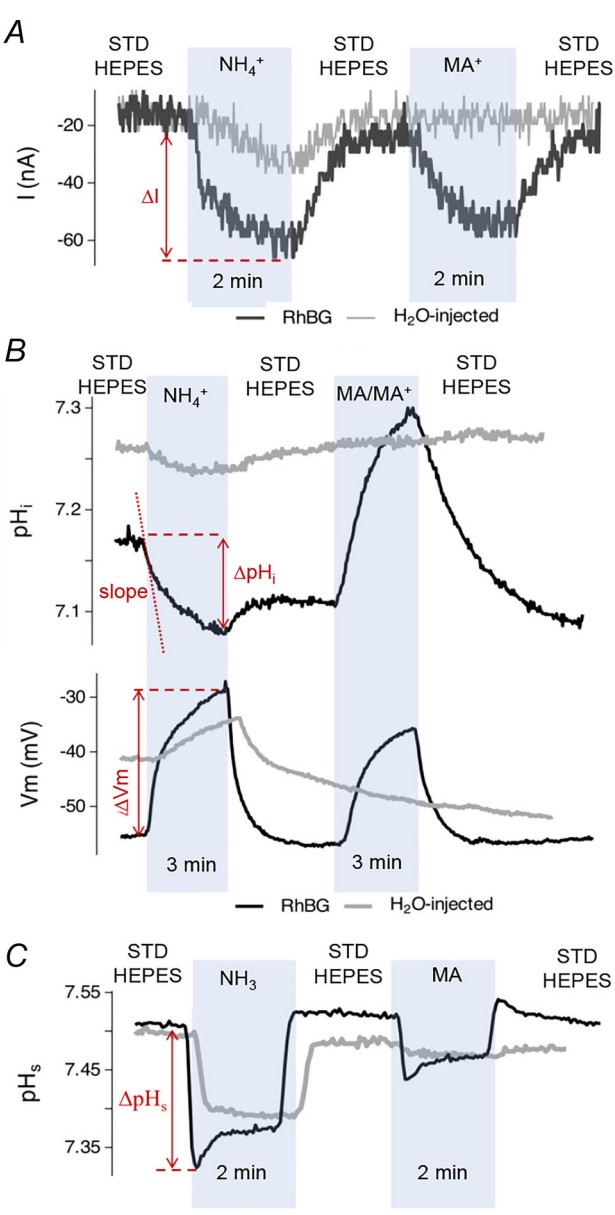

not cause any current change in the control oocyte. *B*, intracellular pH measurements. In RhBG-expressing oocyte, the entry of $NH_4^+$ caused a rapid decrease in $pH_i$ (bigger absolute value of slope) and a larger depolarization of the membrane potential compared to the control (grey tracing). In RhBG-expressing oocyte, $MA^+$ not only induced an increase in $pH_i$ (rather than a decrease) in the RhBG-expressing oocyte, but also induced depolarization of $V_m$. There were no significant $MA/MA^+$-induced changes of $pH_i$ or $V_m$ in $H_2O$-injected control oocytes. *C*, surface pH measurements. In RhBG-expressing oocytes, $NH_3/NH_4^+$ caused rapid decrease in $pH_s$ followed by a partial recovery. Removal of $NH_3/NH_4^+$ caused full recovery of $pH_s$ with a slight overshoot. Exposure to $MA/MA^+$ caused a similar pattern of $pH_s$ change (rapid decrease followed by partial recovery) but the changes were smaller compared to $NH_3/NH_4^+$. In $H_2O$-injected oocytes, $NH_3/NH_4^+$ caused a smaller but sustained decrease in $pH_s$ (no partial recovery). $MA/MA^+$ did not cause any significant change in $pH_s$. The decrease in $pH_s$ is caused by non-charged $NH_3$ and MA entry resulting in a larger $\Delta pH_s$ in the RhBG-expressing oocyte than the control (statistics are shown in Fig. 3C and *D*).

**Figure 1. Measurements in oocytes (WT-RhBG vs. $H_2O$) exposed to $NH_3/NH_4^+$ or $MA/MA^+$**

A light blue rectangle indicates the period when the oocyte was perfused with testing solutions containing either $NH_3/NH_4^+$ or $MA/MA^+$. STD Hepes represent the time when the oocyte was perfused with control solution (Hepes). Black and grey tracings represent the measurements conducted on the oocyte expressing WT-RhBG or on the oocyte injected with water as a control, respectively. Each tracing is a single representative experiment. All changes observed were reversed upon removal of test solutions. nA, nanoamp; $V_m$, membrane potential; mV, millivolt; $\Delta$, measurement difference (end value minus initial value); slope $= \frac{\Delta pHi}{\Delta time\ (s)} \times 10^4$. *A*, whole cell transmembrane current (*I*) measurements at a clamped voltage of $-60$ mV. Positively charged $NH_4^+$ and $MA^+$ entry induced inward current in the RhBG-expressing oocyte. $NH_4^+$ induced a significantly smaller current in the control oocyte than in RhBG-expressing oocyte (statistics are shown in Fig. 2C). $MA^+$ did

and $H_2O$-injected oocytes were described before at a lower $NH_3$ concentration (Geyer et al., 2013). Although consistent with the results of Geyer et al. (2013), this study additionally shows the effects of MA on $pH_s$ using RhBG human clone. The results of these experiments indicated similar transport patterns of human RhBG to its mouse homologue in transporting $NH_3/NH_4^+$ and $MA/MA^+$ (Abdulnour-Nakhoul et al., 2016; Caner et al., 2015). The statistics showing significance are summarized in Figs 2C and *D*, 3C and *D* and 4D and *E*.

### Effects of expressing RhBG mutations on current, $pH_s$ and $pH_i$ changes

To evaluate the functional effects of identified variants, we measured substrate-specific transport ($NH_4^+$, $MA^+$, $NH_3$ and MA) by human RhBG mutant proteins (G86C/S, G148R/W and T250A/S/M) expressed in *Xenopus* oocytes using electrophysiological methods (current, $pH_s$ and $pH_i$) and compared the results with WT RhBG (WT-RhBG). ICC scores of all measurements across different groups demonstrate zero to moderate inter-batch variation (Table 4). Table 5 summarizes the results of electrophysiological and intracellular measurements of all seven mutations compared to WT-RhBG with adjusted *P* values of EMM contrasts.

### G86C and G86S

*Effects of G86C and G86S mutations on electrogenic $NH_4^+$ and $MA^+$ transport: measured by current.* Figure 2 shows $NH_4^+$ (and $MA^+$)-induced currents in oocytes expressing G86C and G86S mutants compared to WT-RhBG. As shown in Fig. 2*A* (tracings in the light blue area), the $NH_4^+$-induced $\Delta I$ in $RhBG_{G86C}$-expressing oocytes

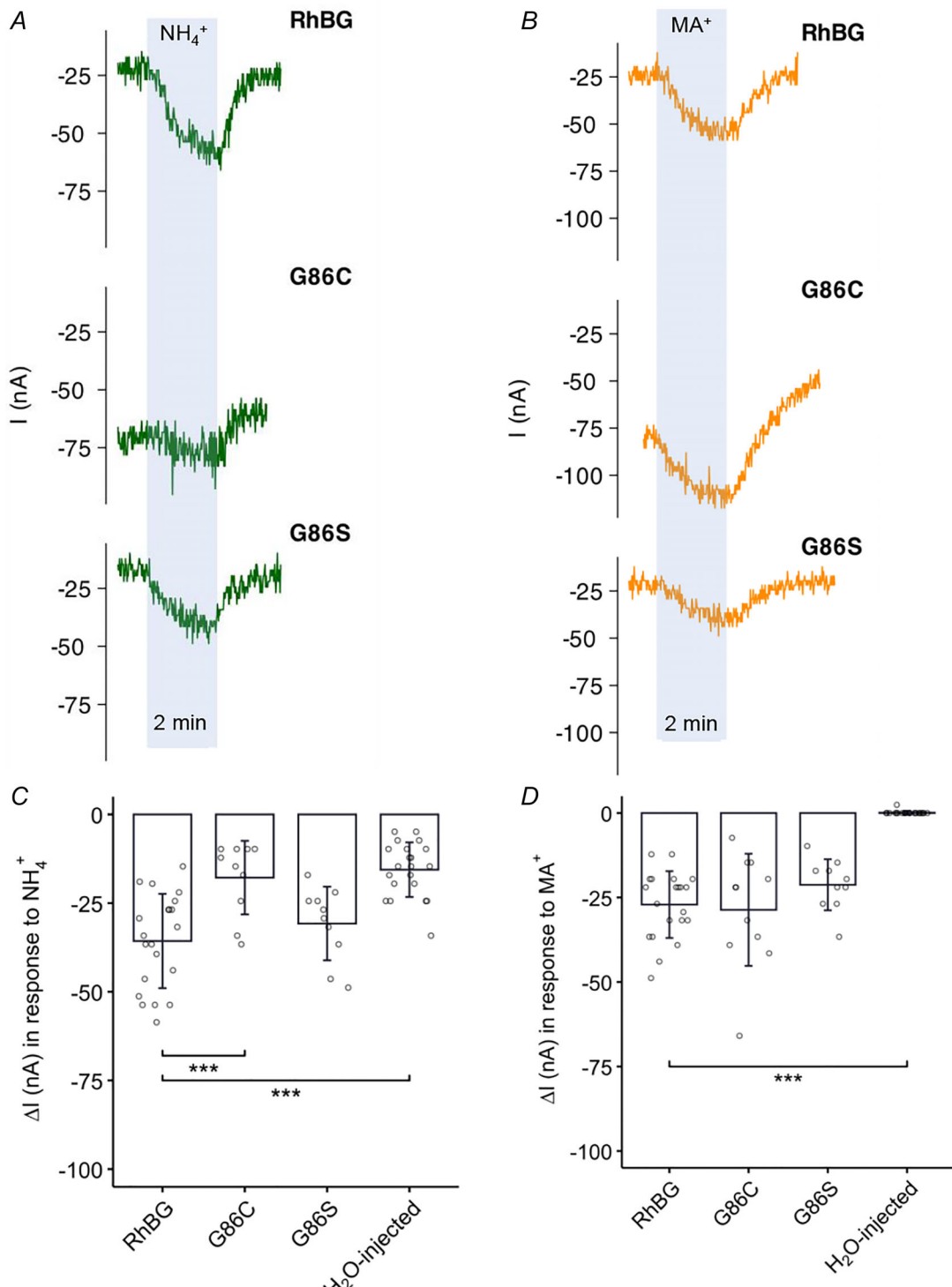

**Figure 2. Effects of G86C/S on current induced by NH₄⁺ and MA⁺**

*A*, NH₄⁺-induced current (I) in oocytes expressing G86C and G86S mutations compared to WT-RhBG. *B*, typical tracings of MA⁺-induced current of oocytes expressing G86C and G86S mutations compared to WT-RhBG. *C* and *D*, bar plots of current change (ΔI) induced by NH₄⁺ and MA⁺, respectively. Error bars were created using the SD. Individual data points represent individual oocyte measurements pooled from three independent injection batches after outlier removal. RhBG, *N* = 21; G86C, *N* = 11; G86S, *N* = 10; H₂O-injected, *N* = 21. ***P < 0.001. Statistical significance was adjusted for multiple comparisons using the false discovery rate method. Bar plots were adapted from multiple comparison tests and broken into separate plots based on mutation groups for clear illustration. For *P* values, see Table 5.

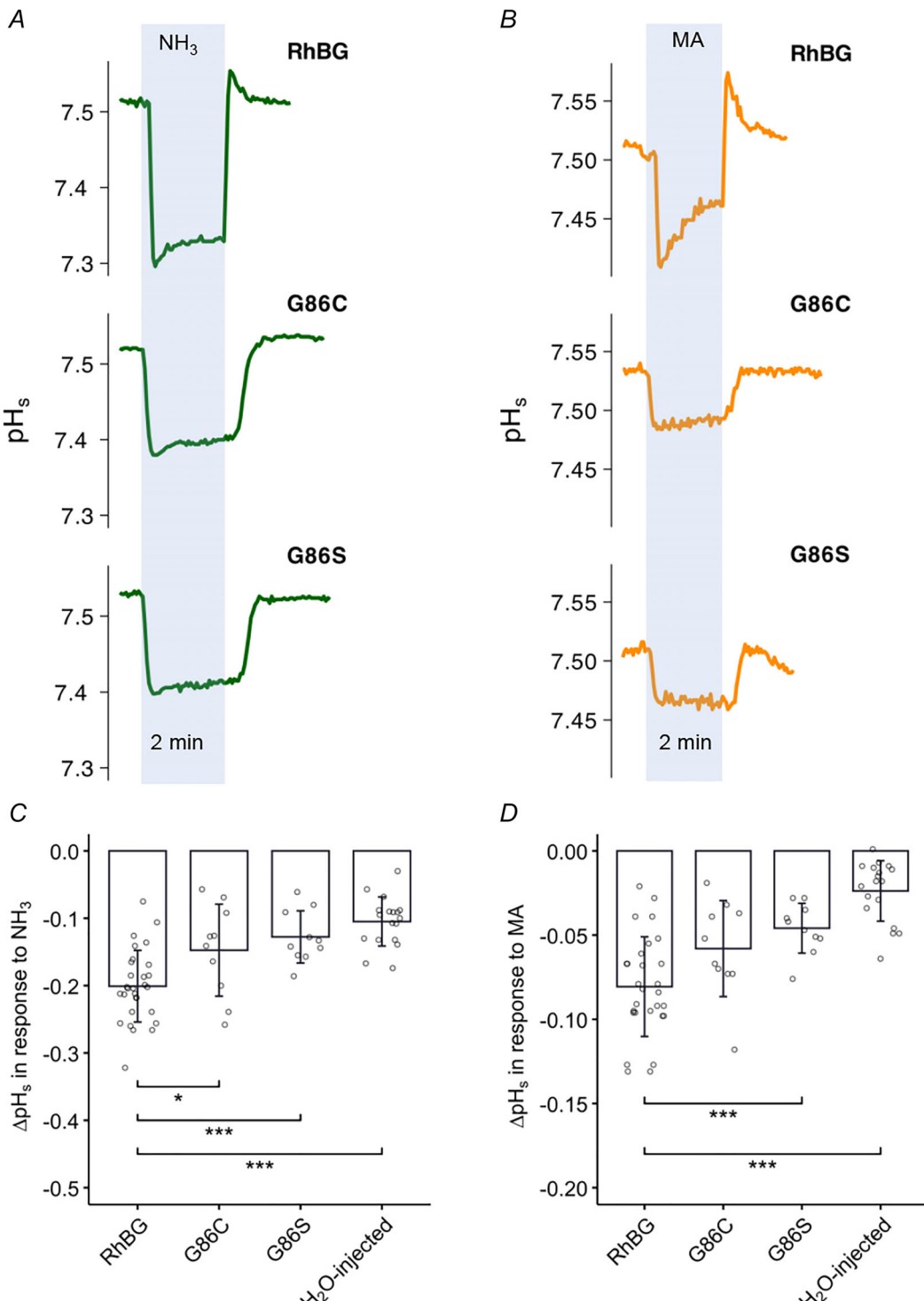

**Figure 3. Effects of G86C/S on pH$_s$ induced by NH$_3$ and MA**
*A*, tracings showing NH$_3$-induced pH$_s$ changes of G86C and G86S mutations compared to WT. *B*, tracings showing MA-induced pH$_s$ changes of G86C and G86S mutations compared to WT. *C* and *D*, summary bar plots of pH$_s$ changes (pH units) induced by NH$_3$ and MA, respectively. Error bars were created using the SD. Individual data points represent individual oocyte measurements pooled from three independent injection batches after outlier removal. RhBG, $N = 29$; G86C, $N = 10$; G86S, $N = 10$; H$_2$O-injected, $N = 17$. ***$P < 0.001$; *$P < 0.05$. Statistical significance of multiple comparisons by mixed-effects models was corrected using the false discovery rate method. Bar plots were adapted from multiple comparison tests and broken into separate plots based on mutation groups for clear illustration. For *P* values, see Table 5.

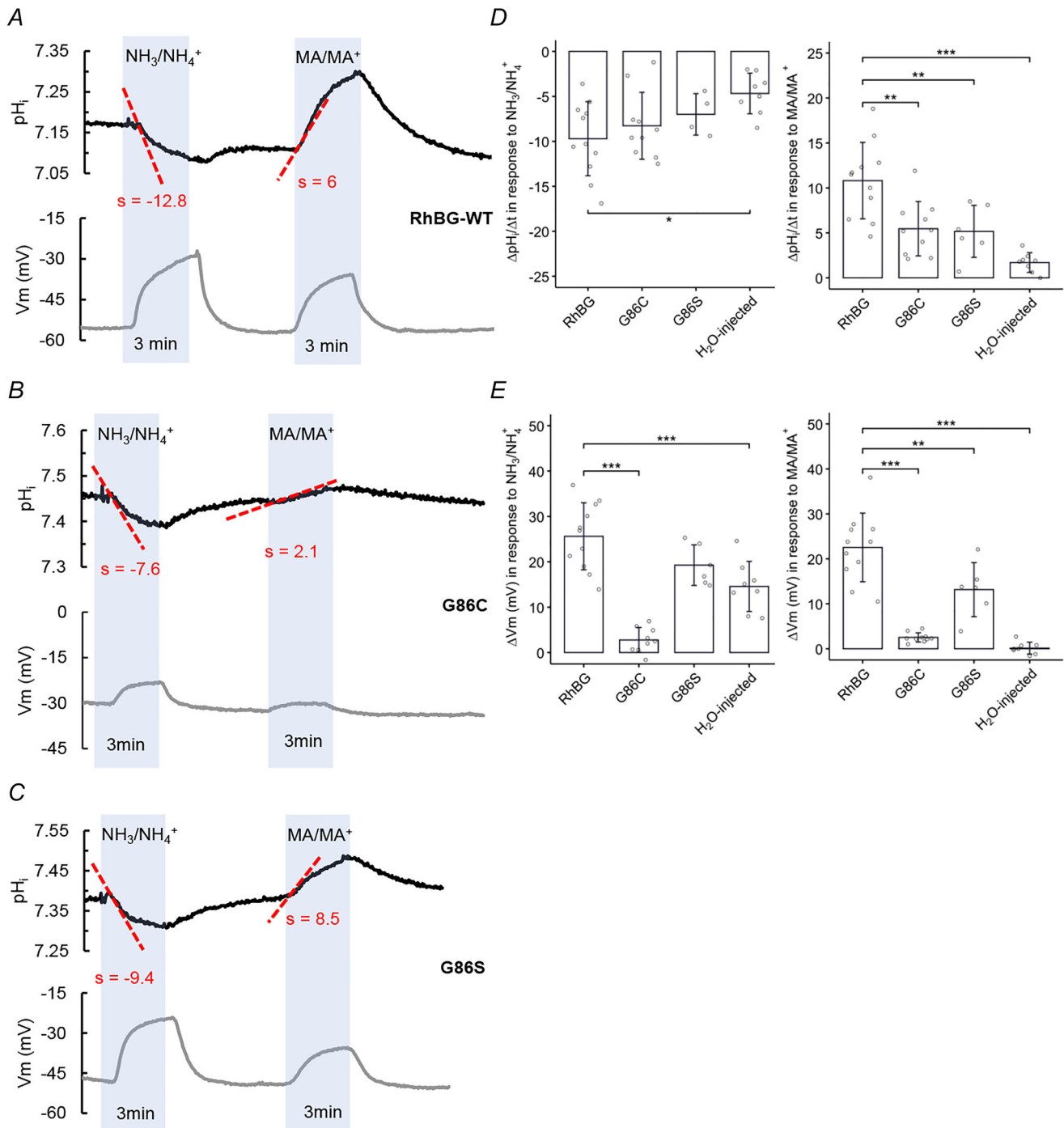

**Figure 4. Effects of NH₃/NH₄⁺ and MA/MA⁺ on intracellular pH (pHᵢ) on oocytes expressing RhBG mutants G86C/S**

pHᵢ (black tracing) and $V_m$ (grey tracing) were measured simultaneously. s, slope ($= \frac{\Delta \text{pHi}}{\Delta \text{time(s)}} \times 10^4$). *A*, effect of NH₃/NH₄⁺ and MA/MA⁺ on pHᵢ in oocytes expressing human RhBG wild type (WT-RhBG). Exposure to 5 mM NH₃/NH₄⁺ caused a rapid acidification (a decrease in pHᵢ) with a slope of −12.8 and depolarization of the cell membrane. Exposure to MA/MA⁺ also caused cell membrane depolarization but pHᵢ alkalinization. *B*, effect of NH₃/NH₄⁺ and MA/MA⁺ on pHᵢ in oocytes expressing mutant RhBG$_{G86C}$. Exposure to 5 mM NH₃/NH₄⁺ induced a rapid acidification (slope = −7.6) but a small depolarization of the cell membrane ($\Delta V_m$ = 6.9 mV). Exposure to MA/MA⁺ caused slight alkalinization of pHᵢ but only a limited depolarization of the cell membrane ($\Delta V_m$ = 2.7 mV)

compared to the oocytes expressing WT-RhBG. The steady state of $V_m$ in $RhBG_{G86C}$-expressing oocytes is relatively low ($V_m = -30$ mV) compared to oocytes expressing WT-RhBG or other RhBG mutants ($V_m < -45$ mV). *C*, effect of $NH_3/NH_4^+$ and MA/MA$^+$ on pH$_i$ in oocytes expressing mutant $RhBG_{G86S}$. Exposure to 5 mM $NH_3/NH_4^+$ caused an acidification (slope $= -9.4$) and a depolarization of the cell membrane ($\Delta V_m = 24$ mV), which are not different from RhBG expressing oocytes ($P > 0.05$). Exposure to MA/MA$^+$ also caused a small alkalinization in pH$_i$ ($P < 0.01$) and a small depolarization of the cell membrane ($\Delta V_m = 13.6$ mV, $P < 0.01$) compared to the oocytes expressing WT-RhBG. *D*, bar plot of the rates of change of pH$_i$ (slope) in oocytes exposed to $NH_3/NH_4^+$ (left) and MA/MA$^+$ (right), respectively. *E*, bar plot of $V_m$ changes ($\Delta V_m$) when oocytes are exposed to $NH_3/NH_4^+$ (left) and MA/MA$^+$ (right), respectively. Error bars were created using the SD. Individual data points represent individual oocyte measurements pooled from three independent injection batches after outlier removal. RhBG, $N = 11$; G86C, $N = 10$; G86S, $N = 6$; H$_2$O-injected, $N = 8$. \*\*\*$P < 0.001$; \*\*$P < 0.01$; \*$P < 0.05$. Statistical significance of multiple comparisons by mixed-effects models was corrected using the false discovery rate method. For $P$ values, see Table 5.

**Table 4. Intraclass Correlation Coefficients (ICC) for electrophysiological measurements across oocyte batches**

| | Measurement | Null model | Full model | | Interpretation |
| --- | --- | --- | --- | --- | --- |
| | | ICC | Unadjusted ICC | Adjusted ICC | |
| In response to ammonia/ ammonium | $\Delta I$ | 0.105 | 0.092 | 0.143 | Minimal random effect variance for batch |
| | $\Delta pH_s$ | 0.189 | 0.152 | 0.278 | Moderate random effect variance for batch |
| | $\Delta pH_i/\Delta t$ | 0.019 | NA | NA | Random effect variance for batch is zero and ICC cannot be computed |
| | $\Delta V_m$ | 0.44 | 0.176 | 0.36 | Moderate random effect variance for batch |
| | $\Delta pH_i$ | 0.244 | 0.096 | 0.11 | Minimal random effect variance for batch |
| In response to methylamine/ methylammonium | $\Delta I$ | 0.129 | 0.015 | 0.055 | Minimal random effect variance for batch |
| | $\Delta pH_s$ | 0.143 | 0.12 | 0.264 | Moderate random effect variance for batch |
| | $\Delta pH_i/\Delta t$ | 0.12 | 0.01 | 0.02 | Minimal random effect variance for batch |
| | $\Delta V_m$ | 0.414 | 0.122 | 0.435 | Moderate random effect variance for batch |
| | $\Delta pH_i$ | 0.188 | 0.003 | 0.008 | Minimal random effect variance for batch |

ICC values were calculated to assess the consistency of measurements across batches for each parameter. The null model includes only the random effect of batch, whereas the full model includes both fixed and random effects. Unadjusted and adjusted ICC values are reported where applicable. Parameters with ICC < 0.2 were interpreted as showing minimal random effect variance for batch, whereas those with ICC between 0.2 and 0.5 were interpreted as moderate. These results support the use of pooled individual data for ANOVA and non-parametric alternatives as the primary analysis. Abbreviations: $\Delta I$, delta current; $\Delta pH_s$, delta surface pH; $\Delta pH_i/\Delta t$, slope; $\Delta V_m$, delta membrane potential; $\Delta pH_i$, delta intracellular pH.

($-17.8 \pm 10.4$ nA, $N = 10$) was significantly smaller ($P < 0.001$) than in oocytes expressing WT-RhBG ($-35.7 \pm 13.3$ nA, $N = 21$). On the other hand, the $NH_4^+$-induced current in oocytes expressing $RhBG_{G86S}$ ($-30.8 \pm 10.4$ nA, $N = 10$, $P = 0.556$) was not significantly different compared to that in WT-RhBG. As shown in Fig. 2*B*, the exposure to MA/MA$^+$ solution also caused an inward MA$^+$-induced current in $RhBG_{G86C}$-expressing oocytes ($-28.6 \pm 16.6$ nA, $N = 10$, $P = 0.658$) and in $RhBG_{G86S}$-expressing oocytes ($-21.2 \pm 7.55$ nA, $N = 10$,

$P = 0.0833$) which were not significantly different from the $\Delta I$ in WT-RhBG-expressing oocytes ($-27.1 \pm 9.88$ nA, $N = 21$). Figure 2*C* and *D* summarizes these data. These results showed that G86C inhibited the electrogenic transport of $NH_4^+$, whereas G86S did not affect this ability (Table 5). The findings above suggest that the transport of $NH_4^+$ and MA$^+$ in *Xenopus* oocytes mediated by RhBG is probably through different mechanisms and amino acid G86 plays a critical role in this difference. A notable finding is that $RhBG_{G86C}$-expressing oocytes tend

**Table 5. Summary table of electrophysiological measurements in oocytes compared to WT-RhBG**

| | $NH_4^+$ | | | $NH_3$ | $MA^+$ | | MA | |
|---|---|---|---|---|---|---|---|---|
| | $\Delta I$ | slope | $\Delta V_m$ | $\Delta pH_s$ | $\Delta I$ | $\Delta V_m$ | $\Delta pH_s$ | slope |
| G86S | – | – | ↓ | ↓ | – | ↓ | ↓ | ↓ |
| P value | 0.556 | 0.283 | 0.0167 | <0.0001 | 0.0833 | <0.0001 | <0.0001 | 0.00220 |
| G86C | ↓ | – | ↓ | ↓ | – | ↓ | ↓ | ↓ |
| P value | 0.000200 | 0.437 | <0.0001 | 0.000700 | 0.658 | <0.0001 | 0.00780 | 0.000900 |
| G148R | ↓ | – | ↓ | ↓ | ↓ | ↓ | ↓ | ↓ |
| P value | 0.000100 | 0.0990 | <0.0001 | <0.0001 | <0.0001 | <0.0001 | <0.0001 | <0.0001 |
| G148W | ↓ | – | ↓ | ↓ | ↓ | ↓ | ↓ | ↓ |
| P value | 0.000100 | 0.218 | 0.0006 | <0.0001 | <0.0001 | <0.0001 | <0.0001 | <0.0001 |
| T250A | – | – | ↓ | ↓ | – | ↓ | – | – |
| P value | 0.616 | 0.283 | 0.0434 | 0.00960 | 0.894 | 0.0187 | 0.439 | 0.387 |
| T250S | – | – | ↓ | ↓ | – | ↓ | – | ↓ |
| P value | 0.227 | 0.156 | 0.001 | 0.00410 | 0.0844 | <0.0001 | 0.106 | 0.00160 |
| T250M | ↓ | ↓ | ↓ | ↓ | ↓ | ↓ | ↓ | ↓ |
| P value | 0.000300 | 0.0280 | 0.000200 | <0.0001 | <0.0001 | <0.0001 | <0.0001 | <0.0001 |
| $H_2O$ | ↓ | ↓ | ↓ | ↓ | ↓ | ↓ | ↓ | ↓ |
| P value | <0.0001 | 0.00150 | <0.0001 | <0.0001 | <0.0001 | <0.0001 | <0.0001 | <0.0001 |

Electrophysiological measurements are listed based on exposure of oocytes to either $NH_3$/$NH_4^+$ or MA/$MA^+$, with each measurement representing the transport of a specific solute ($NH_3$, $NH_4^+$, MA or $MA^+$). P values are generated from estimated marginal means by comparing mutants and water-injected control with WT-RhBG. P values are corrected for multiple comparisons using the false discovery rate method. –, no effect on functions; ↓: significantly impaired compared to WT-RhBG.

to have a lower $V_m$ than oocytes expressing other RhBG mutations or WT (data not shown) even after adjusting for batch effect. This suggests a unique impact of the G86C mutation on cell membrane ionic permeability.

*Effects of G86C and G86S mutations on $NH_3$ and MA transport: measured by $pH_s$.* Exposing oocytes expressing WT-RhBG to $NH_3$/$NH_4^+$ solution caused rapid acidification of surface pH ($pH_s$), followed by a slow recovery (Fig. 3A, tracings in the light blue shading). The acidification of $pH_s$ is probably caused by the influx of $NH_3$ into the cell. At the surface of the oocyte, influx of $NH_3$ shifts the equilibrium balance of $H^+ + NH_3 \rightleftharpoons NH_4^+$ to the left, leading to increased $H^+$ at the surface of the membrane and hence a decrease in $pH_s$. Partial recovery of $pH_s$ is probably a result of subsequent slower influx of $NH_4^+$ that causes some recovery of $pH_s$ (equilibrium is shifted to the right). $pH_s$ of $RhBG_{G86C}$- and $RhBG_{G86S}$-expressing oocytes decreased by $0.147 \pm 0.0683$ ($N = 10$, $P < 0.001$) and $0.128 \pm 0.0388$ pH units ($N = 10$, $P < 0.001$), respectively (Fig. 3C). The maximal $NH_3$-induced $pH_s$ changes of these two mutations were significantly smaller than in $RhBG$-expressing oocytes ($-0.201 \pm 0.0533$ pH units, $N = 29$).

Oocytes exposed to MA/$MA^+$ solution also showed a similar pattern of changes in $pH_s$; acidification at the cell surface followed by small recovery (Fig. 3B, tracings in the light blue area). The acidification of $pH_s$ is pre-sumably caused by influx of MA, and the recovery caused by subsequent entry of $MA^+$ (Abdulnour-Nakhoul et al., 2016; Geyer et al., 2013). The slow recovery throughout the course of exposure was more prominent in oocytes expressing WT-RhBG compared to mutations (Fig. 3B). The $pH_s$ change induced by MA in both $RhBG_{G86S}$-expressing oocytes ($-0.046 \pm 0.015$ pH units, $N = 10$, $P < 0.001$) and $RhBG_{G86C}$-expressing oocytes ($-0.058 \pm 0.029$ pH units, $N = 10$, $P = 0.00780$) was significantly smaller than that in $RhBG$-expressing oocytes ($-0.081 \pm 0.030$ pH units, $N = 29$) (Fig. 3D). This indicates that both mutants impair the ability to transport MA in oocytes. The results above suggest that amino acid G86 plays a critical role in the transport of $NH_3$ and MA in *Xenopus* oocytes mediated by RhBG.

*Effects of G86C and G86S mutations on intracellular pH changes when exposed to $NH_3$/$NH_4^+$ and MA/$MA^+$.* Figure 4 shows representative tracings of $pH_i$ change when exposed to solutions containing either $NH_3$/$NH_4^+$ or MA/$MA^+$ in oocytes expressing RhBG or its mutants G86C and G86S. Exposing the RhBG-expressing oocytes to 5 mM $NH_3$/$NH_4^+$ caused a rapid decrease in $pH_i$ (slope = $-9.71 \pm 4.12$, $N = 11$) and depolarization of the cell membrane ($\Delta V_m = 25.6 \pm 7.37$ mV) (Fig. 4A, blue area on the left). Those changes were reversible upon removal of $NH_3$/$NH_4^+$. Exposing the RhBG-expressing oocytes to 5 mM MA/$MA^+$ also resulted in depolarization of the cell membrane but caused an increase in $pH_i$

(Fig. 4*A*, blue area on the right). In oocytes expressing RhBG$_{G86C}$ (Fig. 4*B*), exposure to 5 mM NH$_3$/NH$_4^+$ caused a decrease in pH$_i$ (slope $= -8.27 \pm 3.72$, $N = 10$) but a smaller depolarization of the cell membrane ($\Delta V_m = 2.78 \pm 2.73$ mV, $N = 10$, $P < 0.001$) compared to RhBG-expressing oocytes. As described in previous studies using mouse Rhbg (Abdulnour-Nakhoul et al., 2016; Nakhoul & Lee Hamm, 2013; Nakhoul et al., 2005), the pH$_i$ decrease was induced by the entry of NH$_4^+$, which releases H$^+$ in the cytoplasm. The cell depolarization was induced by the net influx of positively charged ions which in this case is attributed to the entry of NH$_4^+$. The slopes and $\Delta$pH$_i$ (Table A1) in RhBG$_{G86C}$- and RhBG-expressing oocytes exposed to NH$_3$/NH$_4^+$ were not statistically significant ($P = 0.437$ and $P = 0.622$, respectively) but the $\Delta V_m$ results were ($P < 0.001$).

In RhBG$_{G86C}$-expressing oocytes, exposure to 5 mM MA/MA$^+$ caused a smaller ($\Delta$pH$_i = 0.079 \pm 0.045$ pH units, $N = 10$, $P < 0.001$) and slower increase in pH$_i$ (slope $= 5.46 \pm 3.03$, $N = 10$, $P < 0.001$) but only a small depolarization, compared to WT-RhBG. The increase in pH$_i$ upon exposure to MA/MA$^+$ is probably a result of the influx of MA across the cell membrane (Abdulnour-Nakhoul et al., 2016) as discussed earlier.

In oocytes expressing the other mutant of the same amino acid residue, RhBG$_{G86S}$, NH$_3$/NH$_4^+$ caused pH$_i$ decrease but not significantly different ($P = 0.283$) compared to oocytes expressing WT-RhBG, whereas the depolarization induced by NH$_3$/NH$_4^+$ was significantly smaller ($P = 0.0167$) (Fig. 4*C*). MA/MA$^+$ caused a smaller ($\Delta$pH$_i = 0.064 \pm 0.035$ pH units, $N = 6$, $P < 0.001$) and slower increase in pH$_i$ (slope $= 5.17 \pm 2.9$, $N = 6$, $P < 0.001$) compared to WT-RhBG. This slow increase in pH$_i$ was attributed to the impaired MA transport shown in the pH$_s$ results. Exposure to MA/MA$^+$ also caused depolarization in RhBG$_{G86S}$-expressing oocytes but to a lesser extent ($\Delta V_m = 13.2 \pm 6.01$ mV, $N = 6$, $P = 0.0167$) compared to oocytes expressing WT-RhBG ($\Delta V_m = 22.5 \pm 7.64$ mV, $N = 11$). Changes in pH$_i$ and $V_m$ in oocytes expressing mutants were also reversible upon removal of the testing solution. These data are summarized in Fig. 4*D* and *E* and Table 5.

### G148R and G148W

*Effects on current.* NH$_4^+$-induced inward current (Fig. 5*A*, tracings on the left) was significantly smaller in RhBG$_{G148R}$- ($-18.7 \pm 12.2$ nA, $N = 12$, $P < 0.001$) and RhBG$_{G148W}$-expressing oocytes ($-17.9 \pm 11.6$ nA, $N = 9$, $P < 0.001$) compared to NH$_4^+$ currents in oocytes expressing WT-RhBG. In oocytes expressing these two mutations, no current was recorded when exposing the oocytes to MA/MA$^+$ solution (Fig. 5*A*, tracings on the right). These results indicated impaired electrogenic transport of NH$_4^+$ and MA$^+$ in RhBG$_{G148R}$-

and RhBG$_{G148W}$-expressing oocytes. These data are summarized in Fig. 5*B* and Table 5.

*Effects on pH$_s$.* Surface pH measurements (pH$_s$) were used to investigate the effect of NH$_3$ and MA on oocytes expressing the two mutations RhBG$_{G148R}$ and RhBG$_{G148W}$. As shown in Fig. 6*A* (left tracings in blue area), exposing WT-RhBG oocytes to NH$_3$ caused the usual fast decrease of pH$_s$ followed by partial recovery. In oocytes expressing G148R, exposure to NH$_3$ caused a smaller decrease of pH$_s$ and there was no spontaneous pH$_s$ recovery. Similar inhibition was also observed in oocytes expressing G148W. The absence of spontaneous pH$_s$ recovery indicates absence of significant NH$_4^+$ transport, which is consistent with the data from current measurements as shown in Fig. 5.

Similarly, exposing oocytes to MA (right tracings) caused a typical decrease in pH$_s$ followed by partial recovery in WT-RhBG oocytes, and almost complete inhibition in G148R or G148W oocytes. These data indicate that mutations G148R and G148W completely inhibited the function of RhBG in transporting NH$_3$, NH$_4^+$, MA and MA$^+$. These data are summarized in Fig. 6*B* and Table 5.

*Effects on pH$_i$.* The effects of NH$_3$/NH$_4^+$ and MA/MA$^+$ on pH$_i$ and $V_m$ in RhBG$_{G148R}$-expressing oocytes and RhBG$_{G148W}$-expressing oocytes were inhibited compared to WT-RhBG and were similar to the effects in H$_2$O-injected oocytes. As shown in Fig. 7*A*, 5 mM NH$_3$/NH$_4^+$ caused smaller depolarization ($\Delta V_m = 12.7 \pm 6.82$ mV, $P < 0.001$) in RhBG$_{G148R}$-expressing oocytes compared to WT-RhBG oocytes ($\Delta V_m = 26.6 \pm 6.6$ mV, $N = 10$). When exposed to MA/MA$^+$, RhBG$_{G148R}$-expressing oocytes displayed a very slow increase in pH$_i$ and almost no depolarization ($P < 0.001$). Similarly, RhBG$_{G148W}$-expressing oocytes also had a smaller depolarization ($\Delta V_m = 15.9 \pm 4.77$ mV, $P < 0.001$) induced by NH$_3$/NH$_4^+$ and almost no effect of MA/MA$^+$ (tracing not shown). NH$_3$/NH$_4^+$ also caused a decrease of pH$_i$ in RhBG$_{G148R}$-expressing oocytes (slope $= -6 \pm 2.61$, $N = 8$, $P = 0.0990$) and RhBG$_{G148W}$-expressing oocytes (slope $= -6.77 \pm 3.43$, $N = 6$, $P = 0.218$) but did not reach statistical significance compared to WT-RhBG oocytes (slope $= -10.8 \pm 3.4$, $N = 10$). These data are summarized in Fig. 7*B* and *C* and Table 5. These experiments indicate that these mutations inhibited NH$_4^+$, MA and MA$^+$ transport.

### T250A, T250S and T250M

*Effects on current.* T250A/S and T250M mutations of RhBG, although at the same amino acid site, are caused by different SNPs as mentioned previously (Table 3). Figure 8 shows typical experiments demonstrating the effects of these mutations on NH$_3$/NH$_4^+$ and MA/MA$^+$

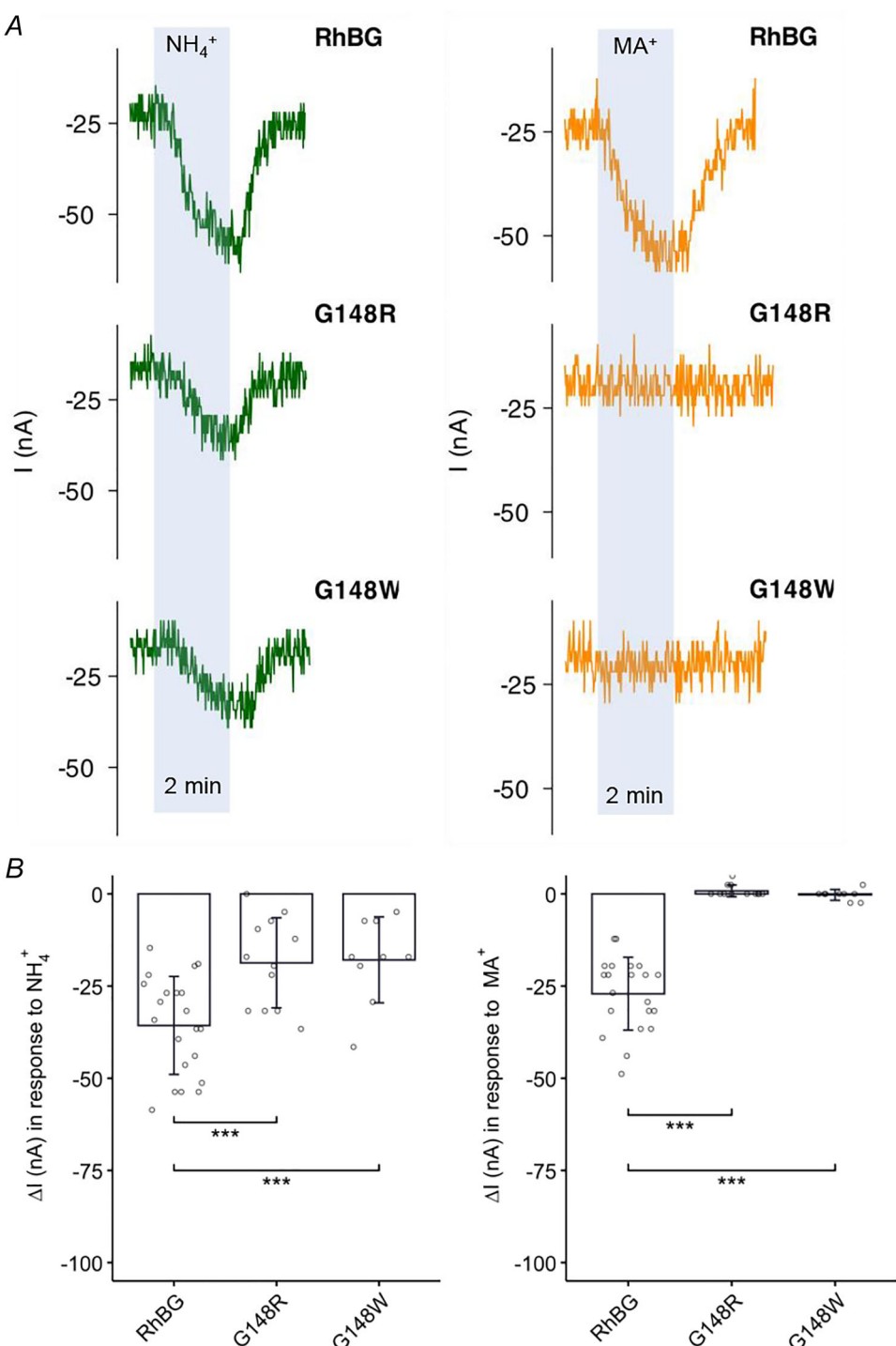

**Figure 5. Effects of G148W/R on current induced by NH₄⁺ and MA⁺**

*A*, tracings of NH₄⁺- (green) and MA⁺-induced (orange) currents (I) of G148W and G148R mutations compared to WT-RhBG. *B*, bar plot of current changes (Δ*I*) induced by NH₄⁺ (left) and MA⁺ (right), in WT-RhBG and G148W/R mutants respectively. Error bars were created using the SD. Individual data points represent individual oocyte measurements pooled from three independent injection batches after outlier removal. RhBG, $N = 21$; G148W, $N = 9$; G148R, $N = 12$. ***$P < 0.001$. Statistical significance of multiple comparisons by mixed-effects models was corrected using the false discovery rate method. For *P* values, see Table 5.

transport. The $NH_4^+$ and $MA^+$-induced inward currents in $RhBG_{T250A}$-expressing oocytes (Fig. 8*A*) and in $RhBG_{T250S}$-expressing oocytes (tracing not shown) resembled the tracing in WT-RhBG-expressing oocytes with non-significant differences in $\Delta I$ (Fig. 8*B*).

$RhBG_{T250M}$-expressing oocytes however displayed a much smaller induced current when exposed to $NH_4^+$ ($-19.5 \pm 8.04$ nA, $N = 8$, $P < 0.001$) compared to WT-RhBG-expressing oocytes ($-35.7 \pm 13.3$ nA) (Fig. 8*B* left). No current was induced when exposing

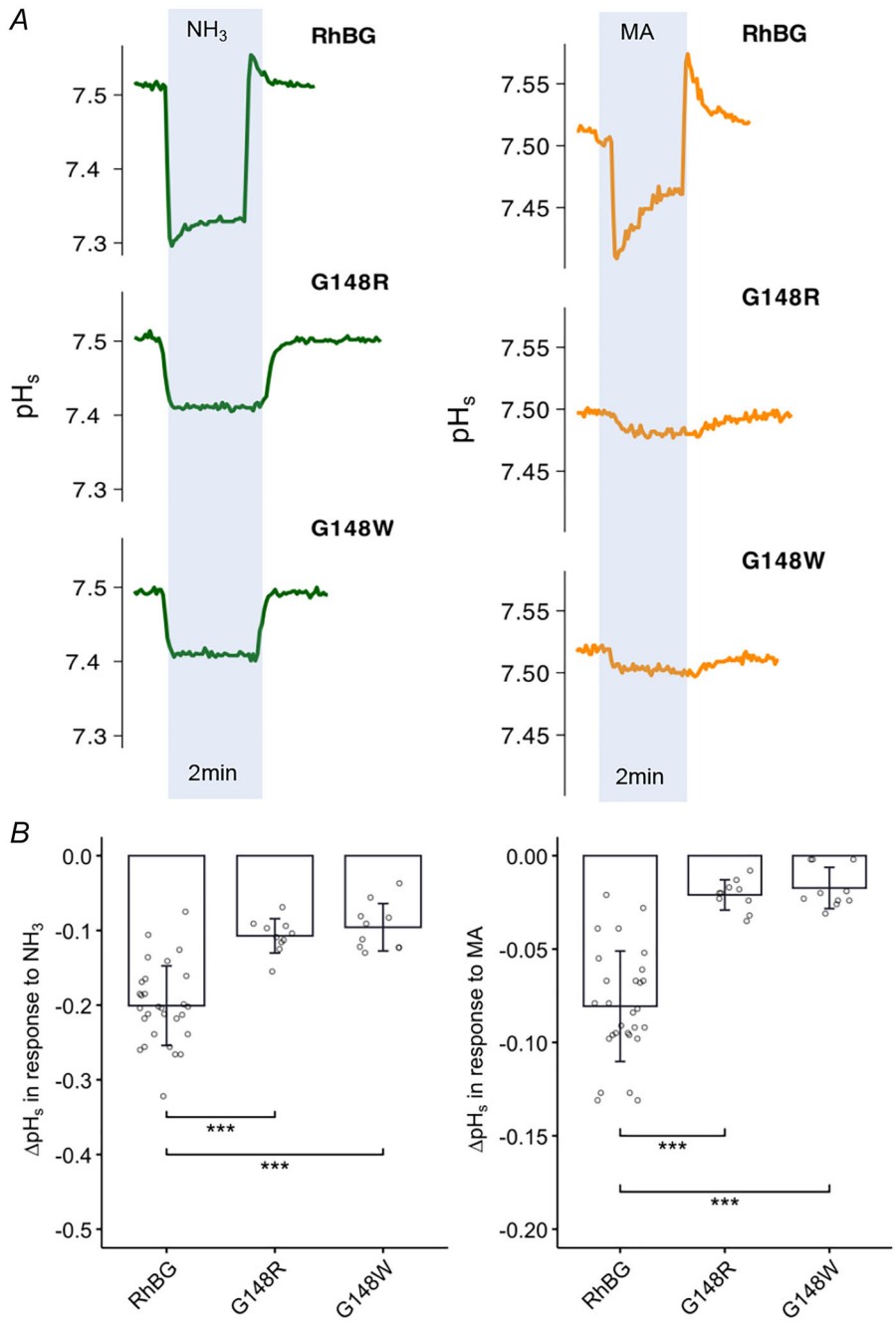

**Figure 6. Effects of G148W/R on pH$_s$ induced by NH$_3$ and MA**
*A*, tracings of NH$_3$-induced (green) and MA-induced (orange) pH$_s$ changes of G148W and G148R mutations compared to WT-RhBG. *B*, bar plot of pH$_s$ changes ($\Delta$pH$_s$) induced by NH$_3$ (left) and MA (right), respectively. Error bars were created using the SD. Individual data points represent individual oocyte measurements pooled from three independent injection batches after outlier removal. RhBG, $N = 29$; G148W, $N = 10$; G148R, $N = 10$. ***$P$ < 0.001. Statistical significance of multiple comparisons by mixed-effects models was corrected using the false discovery rate method. For $P$ values, see Table 5.

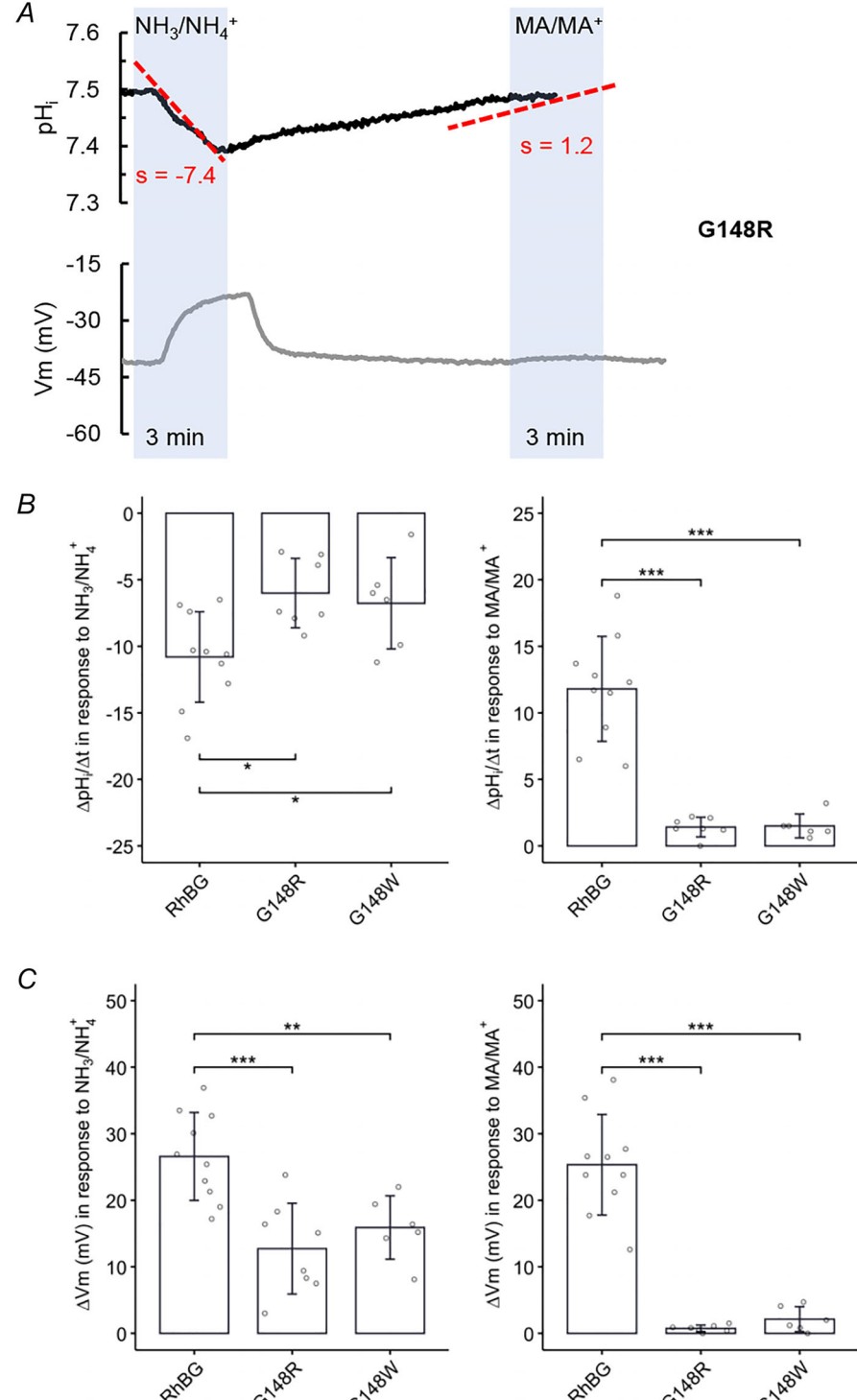

**Figure 7. Effect of NH₃/NH₄⁺ and MA/MA⁺ on pHᵢ in oocytes expressing mutant RhBG_G148R**
*A*, exposure to 5 mM NH₃/NH₄⁺ caused a slow acidification (*P* < 0.05) and small depolarization of the cell membrane (*P* < 0.001) compared to RhBG-expressing oocytes. Exposure to MA/MA⁺ caused a very slow increase in pHᵢ (slope = 1.2) and no depolarization of the cell membrane (*P* < 0.001) compared to the oocytes expressing WT-RhBG. *B*, bar plot summarizing the rates of pHᵢ change (slope) when cells are exposed to NH₃/NH₄⁺ (left) and MA/MA⁺ (right), respectively. *C*, bar plot of *V*ₘ changes (Δ*V*ₘ) of oocytes exposed to NH₃/NH₄⁺ (left) and MA/MA⁺ (right), respectively. Error bars were created the using SD. Individual data points represent individual oocyte measurements pooled from three independent injection batches after outlier removal. RhBG, *N* = 10; G148W, *N* = 6; G148R, *N* = 8. ***P* < 0.001; ***P* < 0.01; **P* < 0.05. Statistical significance of multiple comparisons by mixed-effects models was corrected using the false discovery rate method. For *P* values, see Table 5.

RhBG$_{T250M}$-expressing oocytes to MA/MA$^+$ solution (Fig. 8A, right tracings), which was similar to H$_2$O-injected oocytes and significantly different from RhBG-expressing oocytes ($P < 0.001$) (Fig. 8B right).

*Effects on pH$_s$.* Figure 9 shows typical tracings of how RhBG$_{T250}$ mutations affected NH$_3$ and MA transport. As shown in Fig. 9A (tracings in blue area), exposing RhBG oocytes to NH$_3$/NH$_4^+$ caused the usual decrease

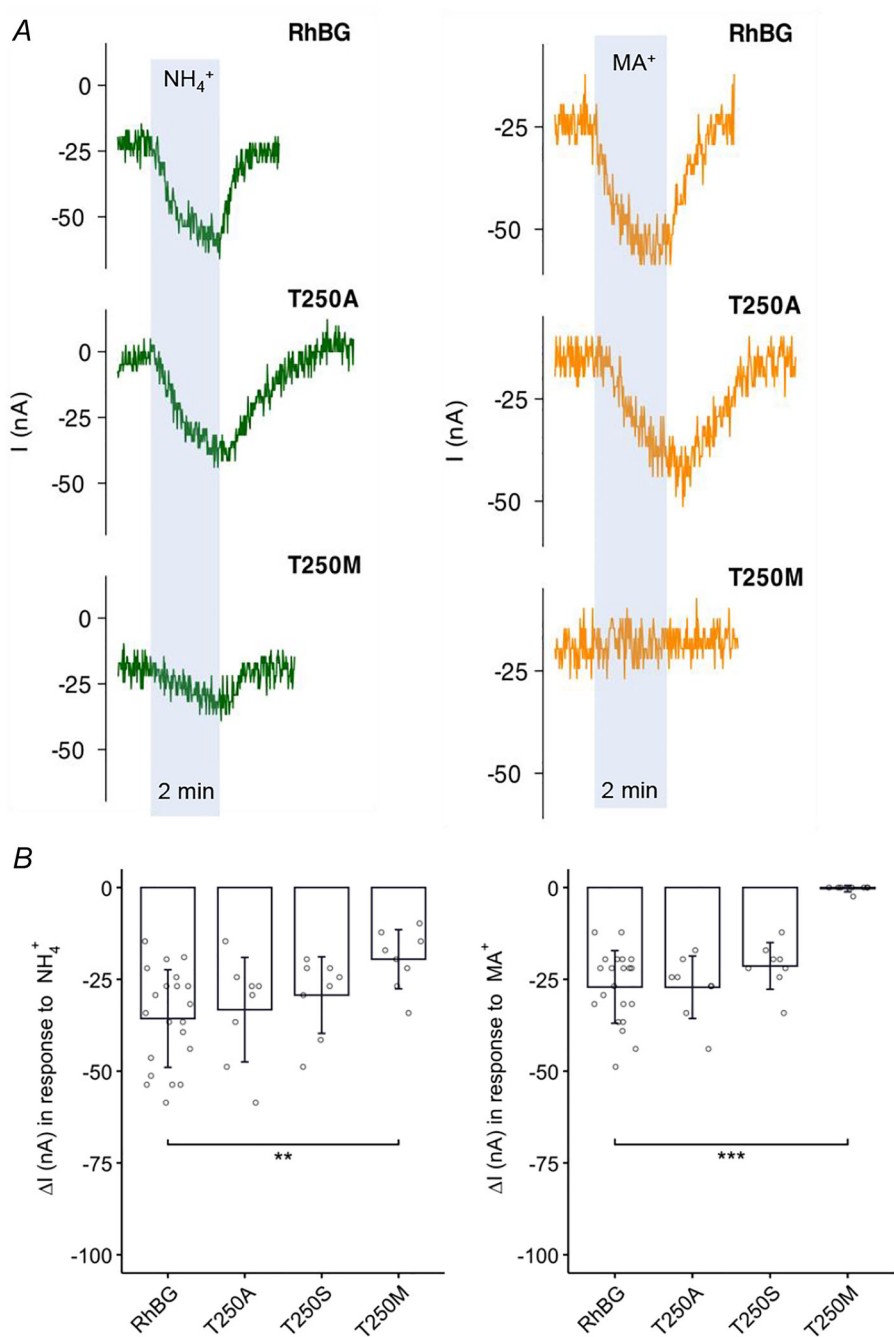

**Figure 8. Effects of T250A/S/M on current induced by NH$_4^+$ and MA$^+$**
A, NH$_4^+$-induced (green) and MA$^+$-induced (orange) current (*I*) in tracings for T250A and T250M mutations compared to WT. The tracing of T250S resembles T250A (tracing not shown). B, bar plot of current change (Δ*I*) induced by NH$_4^+$ (left) and MA$^+$ (right), respectively. Error bars were created using the SD. Individual data points represent individual oocyte measurements pooled from three independent injection batches after outlier removal. RhBG, *N* = 21; T250A, *N* = 8; T250S, *N* = 8; T250M, *N* = 8. ***$P < 0.001$; **$P < 0.01$. Statistical significance of multiple comparisons by mixed-effects models was corrected using the false discovery rate method. For *P* values, see Table 5.

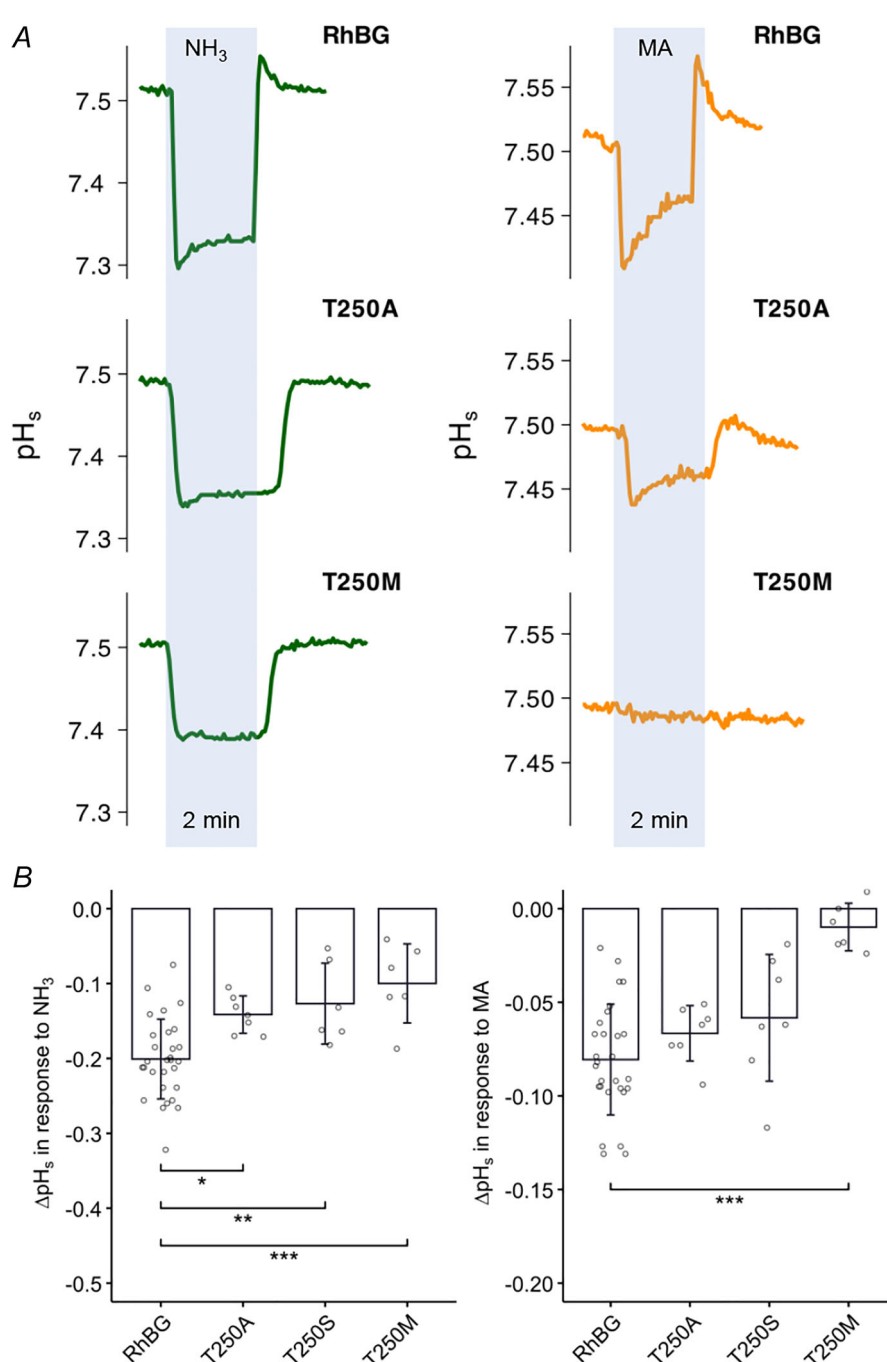

**Figure 9. Effects of T250A/S/M on pH_s induced by NH₃ and MA**

*A*, NH₃-induced (green) and MA-induced (orange) pH_s changes in tracings for T250A and T250M mutations compared to WT. The tracing of T250S resembles T250A (tracing not shown). *B*, bar plot of pH_s changes ($\Delta pH_s$) induced by NH₃ (left) and MA (right), respectively. Error bars were created using the SD. Individual data points represent individual oocyte measurements pooled from three independent injection batches after outlier removal. RhBG, $N = 29$; T250A, $N = 7$; T250S, $N = 7$; T250M, $N = 6$. ***$P < 0.001$; **$P < 0.01$; *$P < 0.05$. Statistical significance of multiple comparisons by mixed-effects models was corrected using the false discovery rate method. For *P* values, see Table 5.

in $pH_s$ followed by a partial and sustained $pH_s$ recovery. In oocytes expressing $RhBG_{T250A}$ and $RhBG_{T250M}$, the $NH_3$-induced decreases in $pH_s$ ($-0.141 \pm 0.025$ pH units, $N = 7$, $P = 0.00960$) and ($-0.127 \pm 0.054$ pH units, $N = 7$, $P = 0.00410$) respectively, were significantly inhibited compared to WT-RhBG ($-0.201 \pm 0.053$ pH units, $N = 29$). In both cases, the partial $pH_s$ recovery typically observed in WT-RhBG was absent. $RhBG_{T250S}$ showed similar inhibition as $RhBG_{T250A}$.

On the other hand, the MA-induced $pH_s$ changes were different between the two mutations, $RhBG_{T250A}$ and $RhBG_{T250M}$. As shown in Fig. 9*A* (right tracings), MA induced a typical change in $pH_s$ in $RhBG_{T250A}$, a decrease in $pH_s$ followed by partial recovery as observed in WT-RhBG. However, when $RhBG_{T250M}$ was expressed, MA did not cause any change in $pH_s$ indicating complete inhibition of MA transport. These data are summarized in Fig. 9*B* and Table 5. $RhBG_{T250S}$ behaved similarly to $RhBG_{T250A}$.

*Effects on $pH_i$.* The effects of $NH_3/NH_4^+$ and $MA/MA^+$ on $pH_i$ and $V_m$ in oocytes expressing $RhBG_{T250A}$ (Fig. 10*A*) and $RhBG_{T250S}$ (tracing not shown) resemble the effects in oocytes expressing WT-RhBG (compared with Figs 1*B* and 4*A*). In five experiments, exposure to $NH_3/NH_4^+$ caused acidification in $RhBG_{T250A}$-expressing oocytes with a slope of $-7.22 \pm 4.72$ and depolarization ($\Delta V_m = 21 \pm 4.54$ mV). In $RhBG_{T250S}$-expressing oocytes, the average slope of $pH_i$ decrease was $-6.08 \pm 2.32$ ($N = 5$) and $\Delta V_m$ was $18.7 \pm 6.25$ mV. The tracings of the other mutant at the same amino acid residue site, T250M, were similar to those of $H_2O$-injected oocytes (Fig. 10*B*). In $RhBG_{T250M}$-expressing oocytes, exposure to 5 mM $NH_3/NH_4^+$ caused a slow acidification (slope $= -4.28 \pm 2.01$, $N = 5$, $P < 0.001$) and a small depolarization of the cell membrane ($\Delta V_m = 14.3 \pm 8.24$ mV, $N = 5$, $P = 0.0125$) compared to WT-RhBG expressing oocytes (slope $= -9.2 \pm 4.08$, $\Delta V_m = 25.4 \pm 6.77$ mV, $N = 14$). Exposure to $MA/MA^+$ caused a small and slow increase in $pH_i$ (slope $= 1.54 \pm 0.97$, $N = 5$, $P < 0.001$) and no depolarization of the cell membrane ($P < 0.001$) compared to the oocytes expressing WT-RhBG (slope $= 10.5 \pm 4.19$, $\Delta V_m = 20.6 \pm 5.26$ mV, $N = 14$). These data are summarized in Fig. 10*C* and *D* and Table 5.

### Expression of RhBG and mutants at the oocyte membrane

Lastly, we investigated whether the RhBG mutations affected expression of the transporters in the cell membrane. Figure 11 demonstrated immuno-histochemistry (IHC) labelling of WT-RhBG and representative mutants G86C, G148R, T250A and T250M. The IHC results of the other mutants whose transport patterns were similar to their selected pairs at the same amino acid location (G86S, G148W and T250S) were not shown. In oocytes expressing WT-RhBG, $RhBG_{T250A}$ and $RhBG_{G86C}$, the cell membranes were positively and intensely stained. In oocytes expressing $RhBG_{T250M}$, there was weak staining at the cell membrane with minimal diffusion within the cytoplasm. Oocytes expressing $RhBG_{G148R}$ showed no staining at the cell membrane or in the cytoplasm. This expression pattern is similar to $H_2O$-injected control oocytes.

## Discussion

### CKD-related RhBG rare variants

The results with respect to identifying rare RhBG variants related to CKD presented intriguing and important insights into the genetic underpinnings of CKD, a multifactorial disorder. Our study, utilizing WES data from the CRIC study, revealed 10 rare variants with a MAF of less than 1% in both ancestry groups. Interestingly, our findings suggest a differential distribution of these rare variants across ancestry groups, with six variants exclusively detected in individuals of European descent. Conversely, two variants were specific to the African ancestry group, whereas only one variant was present in both ancestry groups. Our analyses of association tests, employing chi-squared and Fisher's exact tests (known for conservative type-1 error rates, and suitable for rare variants with common disease) (Derkach et al., 2013; Lee et al., 2018; Schaid & Sinnwell, 2010), revealed four SNPs achieving statistical significance after correction for multiple comparisons. Among these, variant rs150963900 (F81L) exhibited an OR less than one, indicative of a potential protective effect against the development of CKD in European ancestry group. Multiple predictions of RhBG protein structures using machine-learning based models (UniProt; https://www.uniprot.org) showed that F81L mutation was located at the second trans-membrane domain which is also a conserved amino acid site among Rh family and across species (human, mouse and rat). Although this makes F81L a promising and interesting candidate for further investigation, the focus of the present study was on variants that had a negative effect on CKD progression and role of F81L mutation was not investigated further. Variants rs200069134, rs575652633 and rs760016272 were implicated in six distinct types of amino acid substitution (G86S, G86C, G148R, G148W, T250A and T250S). Notably, although variant rs200320178 (T250M) did not reach statistical significance, its potential impact on CKD is underscored by its occurrence at the same amino acid position as SNP rs760016272 (T250A/S), warranting further investigation. To minimize the ancestry-related sampling bias as a result of the unequal distribution of

individuals from different ancestry groups in our study population (e.g. 73% European ancestry), we conducted separate analyses for each ancestry group to assess the robustness of associations across diverse populations. Because of the small sample size of rare variants, it was not feasible to account for population stratification based on demographic factors (age, sex, etc.) within the same ancestry group. Furthermore, the inclusion of participants based on specific clinical parameters or demographic characteristics may introduce sampling bias. Future research based on more representative population cohorts should be performed to assess the robustness of the association of rare variants with CKD. Nevertheless, the identified rare variants in relation to CKD warranted closer examination of whether they affected $NH_3/NH_4^+$ transport, an important mechanism of the role of acidosis in CKD.

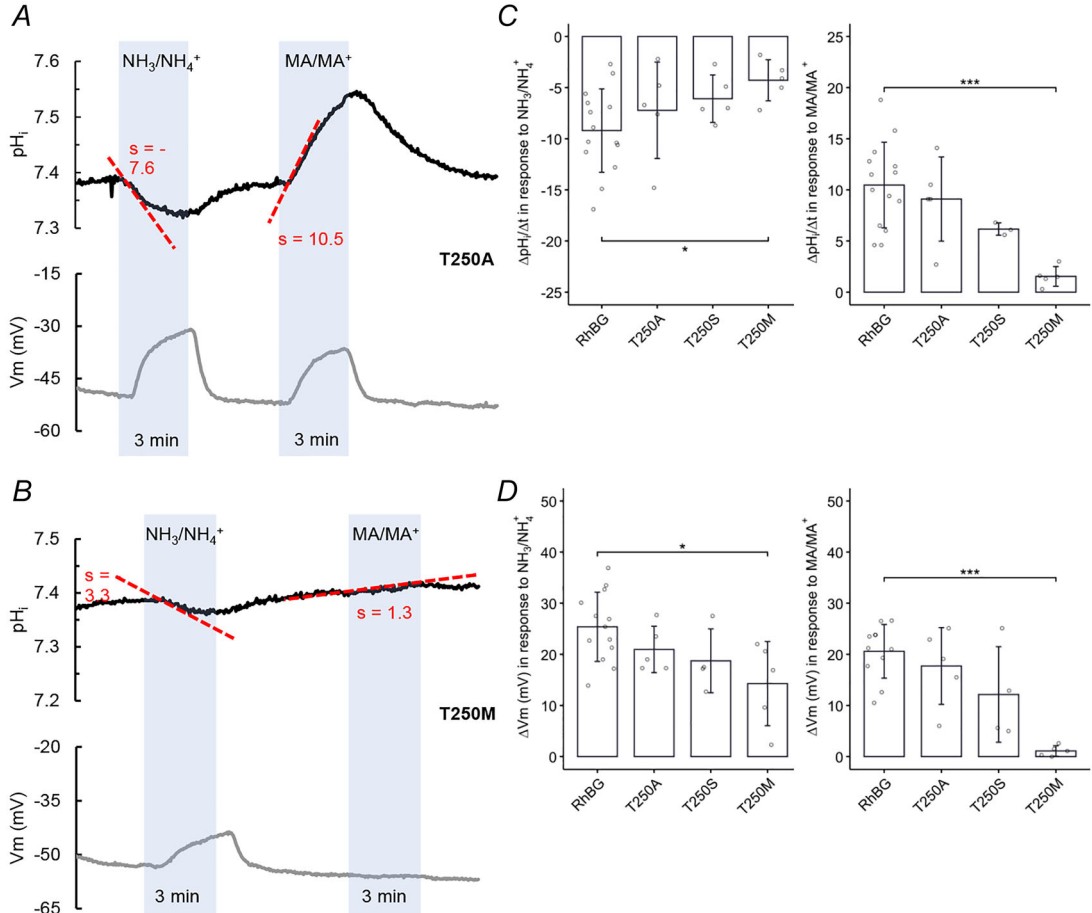

**Figure 10. Effect of $NH_3/NH_4^+$ and $MA/MA^+$ on $pH_i$ in oocytes expressing mutant $RhBG_{T250A}$ and $RhBGT_{250M}$**

*A*, in $RhBG_{T250A}$, exposure to 5 mM $NH_3/NH_4^+$ caused a decrease in $pH_i$ and depolarization of the cell membrane. Exposure to $MA/MA^+$ caused pH alkalinization and a depolarization of the cell membrane. The results of those measurements are not different from those in oocytes expressing WT-RhBG ($P > 0.05$). Effects of $NH_3/NH_4^+$ and $MA/MA^+$ on $pH_i$ in $RhBG_{T250S}$-expressing oocytes (tracing not shown) showed a similar pattern to $RhBG_{T250A}$. *B*, in oocytes expressing RhBG mutant T250M, exposure to 5 mMM $NH_3/NH_4^+$ caused a slow acidification ($P < 0.05$) and a small depolarization of the cell membrane ($P < 0.05$) compared to oocytes expressing WT-RhBG. Exposure to $MA/MA^+$ caused a much slower increase in $pH_i$ (slope = 1.3, $P < 0.001$) and no depolarization of the cell membrane ($P < 0.001$) compared to oocytes expressing WT-RhBG. *C*, bar plot summarizing the rates of $pH_i$ changes (slope) in oocytes exposed to $NH_3/NH_4^+$ (left) and $MA/MA^+$ (right), respectively. *D*, bar plot of $V_m$ change ($\Delta V_m$) in oocytes exposed to $NH_3/NH_4^+$ (left) and $MA/MA^+$ (right), respectively. Error bars were created using the SD. Individual data points represent individual oocyte measurements pooled from three independent injection batches after outlier removal. RhBG, $N = 14$; T250A, $N = 5$; T250S, $N = 5$; T250M, $N = 5$. \*\*\*$P < 0.001$; \*$P < 0.05$. Statistical significance of multiple comparisons by mixed-effects models was corrected using the false discovery rate method. For *P* values, see Table 5.

## Function and expression of identified mutations

The complete functional profile of mutations and adjusted $P$ values are summarized in Table 5. The functional tests described here in detail confirmed the effects of expressing the human RhBG on current, intracellular pH and surface pH changes in oocytes exposed to $NH_3/NH_4^+$ and $MA/MA^+$ solutions. Previous research (Ludewig, 2004) utilizing similar methodologies demonstrated the inward current induced by $NH_4^+$ in RhBG-expressing oocytes but failed to observe such current with $MA^+$ exposure. Notably, Ludewig (2004) only measured the current when exposed to $MA^+$ at the concentration of 1 mM. In the present study, a higher concentration of $MA^+$ (5 mM) did induce an inward current, as reported previously (Nakhoul, Abdulnour-Nakhoul, Boulpaep, et al., 2010). Our study provides new insights by integrating current measurements with direct measurements of intracellular and surface pH in RhBG-expressing oocytes exposed to $NH_3/NH_4^+$ and $MA/MA^+$. Other studies focused on the mouse homologue (Rhbg) using various methods in different expression systems (Abdulnour-Nakhoul et al., 2016; Caner et al., 2015; Mak et al., 2006; Nakhoul et al., 2005, 2006; Nakhoul, Abdulnour-Nakhoul, Boulpaep,

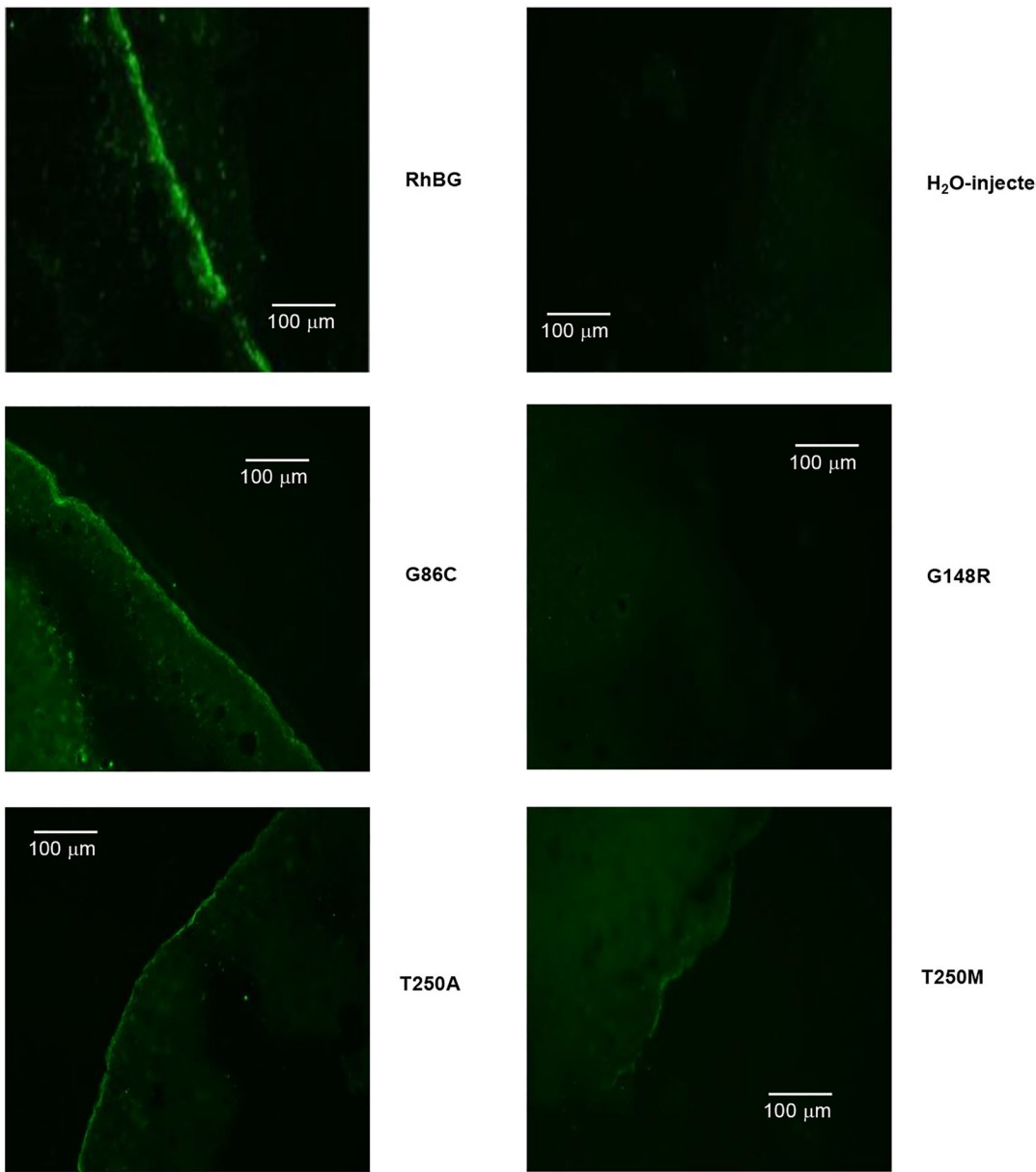

**Figure 11. Immunohistochemistry labelling of human RhBG, G86C, G148R, T250A, T250M in oocytes**
RhBG, T250A and G86C-expressing oocytes showed positive staining on the cell membrane. Labelling of T250M was weak at the cell membrane. No labelling was detected in G148R and $H_2O$-injected oocytes.

et al., 2010; Westhoff & Wylie, 2006). Given that protein isoforms of the same family may play diverse physiological roles in different species or across tissues within the same species it is possible that human RhBG and mouse Rhbg can differ in functions. It was therefore important to confirm that the transport pattern of human RhBG is similar to its mouse homologue Rhbg, as our results show. This provides further biological significance and relevance for studying ammonia transporters using homologues.

Our previous studies show that Rh-B glycoprotein transports both $NH_3$ and $NH_4^+$ (Caner et al., 2015; Nakhoul et al., 2005, 2006; Nakhoul, Abdulnour-Nakhoul, Schmidt, et al., 2010). The transport mechanisms of RhBG have been extensively studied using various experimental approaches, including radioactive uptake assays, pH measurements and stopped-flow fluorimetry (Geyer et al., 2013; Ludewig, 2004; Mak et al., 2006; Zidi-Yahiaoui et al., 2005). Although these studies provided valuable insights into RhBG function, the different methodological approaches have yielded varying interpretations regarding the relative contributions of electrogenic *vs.* electroneutral transport mechanisms. The technical challenges inherent in distinguishing electrogenic from electroneutral transport mechanisms in Amt/Mep/Rh proteins, including methodological inconsistencies and the complexity of multiple potential energetic pathways, have been comprehensively reviewed (Williamson et al., 2024). Although *Xenopus* oocytes contain endogenous $NH_4^+$-permeable channels that may contribute to baseline $NH_4^+$-induced currents, the significant differences observed between water-injected controls and RhBG-expressing oocytes demonstrate that the presence of RhBG contributes to the measured electrogenic responses (Fig. 1). This differential response pattern allows reliable detection of variant-specific functional differences, which is the primary objective of this comparative study. To accurately assess these transport differences, we used a combination of measurements, as described in the Methods. Electrogenic $NH_4^+$ influx (or $MA^+$) is detected as a decrease in pHᵢ, an inward current, depolarization of the membrane potential ($V_m$) and an increase in surface pH (pHₛ). On the other hand, $NH_3$ influx is detected as a decrease in surface pH or as a MA-induced increase in intracellular pH (pHᵢ). Consistent with previous findings (Nakhoul, Abdulnour-Nakhoul, Boulpaep, et al., 2010), $NH_4^+$ induced intracellular acidification, whereas $MA^+$ caused initial alkalinization, reflecting the distinct transport mechanisms for these substrates. Our findings show that there are subtle differences in how the identified mutations affect either $NH_3$ or $NH_4^+$ and their surrogate substrates MA and $MA^+$. This complexity revealed by our study suggests that RhBG transport properties may be more nuanced than previously recognized, with different substrates potentially utilizing different transport properties.

The investigation into the effects of RhBG mutations provided valuable information about their functional roles. The findings revealed distinct and interesting impact patterns of mutations G86C and G86S on $NH_3/NH_4^+$ and $MA/MA^+$ transport. Both G86C and G86S affected $NH_3$ and MA transport as evidenced by smaller $\Delta pH_s$ than WT-RhBG. However, G86C appeared to be more detrimental than G86S based on the results of other measurements (effects on current and membrane potential). G86S retained the ability to transport $NH_4^+$ and $MA^+$ but not the uncharged form $NH_3$ and MA. It is intriguing that the substitution of glycine (G) by serine (S) appeared to disrupt the function of RhBG as a gas channel. It is possible that the hydroxyl group in serine makes it more hydrophilic than glycine, thus leading to an altered gas binding affinity or stabilization of the molecule in the hydrophobic pore of the transporter. The impact of G86C appeared to be more complex. The unique thiol group (-SH) in cysteine (C) makes the residue not only more polar, resulting in gas affinity change, but also highly reactive. Another possibility is that cysteine is known to contribute to many protein-protein interactions that are important for the structural integrity of proteins (Fra et al., 2017; Gutierrez et al., 2018; Jakob et al., 2023; Shi et al., 2021). This change most probably affects the RhBG structure and stability given that residue 86 is a site where the transmembrane domain turns into cytoplasmic loop (UniProt). This could also potentially explain the constantly low membrane potential in RhBG$_{G86C}$-expressing oocytes (Fig. 4*B*). In the pHᵢ experiments of G86C the membrane potential ($V_m$) changes were atypically small upon exposure to $NH_4^+$ and $MA^+$. One possible explanation is that, as a result of altered membrane permeability (low $V_m$), the extent of depolarization was limited in response to $NH_4^+$ and $MA^+$ in RhBG$_{G86C}$-expressing oocytes, which led to a significantly smaller $\Delta V_m$. The $\Delta V_m$ in RhBG$_{G86C}$-expressing oocytes when exposed to $NH_4^+$ was even smaller than it is in $H_2O$-injected oocytes (Fig. 4*E*) ($P < 0.001$, statistics not shown), suggesting that even endogenous total ammonia transport in oocytes was affected. The possible inhibition of endogenous $NH_3/NH_4^+$ transporters in oocytes can also explain the discrepancy in current change ($\Delta I$) in response to $NH_4^+$ (reduced) and $MA^+$ (not affected). Only current measurements related to $NH_4^+$ were inhibited because native oocytes do not express $MA^+$ transporter as described in previous studies (Caner et al., 2015; Nakhoul, Abdulnour-Nakhoul, Boulpaep, et al., 2010) and the results of the present study (Figs 1*A* and 2*D*). This is additional evidence of disrupted membrane integrity and protein–protein interaction.

The findings from the expression of mutations G148R and G148W in RhBG oocytes demonstrated that they are the most detrimental to RhBG functions of all the mutations tested in the present study. The immuno-fluorescence staining results confirmed the absence of these two mutant proteins at the cell membrane. Residue 148 is located in the middle of a cytoplasmic loop and is conserved among RhBG homologues, suggesting this region plays a crucial structural role in RhBG function.

T250A/S and T250M mutations exhibited distinct effects on $NH_3/NH_4^+$-induced current responses. Although $RhBG_{T250A}$- and $RhBG_{T250S}$-expressing oocytes behaved similarly to RhBG-expressing oocytes, $RhBG_{T250M}$-expressing oocytes behaved like $H_2O$-injected oocytes. T250M demonstrated an inhibitory effect on transporting all substrates tested ($NH_3$, $NH_4^+$, MA and $MA^+$). $RhBG_{T250A}$ and $RhBG_{T250S}$ mutants maintained the overall characteristic of electrogenic $NH_4^+/MA^+$ transport, yet both mutants partially inhibited the depolarization ability compared to WT-RhBG. They impaired the transport of $NH_3$ but not MA based on the measurements of surface pH. According to the 3D protein structure established by AlphaFold (UniProt), the T250 residue is located on the inner side (facing the centre of the pore) of a pore formed by 11 transmembrane helices and close to the extracellular loop. In its bacterial homologue AmtB, the residue aligned with T250 in human RhBG is located on the extracellular and inner side of a pore through which $NH_3$ molecules are supposedly conducted (Callebaut et al., 2006). Substitution of threonine to alanine and serine altered $NH_3$ transport. This suggested that the methyl group (–$CH_3$) of threonine at residue 250 site may play a vital role in gas transport ($NH_3$) but not $NH_4^+$. These findings add insights to the current understanding of the distinct mechanisms of $NH_3$ and MA transport by RhBG.

## Mechanistic classification: membrane expression defects *vs*. per-molecule activity impairments

Our combined electrophysiological and immuno-histochemical analyses reveal that RhBG mutations impair transport through distinct molecular mechanisms, providing critical insights into structure–function relationships and potential therapeutic targets.

Mutations causing membrane expression defects represent one major category of functional impairment. The G148R (Fig. 11) and G148W mutations result in complete absence of membrane expression, with corresponding total loss of transport function. This suggests that the G148 residue, located in a conserved cytoplasmic loop, is critical for proper protein folding, trafficking, or membrane stability.

By contrast, several mutations demonstrate per-molecule activity defects despite normal membrane expression. The T250A and T250S mutations maintain normal membrane localization but exhibit selective $NH_3$ transport impairment, suggesting these residues are specifically involved in the $NH_3$ transport pathway. The G86C and G86S mutations show similar patterns of normal membrane expression but differential transport effects. The G86C mutation demonstrates an ~50% reduction in $NH_4^+$ transport, whereas G86S maintains normal electrogenic transport. These differential effects indicate that G86 is specifically required for optimal charge-selective transport rather than protein membrane expression, and that different substitutions at residue G86 affect distinct aspects of the transport mechanism.

The T250M mutation represents a distinct hybrid defect. This mutation causes severely reduced membrane expression with complete functional impairment, suggesting this residue is critical for both protein membrane expression and transport activity. Unlike pure expression defects (G148R/W) where reduced expression fully accounts for functional loss, T250M demonstrates disproportionate functional impairment relative to its expression level, indicating that T250M specifically affects both membrane expression and transport function, contrasting with T250A/S, which only impairs transport activity.

## Heterozygous complexity: from homotrimeric models to heterozygous carrier

Although the mechanistic classification provides insights into individual mutation effects, there is an important consideration for translating these findings to clinical contexts involving the functional complexity of RhBG mutation in heterozygous patients. Functional Rhbg probably exists as a trimer (Conroy et al., 2005). It is valid to consider that the homotrimeric expression of RhBG mutants in the present study may not fully represent the heterotrimeric complexes present in heterozygous patients. Our findings represent the 'worst case scenario' for each mutation, as the functional tests were conducted on homotrimeric complexes. In heterozygous patients, the functional impact would be considerably more complex because of different inheritance patterns of the gene and different assembly patterns of WT and mutant subunits into trimeric complexes.

To fully address this complexity, two key biological factors must be considered: inheritance patterns and trimeric assembly mechanisms. Considering different inheritance patterns, the functional consequences could vary dramatically. Under a dominant-negative inheritance pattern, mutant subunits would actively interfere with WT subunit function, potentially rendering

entire trimers non-functional even when containing normal subunits. By contrast, a co-dominant pattern would result in proportional functional reduction based on the contribution of each subunit type, whereas haploinsufficiency would primarily affect total protein levels rather than individual trimer function. Furthermore, these inheritance patterns are not mutually exclusive and can co-exist within the same mutation, creating complex phenotypes where haploinsufficiency, co-dominant effects and dominant-negative interference may simultaneously contribute to the overall functional impairment in heterozygous carriers.

Beyond inheritance patterns, the actual assembly of subunits into functional trimers introduces additional mechanistic complexity. Several assembly scenarios are possible in heterozygous carriers, which can be categorized based on expression levels and assembly mechanisms. With equal allelic expression and statistical assembly, trimers would contain varying ratios of wild-type and mutant subunits, theoretically producing 12.5% all-WT trimers, 75% heterotrimers and 12.5% all-mutant trimers. However, assembly patterns may deviate from statistical distribution if mutations alter subunit stability, folding kinetics or assembly preferences. Additionally, cellular quality control mechanisms might selectively affect mutant subunits, influencing final trimeric composition. Furthermore, unequal allelic expression would alter subunit ratios, and non-statistical assembly mechanisms could involve preferential incorporation, cooperative effects, or selective degradation processes. The inheritance and subunit assembly patterns for RhBG mutations remain poorly characterized in the current literature, and determining these mechanisms represents an important area for future investigation. Importantly, our homotrimeric functional studies provide a robust and valid characterization of individual mutation effects, establishing critical foundational data that will inform future heterozygous studies.

In summary, the present study provides genetic evidence linking RhBG variants to CKD risk using data from the established CRIC cohort. Our functional validation demonstrates that rare naturally occurring RhBG variants impair renal ammonia transport through distinct mechanisms: complete blockade (G148R/W and T250M) and selective substrate defects (T250A/S and G86C/S). These findings advance mechanistic understanding of RhBG-mediated transport by identifying structural determinants that affect function and membrane expression. Moreover, the results suggest a molecular basis for clinical observations linking reduced ammonia excretion to CKD progression, independent of overt acidosis. Although validation of these associations requires further research using larger, representative cohorts to confirm CKD risk relationships, this integrated genetic and functional approach offers a genetic framework for patient stratification and personalized acid–base management, potentially identifying high-risk patients before clinical acidosis develops.

# Appendix

## Methods

**Mixed-effects model analysis.** Mixed-effects models were fitted using the lme4 package in R, with experimental groups as fixed effects and injection batch as a random effect. Model assumptions were verified through residual plots and normality tests. Complete model outputs including coefficients, standard errors, $t$ values, and confidence intervals are presented in the Appendix (Table A1).

## Univariate statistical analysis

All variables in electrophysiological measurements were compared using one-way ANOVA followed by Dunnett's test for normally distributed data or Kruskal–Wallis $H$ test followed by Dunn's test (non-parametric ANOVA *post hoc* test). In *post hoc* tests, we set RhBG-expressing group as the reference. $P$ value correction for family-wise error was built into those *post hoc* tests. Normality was tested by Shapiro's test. Outliers were identified using 'above Q3 + 1.5 × IQR or below Q1 − 1.5 × IQR' method and examined on dot plots. Summary data were reported as the mean ± SD. Each experimental group size was initially equally represented across all batches, though final group sizes may vary slightly as a result of outlier exclusion applied uniformly across all groups.

**Table A1. Fixed effects estimates from linear mixed-effects models of measurements in response to ammonia/ammonium**

| | | Estimate | SE | d.f. | t value | $P_r(>|t|)$ | | Adjusted P | |
|---|---|---|---|---|---|---|---|---|---|
| $\Delta I$ | (Intercept) | −35.589 | 2.769 | 27.139 | −12.851 | $4.71 \times 10^{-13}$ | *** | | |
| | Experiment G86C | 16.845 | 4.091 | 97.781 | 4.117 | $8.02 \times 10^{-5}$ | *** | 0.0002 | *** |
| | Experiment G86S | 4.545 | 4.101 | 97.892 | 1.108 | 0.270498 | | 0.5561 | |
| | Experiment G148R | 16.937 | 3.903 | 97.812 | 4.339 | $3.49 \times 10^{-5}$ | *** | 0.0001 | *** |
| | Experiment G148W | 18.26 | 4.267 | 97.686 | 4.279 | $4.38 \times 10^{-5}$ | *** | 0.0001 | *** |
| | Experiment T250A | 2.273 | 4.446 | 97.881 | 0.511 | 0.610325 | | 0.6161 | |
| | Experiment T250S | 6.24 | 4.446 | 97.881 | 1.404 | 0.163577 | | 0.2272 | |
| | Experiment T250M | 18.049 | 4.532 | 97.003 | 3.983 | 0.000132 | *** | 0.0003 | *** |
| | Experiment H2O | 20.123 | 3.179 | 94.009 | 6.33 | $8.29 \times 10^{-9}$ | *** | <0.0001 | *** |
| $\Delta pH_s$ | (Intercept) | −0.19817 | 0.01131 | 12.2913 | −17.516 | $4.61 \times 10^{-10}$ | *** | | |
| | Experiment G86C | 0.05105 | 0.01408 | 93.72974 | 3.625 | 0.000469 | *** | 0.0007 | *** |
| | Experiment G86S | 0.07065 | 0.01408 | 93.72974 | 5.017 | $2.48 \times 10^{-6}$ | *** | <0.0001 | *** |
| | Experiment G148R | 0.09105 | 0.01408 | 93.72974 | 6.466 | $4.50 \times 10^{-9}$ | *** | <0.0001 | *** |
| | Experiment G148W | 0.10255 | 0.01408 | 93.72974 | 7.283 | $1.00 \times 10^{-10}$ | *** | <0.0001 | *** |
| | Experiment T250A | 0.04498 | 0.01683 | 94.73779 | 2.673 | 0.008861 | ** | 0.0096 | ** |
| | Experiment T250S | 0.05385 | 0.01783 | 94.51925 | 3.019 | 0.003256 | ** | 0.0041 | ** |
| | Experiment T250M | 0.09086 | 0.01778 | 94.46998 | 5.111 | $1.67 \times 10^{-6}$ | *** | <0.0001 | *** |
| | Experiment H2O | 0.09788 | 0.01173 | 93.4398 | 8.347 | $6.18 \times 10^{-13}$ | *** | <0.0001 | *** |
| $\Delta V_m$ | (Intercept) | 28.805 | 1.88 | 12.412 | 15.323 | $1.96 \times 10^{-9}$ | *** | | |
| | Experiment G86C | −24.851 | 2.689 | 79.724 | −9.241 | $3.04 \times 10^{-14}$ | *** | <0.0001 | *** |
| | Experiment G86S | −7.607 | 3.007 | 77.268 | −2.53 | 0.013457 | * | 0.0167 | * |
| | Experiment G148R | −15.141 | 2.745 | 79.028 | −5.516 | $4.24 \times 10^{-7}$ | *** | <0.0001 | *** |
| | Experiment G148W | −11.593 | 3.057 | 77.573 | −3.792 | 0.000294 | *** | 0.0006 | *** |
| | Experiment T250A | −6.749 | 3.248 | 76.749 | −2.078 | 0.041049 | * | 0.0434 | * |
| | Experiment T250S | −11.509 | 3.248 | 76.749 | −3.544 | 0.000676 | *** | 0.001 | *** |
| | Experiment T250M | −13.429 | 3.248 | 76.749 | −4.135 | $9.00 \times 10^{-5}$ | *** | 0.0002 | *** |
| | Experiment H2O | −12.871 | 2.221 | 77.413 | −5.795 | $1.40 \times 10^{-7}$ | *** | <0.0001 | *** |
| $\Delta pH_i$ | (Intercept) | −0.09809 | 0.008714 | 12.06691 | −11.257 | $9.31 \times 10^{-8}$ | *** | | |
| | Experiment G86C | 0.009949 | 0.015753 | 69.08451 | 0.632 | 0.52975 | | 0.6217 | |
| | Experiment G86S | 0.010416 | 0.022285 | 78.89831 | 0.467 | 0.64151 | | 0.6483 | |
| | Experiment G148R | 0.01805 | 0.016935 | 75.20293 | 1.066 | 0.2899 | | 0.6073 | |
| | Experiment G148W | 0.023436 | 0.01906 | 77.3808 | 1.23 | 0.22259 | | 0.6073 | |
| | Experiment T250A | 0.015628 | 0.020376 | 78.80194 | 0.767 | 0.44539 | | 0.6087 | |
| | Experiment T250S | 0.018428 | 0.020376 | 78.80194 | 0.904 | 0.36854 | | 0.6085 | |
| | Experiment T250M | 0.058028 | 0.020376 | 78.80194 | 2.848 | 0.00561 | ** | 0.05 | . |
| | Experiment H2O | 0.036311 | 0.013887 | 78.93319 | 2.615 | 0.01069 | * | 0.05 | . |

(Continued)

**Table A1. (Continued)**

| | | Estimate | SE | d.f. | $P_r(>|t|)$ | | Adjusted $P$ | |
|---|---|---|---|---|---|---|---|---|
| ΔpH$_i$/Δt | (Intercept) | −9.35 | 0.6535 | 78 | $<2 \times 10^{-16}$ | *** | | |
| | Experiment G86C | 1.08 | 1.3392 | 78 | 0.422449 | | 0.4368 | |
| | Experiment G86S | 2.35 | 1.9604 | 78 | 0.234272 | | 0.2834 | |
| | Experiment G148R | 3.35 | 1.5425 | 78 | 0.032906 | * | 0.0988 | |
| | Experiment G148W | 2.5833 | 1.6446 | 78 | 0.120269 | | 0.2177 | |
| | Experiment T250A | 2.13 | 1.7777 | 78 | 0.234467 | | 0.2834 | |
| | Experiment T250S | 3.27 | 1.7777 | 78 | 0.069648 | † | 0.1558 | |
| | Experiment T250M | 5.07 | 1.7777 | 78 | 0.005558 | ** | 0.0282 | * |
| | Experiment H$_2$O | 4.9038 | 1.2158 | 78 | 0.000127 | *** | 0.0015 | ** |

Analysis results of electrophysiological measurements in response to ammonia/ammonium across different oocyte groups. The model included oocyte batch as a random effect to account for inter-batch variability. The intercept represents the wild-type RhBG, whereas experimental oocyte groups (mutants and water-injected control) are modelled as fixed effects. Linearity of the data was verified using the Ramsey RESET test applied to the corresponding linear model. $P$ values of estimated marginal means (EMMs) from mixed-effects models were adjusted for multiple comparisons using the false discovery rate method. Significance levels: \*\*\*$P < 0.001$, \*\*$P < 0.01$, \*$P < 0.05$, †$P = 0.05$. Abbreviations: Δ$I$, delta current; ΔpH$_s$, delta surface pH; Δ$V_m$, delta membrane potential; ΔpH$_i$, delta intracellular pH; ΔpH$_i$/Δt, slope; SE, standard error; d.f., degree of freedom; $P_r(>|t|)$: $P$ value associated with the $t$ value; Adjusted $P$, adjusted $P$ value of EMMs.

**Table A2. Fixed effects estimates from linear mixed-effects models of measurements in response to methylamine/methylammonium**

| | | Estimate | SE | d.f. | t value | $P_r(>\lvert t\rvert)$ | | Adjusted P | |
|---|---|---|---|---|---|---|---|---|---|
| $\Delta I$ | (Intercept) | −27.185 | 1.8234 | 44.3022 | −14.909 | $<2 \times 10^{-16}$ | *** | | |
| | Experiment G86C | −1.658 | 2.9008 | 98.8876 | −0.572 | $5.69 \times 10^{-1}$ | | $6.58 \times 10^{-1}$ | |
| | Experiment G86S | 6.0183 | 3.0001 | 98.7134 | 2.006 | 0.0476 | * | $8.33 \times 10^{-2}$ | |
| | Experiment G148R | 27.4428 | 2.8401 | 97.7202 | 9.663 | $6.71 \times 10^{-16}$ | *** | <0.0001 | *** |
| | Experiment G148W | 26.2774 | 3.1243 | 98.6418 | 8.411 | $3.21 \times 10^{-13}$ | *** | <0.0001 | *** |
| | Experiment T250A | 0.4442 | 3.2509 | 98.6231 | 0.137 | 0.8916 | | $8.94 \times 10^{-1}$ | |
| | Experiment T250S | 6.2417 | 3.2509 | 98.6231 | 1.92 | 0.0577 | † | $8.44 \times 10^{-2}$ | |
| | Experiment T250M | 27.5298 | 3.2825 | 96.0575 | 8.387 | $4.27 \times 10^{-13}$ | *** | <0.0001 | *** |
| | Experiment H$_2$O | 27.0009 | 2.3621 | 96.3505 | 11.431 | $<2 \times 10^{-16}$ | *** | <0.0001 | *** |
| $\Delta pH_s$ | (Intercept) | −0.07903 | 0.005914 | 10.68968 | −13.364 | $5.21 \times 10^{-8}$ | *** | | |
| | Experiment G86C | 0.021134 | 0.007448 | 93.15029 | 2.838 | 0.00558 | ** | $7.80 \times 10^{-3}$ | ** |
| | Experiment G86S | 0.033234 | 0.007448 | 93.15029 | 4.462 | $2.27 \times 10^{-5}$ | *** | <0.0001 | *** |
| | Experiment G148R | 0.058134 | 0.007448 | 93.15029 | 7.806 | $8.54 \times 10^{-12}$ | *** | <0.0001 | *** |
| | Experiment G148W | 0.061834 | 0.007448 | 93.15029 | 8.302 | $7.81 \times 10^{-13}$ | *** | <0.0001 | *** |
| | Experiment T250A | 0.006986 | 0.008895 | 94.52233 | 0.785 | 0.43417 | | $4.39 \times 10^{-1}$ | |
| | Experiment T250S | 0.015272 | 0.008895 | 94.52233 | 1.717 | 0.08926 | † | $1.06 \times 10^{-1}$ | |
| | Experiment T250M | 0.065158 | 0.009395 | 94.19437 | 6.935 | $5.05 \times 10^{-10}$ | *** | <0.0001 | *** |
| | Experiment H$_2$O | 0.053425 | 0.006283 | 93.94091 | 8.503 | $2.80 \times 10^{-13}$ | *** | <0.0001 | *** |
| $\Delta V_m$ | (Intercept) | 25.258 | 1.659 | 12.105 | 15.223 | $2.93 \times 10^{-9}$ | *** | | |
| | Experiment G86C | −21.308 | 2.089 | 78.942 | −10.202 | $4.53 \times 10^{-16}$ | *** | <0.0001 | *** |
| | Experiment G86S | −11.412 | 2.42 | 76.211 | −4.715 | $1.07 \times 10^{-5}$ | *** | <0.0001 | *** |
| | Experiment G148R | −22.756 | 2.327 | 77.763 | −9.777 | $3.49 \times 10^{-15}$ | *** | <0.0001 | *** |
| | Experiment G148W | −21.554 | 2.461 | 76.437 | −8.757 | $3.73 \times 10^{-13}$ | *** | <0.0001 | *** |
| | Experiment T250A | −6.341 | 2.612 | 75.701 | −2.427 | 0.0176 | * | $1.87 \times 10^{-2}$ | * |
| | Experiment T250S | −13.681 | 2.612 | 75.701 | −5.237 | $1.42 \times 10^{-6}$ | *** | <0.0001 | *** |
| | Experiment T250M | −22.961 | 2.612 | 75.701 | −8.789 | $3.48 \times 10^{-13}$ | *** | <0.0001 | *** |
| | Experiment H$_2$O | −25.324 | 1.788 | 76.421 | −14.164 | $<2 \times 10^{-16}$ | *** | <0.0001 | *** |
| $\Delta pH_i$ | (Intercept) | 0.146908 | 0.006986 | 39.48307 | 21.028 | $<2 \times 10^{-16}$ | *** | | |
| | Experiment G86C | −0.06768 | 0.014171 | 72.79256 | −4.776 | $9.02 \times 10^{-6}$ | *** | <0.0001 | *** |
| | Experiment G86S | −0.08258 | 0.01735 | 78.86507 | −4.76 | $8.64 \times 10^{-6}$ | *** | <0.0001 | *** |
| | Experiment G148R | −0.12201 | 0.016289 | 77.31366 | −7.49 | $9.50 \times 10^{-11}$ | *** | <0.0001 | *** |
| | Experiment G148W | −0.11241 | 0.017387 | 75.28649 | −6.465 | $9.05 \times 10^{-9}$ | *** | <0.0001 | *** |
| | Experiment T250A | −0.02708 | 0.018764 | 78.25898 | −1.443 | 0.153004 | | $1.66 \times 10^{-1}$ | |
| | Experiment T250S | −0.07508 | 0.018764 | 78.25898 | −4.001 | 0.000142 | *** | $2.00 \times 10^{-4}$ | *** |
| | Experiment T250M | −0.12608 | 0.018764 | 78.25898 | −6.719 | $2.65 \times 10^{-9}$ | *** | <0.0001 | *** |
| | Experiment H$_2$O | −0.11459 | 0.012826 | 78.95226 | −8.934 | $1.32 \times 10^{-13}$ | *** | <0.0001 | *** |

(Continued)

**Table A2. (Continued)**

| | | Estimate | SE | d.f. | t value | $P_r(>|t|)$ | | Adjusted P | |
|---|---|---|---|---|---|---|---|---|---|
| ΔpH$_i$/Δt | (Intercept) | 10.7975 | 0.7075 | 36.7467 | 15.261 | $<2 \times 10^{-16}$ | *** | | |
| | Experiment G86C | −5.3036 | 1.4152 | 72.9261 | −3.748 | 0.000355 | *** | $9.00 \times 10^{-4}$ | *** |
| | Experiment G86S | −5.6857 | 1.7262 | 79.0355 | −3.294 | 0.001481 | ** | $2.20 \times 10^{-3}$ | ** |
| | Experiment G148R | −9.3318 | 1.6228 | 77.4174 | −5.75 | $1.69 \times 10^{-7}$ | *** | <0.0001 | *** |
| | Experiment G148W | −9.2182 | 1.7339 | 76.16 | −5.317 | $1.02 \times 10^{-6}$ | *** | <0.0001 | **** |
| | Experiment T250A | −1.6753 | 1.8678 | 78.6206 | −0.897 | 0.372486 | | $3.87 \times 10^{-1}$ | |
| | Experiment T250S | −6.4553 | 1.8678 | 78.6206 | −3.456 | 0.000887 | *** | $1.60 \times 10^{-3}$ | ** |
| | Experiment T250M | −9.2353 | 1.8678 | 78.6206 | −4.945 | $4.23 \times 10^{-6}$ | *** | <0.0001 | *** |
| | Experiment H$_2$O | −9.2508 | 1.276 | 79.1299 | −7.25 | $2.46 \times 10^{-10}$ | *** | <0.0001 | *** |

Analysis results of electrophysiological measurements in response to methylamine/methylammonium across different oocyte groups. The model included oocyte batch as a random effect to account for inter-batch variability. The intercept represents the wild-type RhBG, whereas experimental oocyte groups (mutants and water-injected control) are modelled as fixed effects. Linearity of the data was verified using the Ramsey RESET test applied to the corresponding linear model. *P* values of estimated marginal means (EMMs) from mixed-effects models were adjusted for multiple comparisons using the false discovery rate method. Significance levels: ***$P < 0.001$, **$P < 0.01$, *$P < 0.05$, †$P = 0.05$. Abbreviations: Δ*I*, delta current; ΔpH$_s$, delta surface pH; Δ*V*m, delta membrane potential; ΔpH$_i$, delta intracellular pH; ΔpH$_i$/Δ*t*, slope; SE, standard error; d.f., degree of freedom; $P_r(>|t|)$: *P* value associated with the *t* value; Adjusted *P*, adjusted *P* value of EMMs.

**Table A3. Univariate statistical analysis**

| | NH$_4^+$ | | | NH$_3$ | MA$^+$ | | MA | |
| --- | --- | --- | --- | --- | --- | --- | --- | --- |
| | ΔI | Slope | ΔV$_m$ | ΔpH$_s$ | ΔI | ΔV$_m$ | ΔpH$_s$ | Slope |
| G86S | −30.8 ± 10.4 | −8.08 ± 4.81 | 13.2 ± 6.01 | −0.128 ± 0.0388 | −21.2 ± 7.6 | 10.8 ± 6.81 | −0.0459 ± 0.0148 | 5.81 ± 4.81 |
| P value | 0.851 | 0.429 | 0.08137 | 0.000260 | 0.3114 | 0.00280 | 0.000490 | 0.00420 |
| G86C | −17.8 ± 10.4 | −8.27 ± 3.72 | 2.78 ± 2.73 | −0.147 ± 0.0683 | −28.6 ± 16.6 | 2.60 ± 0.990 | −0.058 ± 0.0285 | 5.17 ± 2.90 |
| P value | 0.000470 | 0.684 | $1.90 \times 10^{-10}$ | 0.0149 | 0.9985 | $5.40 \times 10^{-11}$ | 0.0552 | 0.00150 |
| G148R | −18.7 ± 12.2 | −6 ± 2.61 | 12.7 ± 6.82 | −0.107 ± 0.0230 | 0.8 ± 1.6 | 0.743 ± 0.493 | −0.021 ± 0.0081 | 1.41 ± 0.740 |
| P value | 0.000380 | 0.0120 | $2.70 \times 10^{-4}$ | $1.70 \times 10^{-6}$ | $1.70 \times 10^{-15}$ | $1.10 \times 10^{-8}$ | $1.00 \times 10^{-9}$ | $3.70 \times 10^{-7}$ |
| G148W | −17.9 ± 11.6 | −6.77 ± 3.43 | 15.9 ± 4.77 | −0.958 ± 0.0317 | −0.3 ± 1.5 | 2.13 ± 1.88 | −0.0173 ± 0.011 | 1.50 ± 0.897 |
| P value | 0.000860 | 0.0445 | $6.74 \times 10^{-3}$ | $8.40 \times 10^{-8}$ | $5.60 \times 10^{-13}$ | $6.20 \times 10^{-8}$ | $7.20 \times 10^{-11}$ | $8.40 \times 10^{-7}$ |
| T250A | −33.3 ± 14.2 | −7.22 ± 4.72 | 21 ± 4.54 | −0.141 ± 0.0250 | −27.2 ± 8.5 | 17.7 ± 7.51 | −0.0666 ± 0.0148 | 9.10 ± 4.12 |
| P value | 0.998 | 0.654 | 0.500 | 0.02056 | 1 | 0.745 | 0.65969 | 0.840 |
| T250S | −29.3 ± 10.4 | −6.08 ± 2.32 | 18.7 ± 6.25 | −0.127 ± 0.0540 | −21.4 ± 6.4 | 10.4 ± 9.01 | −0.0583 ± 0.0338 | 4.32 ± 2.57 |
| P value | 0.700 | 0.297 | 0.241 | 0.00392 | 0.427 | 0.0721 | 0.1423 | 0.195 |
| T250M | −19.5 ± 8.04 | −4.28 ± 2.01 | 14.3 ± 8.24 | −0.0998 ± 0.0528 | −0.3 ± 0.9 | 1.10 ± 1.03 | −0.00983 ± 0.0127 | 1.54 ± 0.966 |
| P value | 0.00514 | 0.0488 | 0.0125 | $2.80 \times 10^{-5}$ | $1.50 \times 10^{-12}$ | $1.40 \times 10^{-5}$ | $3.00 \times 10^{-9}$ | 0.000280 |
| H$_2$O-injected | −15.6 ± 7.70 | −4.68 ± 2.26 | 14.6 ± 5.54 | −0.105 ± 0.0364 | 0.1 ± 0.50 | 0.338 ± 0.173 | −0.0238 ± 0.018 | 1.70 ± 1.10 |
| P value | $4.00 \times 10^{-7}$ | 0.0111 | 0.000490 | $2.90 \times 10^{-9}$ | $<2 \times 10^{-16}$ | $7.60 \times 10^{-11}$ | $1.50 \times 10^{-12}$ | $1.80 \times 10^{-6}$ |

All variables were compared using one-way ANOVA followed by Dunnett's test for normally distributed data or Kruskal–Wallis $H$ test followed by Dunn's test (non-parametric ANOVA *post hoc* test). In *post hoc* tests, the RhBG-expressing group was set as the reference. Abbreviations: ΔI, delta current; ΔpH$_s$, delta surface pH; ΔV$_m$, delta membrane potential; ΔpH$_i$, delta intracellular pH; ΔpH$_i$/Δt, slope; P value, P value; P values adjusted with built-in methods in *post hoc* tests.

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

## Additional information

### Data availability statement

The data sets generated and/or analysed during the current study are available from the corresponding author on reasonable request. This includes all data related to the oocyte experiments and analytical code used for statistical analysis in RStudio. The phenotype and genotype data of the CRIC and ARIC data used in this manuscript are from NIDDK (National Institute of Diabetes and Digestive and Kidney Diseases) Central Repository and the BioLINCC (Biologic Specimen and Data Repository Information Coordinating Centre) and can be accessed through dbGaP (Database of Genotypes and Phenotypes). The dbGaP accession numbers are phs000280 for ARIC study and phs000524 for CRIC study.

### Competing interests

The authors declare that they have no competing interests.

## Author contributions

N.L.N. conceived and designed research. H.Z. performed experiments. H.Z. and N.L.N. analysed and interpreted data. H.Z., S.A-N., L.L.H. and N.L.N. discussed results of the experiments. H.Z. prepared figures and drafted the manuscript. S.A-N., L.L.H. and N.L.N. edited and revised the manuscript. H.Z., S.A-N., L.L.H. and N.L.N. approved the final version of the manuscript submitted for publication.

## Funding

This work was supported in part by U54 GM104940 from the National Institute of General Medical Sciences of the National Institutes of Health, which funds the Louisiana Clinical and Translational Science Centre. Other sources of funding include a VA Merit grant BX-001513 (S.A-N.), Lavin-Bernick grant (S.A-N.), a Paul Teschan Research grant (N.L.N.) and a Tulane Bridge Fund Grant (N.L.N.).

## Acknowledgements

We thank Dr Tanika Kelly and her team for initial screening of variants from the CRIC and ARIC databases. We acknowledge Dr Xuenan Mi and Dr Yang Pan for accessing the dbGaP. We thank Mr M. Toriqul Islam for technical assistance.

## Keywords

ammonia, ammonium, chronic kidney disease, CKD, electrophysiology, Rh protein, RhBG

## Supporting information

Additional supporting information can be found online in the Supporting Information section at the end of the HTML view of the article. Supporting information files available:

**Peer Review History**

