## [Peer Review History · The Journal of Physiology]

Mutations in ammonia transporter RhBG that impair $\text{NH}_3/\text{NH}_4^+$ transport in patients with chronic kidney disease

Nazih L. Nakhoul, He Zhou, Abdounour-Nakhoul Solange, and L. Lee Hamm
DOI: 10.1113/JP288958

Corresponding author(s): Nazih Nakhoul (nakhoul@tulane.edu)

The following individual(s) involved in review of this submission have agreed to reveal their identity: Sean Williams (Referee #3)

Review Timeline:

Submission Date:	27-Mar-2025
Editorial Decision:	02-Jun-2025
Revision Received:	12-Aug-2025
Editorial Decision:	17-Sep-2025
Revision Received:	25-Sep-2025
Editorial Decision:	29-Oct-2025
Revision Received:	14-Nov-2025
Accepted:	11-Dec-2025

Senior Editor: *Peying Fong*

Reviewing Editor: *Matthew Bailey*

Transaction Report:

Dear Dr Nakhoul,

Re: JP-RP-2025-288958 "**Mutations in ammonia transporter RhBG that impair $\text{NH}_3/\text{NH}_4^+$ transport in patients with chronic kidney disease**" by Nazih L. Nakhoul, He Zhou, Abdulnour-Nakhoul Solange, and L. Lee Hamm

Thank you for submitting your manuscript to The Journal of Physiology. It has been assessed by a Reviewing Editor and by 2 expert referees and we are pleased to tell you that it is potentially acceptable for publication following satisfactory major revision.

LANGUAGE EDITING AND SUPPORT FOR PUBLICATION: If you would like help with English language editing, or other article preparation support, Wiley Editing Services offers expert help, including English Language Editing, as well as translation, manuscript formatting, and figure formatting at www.wileyauthors.com/eoo/preparation. You can also find resources for Preparing Your Article for general guidance about writing and preparing your manuscript at www.wileyauthors.com/eoo/prepresources.

REVISION CHECKLIST:

Please upload two versions of your manuscript text: one with all relevant changes highlighted and one clean version with no

changes tracked. The manuscript file should include all tables and figure legends, but each figure/graph should be uploaded as separate, high-resolution files.

We look forward to receiving your revised submission.

Yours sincerely,

Peying Fong
Senior Editor
The Journal of Physiology

REQUIRED ITEMS

- Include a Key Points list in the article itself, before the Abstract.
- Author photo and profile. First or joint first authors are asked to provide a short biography (no more than 100 words for one author or 150 words in total for joint first authors) and a portrait photograph. These should be uploaded and clearly labelled together in a Word document with the revised version of the manuscript. See Information for Authors for further details.
- You must start the Methods section with a paragraph headed Ethical approval (https://jp.msubmit.net/cgi-bin/main.plex?form_type=display_requirements#methods).

Research must comply with The Journal's policies regarding animal experiments (<https://physoc.onlinelibrary.wiley.com/hub/animal-experiments>) and adherence to these policies must be stated in the manuscript.

Authors should confirm in their Methods section that their experiments were carried out according to the guidelines laid down by their institution's animal welfare committee, including an ethics approval reference number. The Methods section must contain a statement about access to food, water and housing, details of the anaesthetic regime: anaesthetic used, dose and route of administration, and method of killing the experimental animals.

- Please upload separate high-quality figure files via the submission form.
- Please ensure that any tables are editable and in Word format, and wherever possible, embedded in the article file itself.
- Please ensure that the Article File you upload is a Word file.
- Please include an Abstract Figure file, as well as the Figure Legend text within the main article file. The Abstract Figure is a piece of artwork designed to give readers an immediate understanding of the research and should summarise the main conclusions. If possible, the image should be easily 'readable' from left to right or top to bottom. It should show the physiological relevance of the manuscript so readers can assess the importance and content of its findings. Abstract Figures should not merely recapitulate other figures in the manuscript. Please try to keep the diagram as simple as possible and without superfluous information that may distract from the main conclusion(s). Abstract Figures must be provided by authors no later than the revised manuscript stage and should be uploaded as a separate file during online submission labelled as File Type 'Abstract Figure'. Please also ensure that you include the figure legend in the main article file. All Abstract Figures

should be created using BioRender. Authors should use The Journal's premium BioRender account to export high-resolution images. Details on how to use and access the premium account are included as part of this email.

EDITOR COMMENTS

Reviewing Editor:

The manuscript has been assessed by two expert Reviewers. Both find the study to be novel and important but both identify limitations in the experimental approaches and in the Discussion. All of these points can be addressed with an extensive textual revision.

Reviewer 1 identifies two areas requiring expanded discussion and also highlights revisions needed to improve clarity of "n" and of the presentation of the immuno-fluorescent images. This reviewer also makes helpful suggestions to improve the readability of the paper.

Reviewer 2 highlights limitations in the use of *Xenopus* oocytes and of methylammonium as a surrogate for studying ammonium transport. These need to be considered in an expanded Discussion segment. The Reviewer also identifies several areas in which the Discussion should be revised to better address the results of the current study with previous research.

The Reviewing Editor requests revision of the Method section to clarify the post-operative care of frogs following oocyte removal.

Please also see 'Required Items' above.

Senior Editor:

Review of your manuscript, "Mutations in ammonia transporter RhBG that impair NH₃/NH₄⁺ transport in patients with chronic kidney disease", is now complete. It has been assessed by two Expert Referees, who agree on its potential impactfulness, particularly with regard to its contributions toward the collective understanding of chronic kidney disease. However, they also identify several points of concern that require additional attention.

I anticipate that the Reviewing Editor's succinct summary of both Referees' detailed comments will guide you toward understanding their comments and ultimately how you approach revision.

In preparing your revised manuscript, please consider and address all points raised, including to the request raised by the Reviewing Editor regarding *Xenopus* post-operative care.

Please also ensure that your revised manuscript complies fully with current guidelines pertaining to revised submissions that are provided in the published Information for Authors; https://jp.msubmit.net/cgi-bin/main.plex?form_type=display_requirements#Revised%20submissions.

We look forward to receiving your revised manuscript and thank you for contributing to The Journal of Physiology.

REFEREE COMMENTS

Referee #1:

The manuscript, "Mutations in ammonia transporter Rhbg that impair NH₃/NH₄⁺ transport in patients with chronic kidney disease" by Zhou, Abdulnour-Nakhoul, Hamm, and Nakhoul, reports identification of several nonsynonymous single nucleotide polymorphisms in the Rhbg gene that correlate with the presence of chronic kidney disease in two large human datasets. The authors then show that these polymorphisms are associated either with altered transport function or altered membrane expression. These findings are novel and important and are consistent with the paradigm that altered renal ammonia excretion can contribute to the progression of chronic kidney disease.

Major comments

1. Net transport by a protein is determined by the separate components of membrane expression and per-molecule transport activity. Can the discussion more clearly discuss the differential effects of changes in per-molecular transport activity from those of membrane expression. While the net result is the same, understanding the specific mechanism by which a mutation leads to altered transport is important.
2. Does the 'N' that is reported refer to the number of batches injected or to the number of oocytes injected? In other words, if there were three batches of injections with 4, 5, and 6 oocytes injected, respectively, will the N be reported as three or 15? Given the known intra-batch variations in oocyte transport studies, it seems more appropriate to average all results in a single batch of injections and use that as a count of 1 for statistical analysis. Can the authors provide more information regarding how this was handled?
3. Functional Rhbg probably exists as a trimer. These oocyte expression studies almost certainly result in a homotrimeric protein in which each of the monomers results from the injected RNA. In people who are heterozygous for one of these polymorphisms, would it be likely that the functional protein would be a heterotrimer comprised of both wild type and polymorphism-related monomers? Can the authors speculate as to how this might impact the interpretation of the results?
4. Were the immunofluorescent micrographs taken in figure 11 obtained with identical microscopy settings? If they were not, can this be specifically mentioned so that the reader does not infer quantitatively similar membrane expression from similar immunofluorescence intensity.
5. This manuscript focuses on rare nonsynonymous single nucleotide polymorphisms in Rhbg that correlated with the presence of chronic kidney disease. Although not explicitly stated, this reader would assume that this means that nonsynonymous single nucleotide polymorphisms with a MAF greater than 0.01 did not correlate. Is this correct?

Minor comments

1. On page three of the introduction the authors seem to suggest that increased ammoniogenesis without an increase in urinary ammonia excretion would increase net bicarbonate production. Ammonia that is produced in the kidney and is not excreted in the urine returns to the systemic circulation through the renal veins, and then is metabolized, predominantly by liver and skeletal muscle, through a process that utilizes equimolar bicarbonate. This suggests that it is unlikely that ammoniogenesis in the absence of ammonia excretion improves systemic acid-base balance. Can the authors comment on this?
2. In Table 1, it is not clear to this reader how the frequency of each SNV was calculated. Can this be explicitly detailed?
3. I presume that successful mutagenesis was confirmed by sequencing, but I don't find this stated in the manuscript. Can the authors provide the details as to how the correct mutagenesis was confirmed?
4. There's a very substantial amount of data presented in this manuscript, and it is often difficult to keep all the various findings in mind to interpret them accurately. Can the authors consider summarizing each set of data as to their effect on NH_3 , NH_4^+ , MA^- and MA^- transport? Also, would it be helpful to move figure table 4 earlier in the manuscript, and refer to it as the data is developed.
5. In figure 11, the water injected oocyte tracings are not shown. Is there a reason why not?

Referee #2:

Reviewer Comments - JP-RP-2025-288958

Title: Mutations in ammonia transporter RhBG that impair $\text{NH}_3/\text{NH}_4^+$ transport in patients with chronic kidney disease

Summary:

This manuscript investigates rare non-synonymous variants in the RhBG ammonia transporter and their potential impact on $\text{NH}_3/\text{NH}_4^+$ transport capacity, using *Xenopus laevis* oocytes as an expression system. Several variants are shown to impair or abolish ammonium transport based on electrophysiological and pH-based readouts.

Although the study tackles a clinically relevant issue, how impaired ammonia transport may contribute to CKD progression, methodological flaws, overinterpretation of the results, and a lack of engagement with existing literature weaken the

physiological validity of the conclusions.

1. Exclusive Reliance on *Xenopus* Oocytes as a Model System

The exclusive use of *Xenopus* oocytes for the functional characterization of ammonia transporters constitutes a major methodological limitation. This model system has been repeatedly questioned in the literature due to its potential for confounding results. Specifically, previous studies have demonstrated that currents elicited by ammonium pulses may originate from endogenous NH_4^+ -permeable channels, rather than from the heterologously expressed transporter itself. This significantly complicates the interpretation of RhBG-mediated transport activity.

Furthermore, *Xenopus* oocytes exhibit a pronounced ability to buffer ammonium-induced changes in intracellular pH (pH_i), which is likely an evolutionary adaptation to the elevated levels of ammonium present in their natural aquatic environments. This intrinsic buffering capacity can significantly distort the cellular responses to ammonium exposure, resulting in inconsistent and potentially misleading physiological data. These limitations have been clearly documented by Musa-Aziz et al. (2009) and further discussed in the context of Rh protein function by Williamson et al. (2004). The authors should explicitly cite these studies and critically discuss their findings in relation to the present data, as they are highly relevant to the interpretation of results obtained using the *Xenopus* oocyte system. Given these well-established caveats, it is critical that the authors acknowledge these limitations and discuss their implications when interpreting the data.

Numerous studies addressing Rh-mediated transport have included complementary validation using alternative systems, such as yeast complementation assays or expression in mammalian cell lines, to overcome the inherent limitations of the *Xenopus* model. The absence of such validation in the current manuscript weakens the physiological relevance of the conclusions and should be acknowledged.

2. Use of Methylammonium (MeA) Transport Assay

While the MeA is commonly used to assess Amt/Rh transporter activity in various systems including oocytes, its utility as a mechanistic proxy for ammonium transport is increasingly contested. A growing body of evidence has demonstrated that MeA and ammonium are not always handled similarly by these transporters, especially in variant proteins (Wang et al., 2013; Hall et al., 2011; Javelle et al., 2005).

This discrepancy is particularly evident in *Xenopus* oocytes, where studies, including Ludwig (2004), have shown that Rh proteins handle MeA and ammonium differently. Importantly, the authors' own data support this distinction: they report that an ammonium pulse induces intracellular acidification, while a MeA pulse results in alkalinization. Such inconsistency is not discussed in the manuscript.

Given the extent of the evidence challenging the validity of MeA as a reliable surrogate for ammonium, the authors must exercise caution when interpreting results based on the use of MeA. At a minimum, this methodological issue should be explicitly acknowledged and critically discussed in the context of the broader literature.

3. Misrepresentation of Prior Literature

Throughout the Results and Discussion sections, the authors repeatedly cite Ludwig et al. (2004), asserting that their findings are in agreement with that study. However, this is a misrepresentation. As stated earlier, Ludwig explicitly cautions that observed currents in *Xenopus* oocytes may result from endogenous NH_4^+ -permeable channels rather than from RhBG itself. More importantly, Ludwig concludes that RhBG functions as an electroneutral NH_4^+/H^+ antiporter, which is fundamentally at odds with the mechanistic interpretation presented by the authors.

This electroneutral mechanism has since been corroborated by several independent studies, including Mak et al. (2006) and Greyer et al. (2013). Additional evidence from studies using HEK-293 and MDCK cell systems expressing RhBG further supports this conclusion, as these systems also exhibit electroneutral transport activity in response to both ammonium and MeA.

The authors should revise their interpretation to accurately reflect the current consensus in the field and avoid misrepresenting previous findings. At a minimum, a balanced discussion of alternative mechanistic models is essential for an accurate and fair representation of RhBG function.

In summary, the manuscript suffers from significant methodological and interpretative limitations that undermine the reliability of its conclusions. A more rigorous discussion of the model system's constraints and a critical engagement with the relevant literature are essential to support the authors' claims.

END OF COMMENTS

Reviewer Comments – JP-RP-2025-288958

Title: *Mutations in ammonia transporter RhBG that impair NH₃/NH₄⁺ transport in patients with chronic kidney disease*

Summary:

This manuscript investigates rare non-synonymous variants in the RhBG ammonia transporter and their potential impact on NH₃/NH₄⁺ transport capacity, using *Xenopus laevis* oocytes as an expression system. Several variants are shown to impair or abolish ammonium transport based on electrophysiological and pH-based readouts.

Although the study tackles a clinically relevant issue, how impaired ammonia transport may contribute to CKD progression, methodological flaws, overinterpretation of the results, and a lack of engagement with existing literature weaken the physiological validity of the conclusions.

1. Exclusive Reliance on *Xenopus* Oocytes as a Model System

The exclusive use of *Xenopus* oocytes for the functional characterization of ammonia transporters constitutes a major methodological limitation. This model system has been repeatedly questioned in the literature due to its potential for confounding results. Specifically, previous studies have demonstrated that currents elicited by ammonium pulses may originate from endogenous NH₄⁺-permeable channels, rather than from the heterologously expressed transporter itself. This significantly complicates the interpretation of RhBG-mediated transport activity.

Furthermore, *Xenopus* oocytes exhibit a pronounced ability to buffer ammonium-induced changes in intracellular pH (pH_i), which is likely an evolutionary adaptation to the elevated levels of ammonium present in their natural aquatic environments. This intrinsic buffering capacity can significantly distort the cellular responses to ammonium exposure, resulting in inconsistent and potentially misleading physiological data. These limitations have been clearly documented by Musa-Aziz et al. (2009) and further discussed in the context of Rh protein function by Williamson et al. (2004). The authors should explicitly cite these studies and critically discuss their findings in relation to the present data, as they are highly relevant to the interpretation of results obtained using the *Xenopus* oocyte system. Given these well-established caveats, it is critical that the authors acknowledge these limitations and discuss their implications when interpreting the data.

Numerous studies addressing Rh-mediated transport have included complementary validation using alternative systems, such as yeast complementation assays or expression in mammalian cell lines, to overcome the inherent limitations of the *Xenopus* model. The absence of such validation in the current manuscript weakens the physiological relevance of the conclusions and should be acknowledged.

2. Use of Methylammonium (MeA) Transport Assay

While the MeA is commonly used to assess Amt/Rh transporter activity in various systems including oocytes, its utility as a mechanistic proxy for ammonium transport is increasingly contested. A growing body of evidence has demonstrated that MeA and ammonium are not always handled similarly by these transporters, especially in variant proteins (Wang et al., 2013; Hall et al., 2011; Javelle et al., 2005).

This discrepancy is particularly evident in *Xenopus* oocytes, where studies, including Ludewig (2004), have shown that Rh proteins handle MeA and ammonium differently. Importantly, the authors' own data support this distinction: they report that an ammonium pulse induces intracellular acidification, while a MeA pulse results in alkalinization. Such inconsistency is not discussed in the manuscript.

Given the extent of the evidence challenging the validity of MeA as a reliable surrogate for ammonium, the authors must exercise caution when interpreting results based on the use of MeA. At a minimum, this methodological issue should be explicitly acknowledged and critically discussed in the context of the broader literature.

3. Misrepresentation of Prior Literature

Throughout the Results and Discussion sections, the authors repeatedly cite Ludewig et al. (2004), asserting that their findings are in agreement with that study. However, this is a misrepresentation. As stated earlier, Ludewig explicitly cautions that observed currents in *Xenopus* oocytes may result from endogenous NH_4^+ -permeable channels rather than from RhBG itself. More importantly, Ludewig concludes that RhBG functions as an electroneutral NH_4^+/H^+ antiporter, which is fundamentally at odds with the mechanistic interpretation presented by the authors.

This electroneutral mechanism has since been corroborated by several independent studies, including Mak et al. (2006) and Greyer et al. (2013). Additional evidence from studies using HEK-293 and MDCK cell systems expressing RhBG further supports this conclusion, as these systems also exhibit electroneutral transport activity in response to both ammonium and MeA.

The authors should revise their interpretation to accurately reflect the current consensus in the field and avoid misrepresenting previous findings. At a minimum, a balanced discussion of alternative mechanistic models is essential for an accurate and fair representation of RhBG function.

In summary, the manuscript suffers from significant methodological and interpretative limitations that undermine the reliability of its conclusions. A more rigorous discussion of the model system's constraints and a critical engagement with the relevant literature are essential to support the authors' claims.

EDITOR COMMENTS

Reviewing Editor:

The manuscript has been assessed by two expert Reviewers. Both find the study to be novel and important, but both identify limitations in the experimental approaches and in the Discussion. All these points can be addressed with an extensive textual revision.

Reviewer 1 identifies two areas requiring expanded discussion and also highlights revisions needed to improve clarity of "n" and of the presentation of the immunofluorescent images. This reviewer also makes helpful suggestions to improve the readability of the paper.

Reviewer 2 highlights limitations in the use of *Xenopus* oocytes and of methylammonium as a surrogate for studying ammonium transport. These need to be considered in an expanded Discussion segment. The Reviewer also identifies several areas in which the Discussion should be revised to better address the results of the current study with previous research.

The Reviewing Editor requests revision of the Method section to clarify the post-operative care of frogs following oocyte removal.

We appreciate Reviewing Editor's suggestion. We have revised the Method section- Ethical approval and animal procedures (lines 98-100).

Please also see 'Required Items' above.

As requested, the required items are included in this manuscript or prepared for separate submission.

Senior Editor:

Review of your manuscript, "Mutations in ammonia transporter RhBG that impair NH₃/NH₄⁺ transport in patients with chronic kidney disease", is now complete. It has been assessed by two Expert Referees, who agree on its potential impactfulness, particularly with regard to its contributions toward the collective understanding of chronic kidney disease. However, they also identify several points of concern that require additional attention.

I anticipate that the Reviewing Editor's succinct summary of both Referees' detailed comments will guide you toward understanding their comments and ultimately how you approach revision.

In preparing your revised manuscript, please consider and address all points raised, including to the request raised by the Reviewing Editor regarding *Xenopus* post-operative

care.

Please also ensure that your revised manuscript complies fully with current guidelines pertaining to revised submissions that are provided in the published Information for Authors; https://jp.msubmit.net/cgi-bin/main.plex?form_type=display_requirements#Revised%20submissions.

We look forward to receiving your revised manuscript and thank you for contributing to The Journal of Physiology.

We thank the Senior Editor for the summary and guidance. We have addressed all points raised by the Reviewing Editor and the two referees.

REFeree COMMENTS

Referee #1:

The manuscript, "Mutations in ammonia transporter Rhbg that impair NH₃/NH₄⁺ transport in patients with chronic kidney disease" by Zhou, Abdalnour-Nakhoul, Hamm, and Nakhoul, reports identification of several nonsynonymous single nucleotide polymorphisms in the Rhbg gene that correlate with the presence of chronic kidney disease in two large human datasets. The authors then show that these polymorphisms are associated either with altered transport function or altered membrane expression. These findings are novel and important and are consistent with the paradigm that altered renal ammonia excretion can contribute to the progression of chronic kidney disease.

Major comments

1. Net transport by a protein is determined by the separate components of membrane expression and per-molecule transport activity. Can the discussion more clearly discuss the differential effects of changes in per-molecular transport activity from those of membrane expression. While the net result is the same, understanding the specific mechanism by which a mutation leads to altered transport is important.

We thank the reviewer for this excellent suggestion to more clearly distinguish between membrane expression defects and per-molecule transport activity changes. We have commented on this mechanistic question throughout our Discussion by combining our electrophysiological transport data with immunohistochemical membrane expression

analysis for each individual mutant. Following the reviewer's suggestion, we have reorganized this mechanistic analysis into a dedicated subsection titled "Mechanisms of transport impairment: membrane expression vs. per-molecule activity" in our revised Discussion (lines 603-629). We appreciate the reviewer's guidance in improving the clarity and impact of our work and hope this revision will make the mechanistic insights more accessible to readers.

2. Does the 'N' that is reported refer to the number of batches injected or to the number of oocytes injected? In other words, if there were three batches of injections with 4, 5, and 6 oocytes injected, respectively, will the N be reported as three or 15? Given the known intra-batch variations in oocyte transport studies, it seems more appropriate to average all results in a single batch of injections and use that as a count of 1 for statistical analysis. Can the authors provide more information regarding how this was handled?

We thank the reviewer for this important methodological question regarding our statistical analysis approach. We appreciate the opportunity to clarify our experimental design and statistical methodology. In our study, the reported 'N' values represent the total number of individual oocytes from all injection batches combined. For example, if we had three batches with 4, 5, and 6 oocytes respectively, we report N=15 for statistical analysis. This approach follows established conventions in transport physiology studies using *Xenopus* oocytes (as referenced in our cited papers: Nakhoul et al., 2005, 2006; Abdulnour-Nakhoul, 2016, and other studies using similar method listed below:(Lee & Boron, 2018; Yarcusko et al., 2024)) where the primary research question focuses on functional differences between experimental groups (e.g., wild-type vs. mutant proteins) rather than batch-to-batch variability. Usually, when comparing one or two mutants with the control, the conventional practice is to perform all measurements on one batch of oocytes (Lee & Boron, 2018). As the number of mutants increases, due to the resource-intensive, low-throughput nature of the electrophysiological measurements, it is not feasible to collect the target sample size from one batch of oocytes. We implemented a rigorous batch-matching protocol to enable the feasibility of obtaining optimal sample size based on power calculation, and to control for inter-batch variability where all experimental and control groups were represented in each batch to ensure balanced representation and we repeated the experiments for each mutation in 3 batches (i.e., 3 different frogs). Mutants were only compared with the control groups from the same batch. For example, we obtained data from 5 control oocytes and 5 mutant oocytes from the same frog and repeated the measurements in two more frogs. The sample sizes of control and mutant group included in data analysis were 15 each. This way, the biases due to uneven batch size can be reduced. We have revised the Method section – “Oocytes Injection”, “Electrophysiological Measurements”, “Statistics” (lines 178, 180-181, 184, 191-192), as well as the figure legends for bar plots (Figures 2-10) in the manuscript to clarify this.

Additional references (not cited in the manuscript):

Lee SK & Boron WF. (2018). Exploring the autoinhibitory domain of the electrogenic Na⁽⁺⁾/HCO₃⁽⁻⁾ transporter NBCe1-B, from residues 28 to 62. *J Physiol* 596, 3637-3653.

Yarcusko RS, Song MH, Neuger GC, Romero MF, Piermarini PM & Gillen CM. (2024). Function and regulation of the insect NaCCC2 sodium transport proteins. *Comp Biochem Physiol A Mol Integr Physiol* 296, 111685.

3. Functional Rhbg probably exists as a trimer. These oocyte expression studies almost certainly result in a homotrimeric protein in which each of the monomers results from the injected RNA. In people who are heterozygous for one of these polymorphisms, would it be likely that the functional protein would be a heterotrimer comprised of both wild type and polymorphism-related monomers? Can the authors speculate as to how this might impact the interpretation of the results?

This is an excellent and clinically relevant question that highlights an important limitation of our current study. To address this, we have added a subsection titled " Heterozygous complexity: from homotrimeric models to heterozygous patient" in our revised Discussion (line 630-679).

Besides the biological complexities described in Discussion, addressing this question in our current study faces several practical constraints: 1) Regarding the clinical implications: we did not have access to individual-level patient data to compare impacts on phenotypes between heterozygous and homozygous carriers. However, this represents a critical knowledge gap in clinical genetic nephrology, as it directly relates to potential dominant-negative effects and gene dosage sensitivity; 2) Regarding experimental approach: while the *Xenopus* oocyte expression system is technically well-suited for testing dominant-negative effects and genetic dosage sensitivity by co-injecting wild-type and mutant RhBG mRNAs in defined ratios, such functional studies would be less meaningful without prior genetic-level evidence demonstrating differential clinical outcomes between heterozygous and homozygous carriers. The experimental design should be informed by observed genotype-phenotype correlations to ensure clinical relevance. Therefore, the most appropriate approach would be to first establish genetic associations comparing heterozygous versus homozygous carriers in patient cohorts, followed by targeted functional studies based on these clinical observations.

While this represents an important research direction, functional speculation about heterozygous effects alone would be premature without prior clinical genetic evidence. Although beyond the scope of the current study, investigating heterozygous functional effects is a high-priority in our next-step investigation. We appreciate the reviewer's insight in identifying this important translational research opportunity.

4. Were the immunofluorescent micrographs taken in figure 11 obtained with identical microscopy settings? If they were not, can this be specifically mentioned so that the reader does not infer quantitatively similar membrane expression from similar immunofluorescence intensity.

Yes, all immunofluorescent micrographs in Figure 11 were obtained with identical microscopy settings. We have also added clarification of this standardization to the Methods section (lines 238-240).

5. This manuscript focuses on rare nonsynonymous single nucleotide polymorphisms in Rhbg that correlated with the presence of chronic kidney disease. Although not explicitly stated, this reader would assume that this means that nonsynonymous single nucleotide polymorphisms with a MAF greater than 0.01 did not correlate. Is this correct?

Not exactly. This study did not examine the correlation of common nonsynonymous single nucleotide polymorphisms (nsSNPs) with CKD presence for several reasons. First, from a biological perspective, we prioritized rare variants (MAF $\leq 1\%$) as they are more likely to have larger effect sizes on the function of transport proteins like RhBG compared to common nsSNPs. Second, methodologically, our analysis was constrained by the available data type. The summary-level data containing population size and minor allele frequency of each SNP in the cohort from CRIC (and ARIC) WES was suitable for rare variant association testing using Chi-square tests with Yates' continuity correction and Fisher's exact tests, which are specifically designed for low-frequency variants and maintain appropriate statistical power despite small cell counts (as referenced in our cited papers: Schaid & Sinnwell, 2010; Lee et al., 2014). In contrast, common variant association analysis typically employs logistic regression models which are better suited for detecting the smaller effect sizes (characteristic of common variants). Logistic regression requires individual-level patient (and healthy control) data which was not available to us at the time this study was conducted. We have revised Method section – Association analysis subsection (lines 117-121, 128) to clarify this.

Minor comments

1. On page three of the introduction the authors seem to suggest that increased ammoniogenesis without an increase in urinary ammonia excretion would increase net

bicarbonate production. Ammonia that is produced in the kidney and is not excreted in the urine returns to the systemic circulation through the renal veins, and then is metabolized, predominantly by liver and skeletal muscle, through a process that utilizes equimolar bicarbonate. This suggests that it is unlikely that ammoniogenesis in the absence of ammonia excretion improves systemic acid-base balance. Can the authors comment on this?

We thank the reviewer for highlighting the ambiguity in our introduction. We have revised the manuscript to eliminate this ambiguity (lines 51-54).

2. In Table 1, it is not clear to this reader how the frequency of each SNV was calculated. Can this be explicitly detailed?

We thank the reviewer for seeking clarification on this important methodological detail. The minor allele frequencies (MAF) values in Table 1 were calculated using standard population genetics methodology: $MAF = (\# \text{ of alternate alleles}) / (\# \text{ of total alleles})$, calculated separately for each ancestry group. Genotypes contribute as follows:

- Homozygous reference (0/0): 0 alternate alleles
- Heterozygous (0/1): 1 alternate allele
- Homozygous alternate (1/1): 2 alternate alleles

The calculation is: $MAF = (\# \text{ of heterozygous individuals} \times 1 + \# \text{ of homozygous individuals} \times 2) / (\# \text{ of total individuals with valid genotype calls} \times 2)$

We obtained these pre-calculated frequencies from the database through our collaborators rather than performing the calculations ourselves.

We have revised Methods section to clarify this (lines 119-120).

3. I presume that successful mutagenesis was confirmed by sequencing, but I don't find this stated in the manuscript. Can the authors provide the details as to how the correct mutagenesis was confirmed?

All mutations were confirmed by Sanger sequencing as originally described in the Methods section (lines 158-159): "We purified plasmid DNA from selected colonies and confirmed the mutation by Sanger sequencing (GENEWIZ, Azenta Life Science)."

4. There's a very substantial amount of data presented in this manuscript, and it is often difficult to keep all the various findings in mind to interpret them accurately. Can the authors consider summarizing each set of data as to their effect on NH_3 , NH_4^+ , MA⁺ and MA transport? Also, would it be helpful to move figure table 4 earlier in the manuscript, and

refer to it as the data is developed.

We appreciate this valuable suggestion to improve manuscript readability. To address this concern, we have made the following changes:

- 1) we have moved Table 4 to the beginning of the Results sub-section “Effects of expressing RhBG mutations on current, pH_s and pH_i changes” (lines 310-312).
- 2) we now explicitly refer readers to Table 4 throughout Results and Discussion sections when introducing each set of transport measurements, helping readers connect individual findings within the broader functional profile of each mutant (lines 328, 388, 397, 412, 423, 452, 467, 514).

5. In figure 11, the water injected oocyte tracings are not shown. Is there a reason why not?

We believe there is a misunderstanding in this comment. Figure 11 shows immunofluorescent images, and the water-injected oocyte control is appropriately shown in this figure (upper right panel). The water-injected oocyte does not show any RhBG fluorescence. We will ensure high image quality upon resubmission.

Referee #2:

Reviewer Comments - JP-RP-2025-288958

Title: Mutations in ammonia transporter RhBG that impair NH_3/NH_4^+ transport in patients with chronic kidney disease

Summary:

This manuscript investigates rare non-synonymous variants in the RhBG ammonia transporter and their potential impact on NH_3/NH_4^+ transport capacity, using *Xenopus laevis* oocytes as an expression system. Several variants are shown to impair or abolish ammonium transport based on electrophysiological and pH-based readouts.

Although the study tackles a clinically relevant issue, how impaired ammonia transport may contribute to CKD progression, methodological flaws, overinterpretation of the results, and a lack of engagement with existing literature weaken the physiological validity of the

conclusions.

1. Exclusive Reliance on *Xenopus* Oocytes as a Model System

The exclusive use of *Xenopus* oocytes for the functional characterization of ammonia transporters constitutes a major methodological limitation. This model system has been repeatedly questioned in the literature due to its potential for confounding results. Specifically, previous studies have demonstrated that currents elicited by ammonium pulses may originate from endogenous NH_4^+ -permeable channels, rather than from the heterologously expressed transporter itself. This significantly complicates the interpretation of RhBG-mediated transport activity.

Furthermore, *Xenopus* oocytes exhibit a pronounced ability to buffer ammonium-induced changes in intracellular pH (pH_i), which is likely an evolutionary adaptation to the elevated levels of ammonium present in their natural aquatic environments. This intrinsic buffering capacity can significantly distort the cellular responses to ammonium exposure, resulting in inconsistent and potentially misleading physiological data. These limitations have been clearly documented by Musa-Aziz et al. (2009) and further discussed in the context of Rh protein function by Williamson et al. (2004). The authors should explicitly cite these studies and critically discuss their findings in relation to the present data, as they are highly relevant to the interpretation of results obtained using the *Xenopus* oocyte system. Given these well-established caveats, it is critical that the authors acknowledge these limitations and discuss their implications when interpreting the data.

Numerous studies addressing Rh-mediated transport have included complementary validation using alternative systems, such as yeast complementation assays or expression in mammalian cell lines, to overcome the inherent limitations of the *Xenopus* model. The absence of such validation in the current manuscript weakens the physiological relevance of the conclusions and should be acknowledged.

We thank the reviewer for these important methodological considerations. We address each point below:

1. Endogenous NH_4^+ -permeable channels and confounding effects

We acknowledge the reviewer's concern about potential confounding from endogenous NH_4^+ -permeable channels in *Xenopus* oocytes. Our data clearly demonstrate significant differences between these groups (Figure 1), indicating that the presence of RhBG itself contributes to the electrogenic activity observed. Additionally, this was also addressed in our previous work (Nakhoul 2010a) which extensively discussed why endogenous mechanisms activated by heterologous expression of Rhbg in *Xenopus* oocytes were unlikely responsible for the observed NH_4^+ (and MA^+) transport in electrophysiological measurements. While endogenous channels may contribute to baseline activity, the differential responses between control and RhBG-expressing oocytes, as well as between different RhBG variants, demonstrate that our system can reliably detect RhBG-specific functional differences, which is the primary objective of our variant comparison study. To address this concern, we have revised the manuscript in Discussion (lines 539-544).

2. Intrinsic pH buffering capacity of xenopus oocytes

The reviewer raises an important point regarding the study by Musa-Aziz et al. (2009) (Concentration-dependent effects on intracellular and surface pH of exposing *Xenopus* oocytes to solutions containing $\text{NH}_3/\text{NH}_4^+$). *J Membr Biol* 228, 15-31), which demonstrates that *Xenopus* oocytes sequester incoming NH_3 in acidic yolk granules where it becomes trapped as NH_4^+ , leading to paradoxical pH responses (acidification rather than alkalinization). We understand the reviewer's concern that this intrinsic buffering capacity may distort cellular responses, create inconsistent data, and lead to potentially misleading physiological interpretations. However, we were unable to locate the Williamson et al. (2004) citation in PubMed or other databases.

While we acknowledge that *Xenopus* oocytes exhibit unique $\text{NH}_3/\text{NH}_4^+$ handling characteristics, it was not demonstrated how these general oocyte properties would specifically compromise our comparative analysis of RhBG variants. This intrinsic buffering capacity may affect the absolute magnitude of responses. However, there is no evidence showing that systematic effects of oocyte physiology, including this intrinsic buffering capacity, affect experimental groups differently. Our experiments focus on relative functional differences between RhBG variants and wild-type using standardized conditions, where any systematic effects of oocyte physiology would affect all experimental groups similarly. Our multi-parameter approach (combining current, intracellular pH, surface pH, and membrane potential measurements) provides complementary evidence that supports variant-specific functional differences that cannot be explained by general oocyte artifacts alone. Without specific evidence that these oocyte characteristics differentially affect our experimental groups, our comparative interpretation remains scientifically valid and our conclusions regarding variant-specific functional impairments are well-supported by the data.

3. Complementary validation approach using alternative systems

We respectfully disagree that alternative expression systems would strengthen our study. Our research addresses specific technical questions that require the unique characteristics of the *Xenopus* oocyte system. These include distinguishing NH_3 versus NH_4^+ transport

mechanisms in variant proteins, measurement of multiple transport parameters including current, intracellular pH, surface pH, and membrane potential.

Regarding mammalian cell expression, we have previously expressed mouse Rhbg in HEK293 and IMCD3 cells for localization studies, confirming proper protein expression and trafficking. However, mammalian cells exhibit substantial passive NH_3 permeability that masks transporter-mediated effects and cannot provide the multi-parameter measurements. These limitations create significant difficulties for mechanistic discrimination between NH_3 and NH_4^+ transport.

While yeast complementation assays are useful for basic functional validation, they cannot distinguish between NH_3 and NH_4^+ transport mechanisms, or provide direct measurements of transport, and lack the precision needed to detect subtle variant-specific differences required for our comparative analysis.

The *Xenopus* oocyte system provides optimal technical advantages for our specific research questions: minimal background NH_3 permeability and capability for simultaneous multi-parameter measurements essential for mechanistic discrimination between NH_3 and NH_4^+ transport.

To meet such important criticisms as point #2 and #3 seriously, we should have to expand this study so much as to destroy the whole balance of this manuscript. We therefore kept the relevant sections unaltered.

2. Use of Methylammonium (MeA) Transport Assay

While the MeA is commonly used to assess Amt/Rh transporter activity in various systems including oocytes, its utility as a mechanistic proxy for ammonium transport is increasingly contested. A growing body of evidence has demonstrated that MeA and ammonium are not always handled similarly by these transporters, especially in variant proteins (Wang et al., 2013; Hall et al., 2011; Javelle et al., 2005).

This discrepancy is particularly evident in *Xenopus* oocytes, where studies, including Ludwig (2004), have shown that Rh proteins handle MeA and ammonium differently. Importantly, the authors' own data support this distinction: they report that an ammonium pulse induces intracellular acidification, while a MeA pulse results in alkalinization. Such inconsistency is not discussed in the manuscript.

Given the extent of the evidence challenging the validity of MeA as a reliable surrogate for

ammonium, the authors must exercise caution when interpreting results based on the use of MeA. At a minimum, this methodological issue should be explicitly acknowledged and critically discussed in the context of the broader literature.

We thank the reviewer for this important observation. It is valid to question MeA (Based on the context in Ludewig's study, we believe that the reviewer meant MA⁺ here.) as a direct surrogate for NH₄⁺. To avoid confusion, in the answer below, we used MA (and NH₃) to indicate uncharged format and MA⁺ (and NH₄⁺) to show charged molecule. We acknowledge that while our data clearly demonstrate different pH_i responses to NH₃/NH₄⁺ and MA/MA⁺, we did not explicitly discuss this discrepancy in our current manuscript. However, we have extensively characterized and discussed this issue in our previous work (Nakhoul et al., 2010a), where we demonstrated that MA/MA⁺ is transported by RhBG through distinct mechanisms compared to NH₃/NH₄⁺, with opposite effects on intracellular pH, but similar effects on current (inward current) and membrane potential (depolarization). Although we agree that MA⁺ and ammonium are handled differently by RhBG, extensive discussion of these underlying transport mechanisms is beyond the scope of this study. The established substrate differences serve as validated tools to examine variant function, and our findings provide insights that further our understanding of why NH₄⁺ and MA⁺ are handled differently by RhBG through variant-specific responses.

To address the reviewer's concern, we have revised our discussion to explicitly acknowledge this discrepancy, referencing our previous study (Nakhoul et al., 2010a) (lines 548-551).

3. Misrepresentation of Prior Literature

Throughout the Results and Discussion sections, the authors repeatedly cite Ludewig et al. (2004), asserting that their findings are in agreement with that study. However, this is a misrepresentation. As stated earlier, Ludewig explicitly cautions that observed currents in *Xenopus* oocytes may result from endogenous NH₄⁺-permeable channels rather than from RhBG itself. More importantly, Ludewig concludes that RhBG functions as an electroneutral NH₄⁺/H⁺ antiporter, which is fundamentally at odds with the mechanistic interpretation presented by the authors.

This electroneutral mechanism has since been corroborated by several independent studies, including Mak et al. (2006) and Greyer et al. (2013). Additional evidence from studies using HEK-293 and MDCK cell systems expressing RhBG further supports this conclusion, as these systems also exhibit electroneutral transport activity in response to both ammonium and MeA.

The authors should revise their interpretation to accurately reflect the current consensus in the field and avoid misrepresenting previous findings. At a minimum, a balanced discussion of alternative mechanistic models is essential for an accurate and fair representation of RhBG function.

In summary, the manuscript suffers from significant methodological and interpretative limitations that undermine the reliability of its conclusions. A more rigorous discussion of the model system's constraints and a critical engagement with the relevant literature are essential to support the authors' claims.

We thank the reviewer for this detailed critique. We address each point below:

1. Regarding Ludewig et al. (2004) citations and agreement claims

We believe there may be a misunderstanding regarding our citations of Ludewig et al. (2004). Our two citations are factually accurate and do not claim mechanistic agreement: 1) In lines 276-278, we cited Ludewig for the observation that MA⁺ does not induce current in H₂O-injected control oocytes - which is exactly what Ludewig reported. 2) In lines 511-513, we cited Ludewig for the factual observation that NH₄⁺ induced inward current while MA⁺ failed to do so in their system at the concentration of 1mM, which is what their data showed.

In both references above, our focus was on the differential observations between NH₄⁺ and MA⁺. We did not cite Ludewig to support our firm interpretation that RhBG is electrogenic. Instead, we cited specific experimental observations that are consistent among studies, regardless of mechanistic interpretation. These are factual data points, not mechanistic claims. Our own interpretation of electrogenic RhBG transport is based entirely on our experimental data in current and previous studies cited in the manuscript, not on Ludewig's conclusions.

We believe that our citations are accurate and appropriate for the specific points being made.

2. Regarding mechanistic interpretation

We appreciate the reviewer's thorough consideration of the existing literature on RhBG transport mechanisms. While we acknowledge the reviewer's concern regarding endogenous channel activation, the three foundational papers cited (Ludewig et al., 2004; Mak et al., 2006; Geyer et al., 2013) actually highlight the limitations of previous approaches rather than definitively resolving this interpretive challenge. Critically, none of these studies

provided experimental evidence to prove that observed NH_4^+ -induced currents result solely from endogenous channel activation rather than direct RhBG transport function.

We would like to clarify several key points regarding these studies and our findings. **Ludewig et al. (2004)** observed NH_4^+ -induced currents in RhBG-expressing oocytes (consistent with our observations) but concluded electroneutral transport based on the absence of MA^+ currents at resting membrane potential. Importantly, Ludewig explicitly acknowledged the interpretive challenges, noting that NH_4^+ -induced currents "may represent direct NH_4^+ transport by RhBG, or may be due to endogenous NH_4^+ -permeable channels." No experiments were performed to test this hypothesis. In contrast, our experiments using the same method did show that MA^+ induced electrogenic transport in RhBG-expressing oocytes at 5mM concentration (vs 1mM in Ludewig's study). Similarly, **Mak et al. (2006)** observed NH_4^+ -induced currents in RhBG-expressing oocytes using voltage-clamp methods but interpreted these as unrelated to RhBG transport based on voltage-independent methylamine (MA) uptake. Mak concluded that NH_4^+ currents are endogenous based on the assumption that MA transport represents the definitive transport mechanism for RhBG. In other words, this assumption can be understood as: if RhBG transports MA electroneutrally (concluded from Mak's radioactive MA uptake assay), then any electrogenic currents with NH_4^+ must be from a different mechanism (endogenous channels). However, our demonstration of electrogenic MA^+ currents with variant-specific changes suggests that RhBG handles MA/MA^+ and $\text{NH}_3/\text{NH}_4^+$ differently, in agreement with the point which the reviewer raised in the comment earlier ("2. Use of Methylammonium (MeA) Transport Assay"). Mak's study did not directly measure whether MA^+ could also produce electrogenic currents. This methodological gap invalidates the basis for attributing NH_4^+ currents as purely endogenous effects. Despite testing transporter inhibitors' effects on MA uptake, no systematic analysis was performed in Mak's study to distinguish between direct transport and endogenous channel modulation mechanisms. **Geyer et al. (2013)** primarily focused on the electroneutral gas transport of NH_3 and CO_2 by measuring surface pH. They only monitored membrane potential as a quality control criterion (-40 mV threshold) and explicitly stated "our recordings of V_m reveal no evidence of an electrogenic flux of NH_4^+ " without presenting any voltage data, while our study reported both current changes and membrane potential changes at 5mM concentration vs 0.5mM in Geyer's study.

3. Regarding the current consensus

The prevailing view that RhBG mediates only electroneutral transport stems from Ludewig's 2004 study, which was subsequently reinforced by Mak et al. (2006) and Geyer et al. (2013). However, this paradigm was established based on methodological limitations: Ludewig calculated that MA^+ transport should produce ~32 nA currents but didn't detect them, concluding transport was electroneutral. Our direct detection of significant electrogenic currents induced by MA^+ (-35.7 ± 13.3 nA, $p < 0.001$) demonstrates that previous studies

lacked the condition to detect electrogenic transport, challenging the assumption that has influenced the field for nearly two decades.

4. Regarding the reviewer's comment about "additional evidence from studies using HEK-293 and MDCK cell systems"

We could locate **only one** such study (Zidi-Yahiaoui et al., 2005), which used both HEK-293 and MDCK cell lines but represents a single investigation. This study used stopped-flow fluorimetry to measure NH_3 permeability but did not directly address electrogenic versus electroneutral transport mechanisms. The study concluded electroneutral NH_3 transport based on pH measurements, though we note that pH changes alone cannot definitively distinguish between NH_3 and NH_4^+ transport mechanisms. We have previously expressed RhBG in HEK-293 and IMCD3 cells for localization studies, but the main challenge in mammalian systems is distinguishing NH_3 from NH_4^+ transport due to high background passive NH_3 permeability and difficulty obtaining reliable electrophysiological measurements.

5. Regarding the reviewer's summary of comments

We would like to emphasize that our study's primary objective is to comprehensively characterize variant-specific functional effects on RhBG transport properties with different substrates rather than to resolve debates about RhBG's electrogenic vs electroneutral transport mechanisms. Our data reveal that RhBG exhibits fundamentally different behaviours with MA/MA⁺ versus $\text{NH}_3/\text{NH}_4^+$, and that specific mutations dramatically alter these transport properties in substrate-dependent ways. This complexity was not systematically explored in previous studies.

To accommodate the reviewer's overall concern, we have revised the Discussion section (lines 533-539, 542-544, 552-554) and integrated the changes with revisions for other reviewers' comments.

END OF COMMENTS

Dear Dr Nakhoul,

Re: JP-RP-2025-288958R1 "**Mutations in ammonia transporter RhBG that impair $\text{NH}_3/\text{NH}_4^+$ transport in patients with chronic kidney disease**" by Nazih L. Nakhoul, He Zhou, Abdunour-Nakhoul Solange, and L. Lee Hamm

Thank you for submitting your manuscript to The Journal of Physiology. It has been assessed by a Reviewing Editor and by 3 expert referees and we are pleased to tell you that it is acceptable for publication following satisfactory revision.

REVISION CHECKLIST:

We look forward to receiving your revised submission.

Yours sincerely,

Peying Fong
Senior Editor
The Journal of Physiology

EDITOR COMMENTS

Reviewing Editor:

The revisions have mostly addressed the Reviewer comments. Can the authors please revise the Discussion to include reference to the technical challenges of studying Amt/Mep/Rh proteins, as outlined by Reviewer 2. This is a useful addition to give context to this study.

Senior Editor:

I commend to your attention feedback offered by the Statistics Editor. This input arises from your response to a concern raised by Referee 1 in review of the initial submission. The recommendation is clearly articulated and provides guidelines for reconciling the issue at hand.

Review of your manuscript, "Mutations in ammonia transporter RhBG that impair NH₃/NH₄⁺ transport in patients with chronic kidney disease" is now complete. In the attached reviews, you will read that the Referees are largely satisfied with most conceptual concerns raised in revision of the previous version.

Note, however, that Referee 1 signals (in points 2a and 2b of their report) lingering concerns related to your rebuttal of an original point (2) regarding statistical analysis that was raised during review of the initial submission. Consultation with a Statistics Editor therefore was sought. The outcome of this analysis is clearly and emphatically articulated in the Statistics Editor's accompanying report. There are two possible approaches suggested that will correct the matter. Note that the second option also was offered by Referee 1. These advised actions must be implemented in your revised manuscript before it can proceed.

In addition to this critical action requested, an additional point pertaining to the technical challenges of studying Amt/Mep/Rh proteins was raised by Referee 2; we anticipate you will be able to address this point readily.

REFEREE COMMENTS

Referee #1:

This is a revision of the manuscript, "Mutations in ammonia transporter Rhbg that impair NH₃/NH₄⁺ transport in patients with chronic kidney disease" by Zhou, Abdunour-Nakhoul, Hamm, and Nakhoul. The authors have been responsive to the previous review.

Major comments

1. Adequately addressed

2a. The authors suggest that their statistical analysis is correct because they and others have used this approach previously. This reviewer is concerned that it is not correct. In particular, a fundamental assumption of the unpaired t-test is that the data

in each group is normally distributed and that the correlation between any two data points is identical. These two assumptions do not hold when studying measurements such as those in this study, particularly when there is a known batch effect. If this assumption is violated, isn't the appropriate analytic approach to calculate the mean in each batch and use this as an "n" of 1 for statistical analysis? In other words, if there are 4, 5, and 6 replicates in three separate batches, the preferred 'n' is 3, not 15. Given the description of the similarity of data in each batch, this reviewer expects that a similar statistical finding will be obtained. Because the reviewer and the authors have different opinions as to the necessity of this reanalysis of the already available data, this reviewer will leave the decision regarding the necessity of this to the editors.

2b. The statement in line 191 that similar group sizes remove batch effects is not correct and should be removed.

3. Adequately addressed.

4. Adequately addressed.

5. Adequately addressed.

Minor comments

1. Adequately addressed.

2. Adequately addressed.

3. Adequately addressed.

4. Adequately addressed.

5. The reviewer apologizes for the typographical error in my review. This should have been a reference to Figure 2. If this can be easily corrected, then great. However, the reviewer does not consider a major concern at this time.

Referee #2:

The authors' response and the added text are satisfactory. However, I believe it is important to include a sentence in the added paragraph (lines 536-544) to highlight that the technical challenges of studying Amt/Mep/Rh proteins, particularly in the context of characterizing whether their transport activity is electrogenic or electroneutral, were recently reviewed by Williamson et al. (2024), DOI: 10.1042/BSR20211209. This review addresses the technical difficulties of studying this transporter superfamily, a topic that has not been covered in depth before. Citing this review would provide valuable context and likely be of interest to the audience of the paper.

Referee #3:

I share Reviewer 1's concerns regarding the statistical methodology employed in this study. The authors' approach of treating individual oocytes as independent observations represents a clear case of pseudoreplication that violates the independence assumptions underlying their chosen statistical tests. The hierarchical structure of the data-oocytes nested within batches within frogs-requires analytical methods that account for this clustering, yet the current analysis inappropriately pools all observations and inflates the effective sample size from the true value of N=3 independent batches to N=15 individual oocytes per group.

The statistical section confirms these concerns by explicitly stating that "N represents the total number of individual oocytes pooled from 3 independent injection batches," which directly contradicts proper statistical practice for clustered data. This approach will systematically underestimate standard errors and produce artificially significant results, as the correlation between oocytes within the same batch is ignored. The power calculations based on individual oocytes are therefore meaningless, and the study is likely severely underpowered when considering the appropriate unit of analysis. I recommend that the authors reanalyse their data using either mixed-effects models with batch as a random effect or, as Reviewer 1 suggests, calculate batch means and treat these as the unit of analysis (N=3). Without this correction, the statistical inferences drawn from this study cannot be considered reliable. The paper referenced below provides a nice outline of this issue, and the solutions:

Aarts, E., Verhage, M., Veenvliet, J. V., Dolan, C. V., & Van Der Sluis, S. (2014). A solution to dependency: using multilevel analysis to accommodate nested data. *Nature neuroscience*, 17(4), 491-496.

END OF COMMENTS

EDITOR COMMENTS

Reviewing Editor:

The revisions have mostly addressed the Reviewer comments. Can the authors please revise the Discussion to include reference to the technical challenges of studying Amt/Mep/Rh proteins, as outlined by Reviewer 2. This is a useful addition to give context to this study.

We thank the Reviewing Editor's suggestion. We have revised the discussion and included the reference as suggested by Reviewer 2 (lines 546-549).

Senior Editor:

I commend to your attention feedback offered by the Statistics Editor. This input arises from your response to a concern raised by Referee 1 in review of the initial submission. The recommendation is clearly articulated and provides guidelines for reconciling the issue at hand.

Review of your manuscript, "Mutations in ammonia transporter RhBG that impair NH₃/NH₄⁺ transport in patients with chronic kidney disease" is now complete. In the attached reviews, you will read that the Referees are largely satisfied with most conceptual concerns raised in revision of the previous version.

Note, however, that Referee 1 signals (in points 2a and 2b of their report) lingering concerns related to your rebuttal of an original point (2) regarding statistical analysis that was raised during review of the initial submission. Consultation with a Statistics Editor therefore was sought. The outcome of this analysis is clearly and emphatically articulated in the Statistics Editor's accompanying report. There are two possible approaches suggested that will correct the matter. Note that the second option also was offered by Referee 1. These advised actions must be implemented in your revised manuscript before it can proceed.

In addition to this critical action requested, an additional point pertaining to the technical challenges of studying Amt/Mep/Rh proteins was raised by Referee 2; we anticipate you will be able to address this point readily.

We appreciate the Senior Editor' summary and suggestions. We have addressed all points raised by the Reviewing Editor and the 3 referees.

REFEREE COMMENTS

Referee #1:

This is a revision of the manuscript, "Mutations in ammonia transporter Rhbg that impair $\text{NH}_3/\text{NH}_4^+$ transport in patients with chronic kidney disease" by Zhou, Abdulnour-Nakhoul, Hamm, and Nakhoul. The authors have been responsive to the previous review.

Major comments

1. Adequately addressed

2a. The authors suggest that their statistical analysis is correct because they and others have used this approach previously. This reviewer is concerned that it is not correct. In particular, a fundamental assumption of the unpaired t-test is that the data in each group is normally distributed and that the correlation between any two data points is identical. These two assumptions do not hold when studying measurements such as those in this study, particularly when there is a known batch effect. If this assumption is violated, isn't the appropriate analytic approach to calculate the mean in each batch and use this as an "n" of 1 for statistical analysis? In other words, if there are 4, 5, and 6 replicates in three separate batches, the preferred 'n' is 3, not 15. Given the description of the similarity of data in each batch, this reviewer expects that a similar statistical finding will be obtained. Because the reviewer and the authors have different opinions as to the necessity of this reanalysis of the already available data, this reviewer will leave the decision regarding the necessity of this to the editors.

We sincerely appreciate the reviewer's continued engagement and thank the reviewer for the comments that led toward this methodological improvement.

We have completely re-analyzed our entire dataset using linear mixed-effects models (MEM) with batch as a random effect, as the Statistics Editor (Reviewer #3) recommended. This approach directly addresses the reviewer's concerns about independence assumption and correlation between data points within batches.

The mixed-effects analysis has yielded remarkable results:

- It confirmed robustness of our primary findings using ANOVA: All previously significant differences remained statistically significant, validating the biological relevance of our original conclusions.

- The MEM approach revealed additional statistically significant differences that were masked in our previous analysis due to batch-related noise and further validated our interpretations of our findings.

We also conducted intraclass correlation coefficient (ICC) analysis. The ICC results revealed batch effects ranging from zero to moderate across all experimental oocyte groups, confirming that while batch effects are present, they are generally small to moderate and appropriately handled by the mixed-effects framework. This also supports the robustness of our primary analysis using ANOVA.

We have included the new analysis results in Appendix Tables A1 and A2, added an Appendix Methods section, and revised the Methods and Discussion section (lines 255-257, 518-521).

2b. The statement in line 191 that similar group sizes remove batch effects is not correct and should be removed.

We thank the reviewer for pointing this out. We have revised the sentence (lines 191-192).

3. Adequately addressed.

4. Adequately addressed.

5. Adequately addressed.

Minor comments

1. Adequately addressed.

2. Adequately addressed.

3. Adequately addressed.

4. Adequately addressed.

5. The reviewer apologizes for the typographical error in my review. This should have been a reference to Figure 2. If this can be easily corrected, then great. However, the reviewer does not consider a major concern at this time.

We thank the reviewer for clarifying this. The comparison between wild-type RhBG-expressing oocytes and water-injected oocytes was depicted in Figure 1. We include only the statistical result of water-injected oocytes in Figure 2 to maintain a focused result presentation and to avoid redundancy.

Referee #2:

The authors' response and the added text are satisfactory. However, I believe it is important to include a sentence in the added paragraph (lines 536-544) to highlight that the technical challenges of studying Amt/Mep/Rh proteins, particularly in the context of characterizing whether their transport activity is electrogenic or electroneutral, were recently reviewed by Williamson et al. (2024), DOI: 10.1042/BSR20211209. This review addresses the technical difficulties of studying this transporter superfamily, a topic that has not been covered in depth before. Citing this review would provide valuable context and likely be of interest to the audience of the paper.

We thank the reviewer for providing a valuable citation to help increase our study impact. We have added a sentence in the relevant paragraph with updated reference (lines 546-549).

Referee #3:

I share Reviewer 1's concerns regarding the statistical methodology employed in this study. The authors' approach of treating individual oocytes as independent observations represents a clear case of pseudoreplication that violates the independence assumptions underlying their chosen statistical tests. The hierarchical structure of the data-oocytes nested within batches within frogs-requires analytical methods that account for this clustering, yet the current analysis inappropriately pools all observations and inflates the effective sample size from the true value of $N=3$ independent batches to $N=15$ individual oocytes per group.

The statistical section confirms these concerns by explicitly stating that "N represents the total number of individual oocytes pooled from 3 independent injection batches," which directly contradicts proper statistical practice for clustered data. This approach will systematically underestimate standard errors and produce artificially significant results, as the correlation between oocytes within the same batch is ignored. The power calculations based on individual oocytes are therefore meaningless, and the study is likely severely

underpowered when considering the appropriate unit of analysis. I recommend that the authors reanalyse their data using either mixed-effects models with batch as a random effect or, as Reviewer 1 suggests, calculate batch means and treat these as the unit of analysis (N=3). Without this correction, the statistical inferences drawn from this study cannot be considered reliable. The paper referenced below provides a nice outline of this issue, and the solutions:

Aarts, E., Verhage, M., Veenvliet, J. V., Dolan, C. V., & Van Der Sluis, S. (2014). A solution to dependency: using multilevel analysis to accommodate nested data. *Nature neuroscience*, 17(4), 491-496.

We have re-analyzed our entire dataset using linear mixed-effects models with batch as a random effect with FDR correction for multiple comparisons. The mixed-effects analysis has addressed the reviewer's statistical concerns. All major conclusions from our original analysis remained statistically significant, demonstrating the biological robustness of our findings. The improved statistical approach revealed several additional significant differences that were previously masked.

We also calculated intraclass correlation coefficient based on null model and full model of linear mixed-effects models. The adjusted ICC results revealed batch effects ranging from zero to 0.435 across parameters, confirming that while batch effects are present, they are generally small to moderate and appropriately handled by the mixed-effects framework.

We have included the new analysis results in Appendix Tables S1 and S2, added an Appendix Methods section, and revised the Methods and Discussion section (lines 255-257, 518-521).

We are grateful for the reviewer's statistical expertise. The reviewer's recommendation for mixed-effects modeling has fundamentally improved the rigor of our current study.

END OF COMMENTS

Dear Dr Nakhoul,

Re: JP-RP-2025-288958R2 "**Mutations in ammonia transporter RhBG that impair $\text{NH}_3/\text{NH}_4^+$ transport in patients with chronic kidney disease**" by Nazih L. Nakhoul, He Zhou, Abdulnour-Nakhoul Solange, and L. Lee Hamm

Thank you for submitting your manuscript to The Journal of Physiology. It has been assessed by a Reviewing Editor and by 3 expert referees and we are pleased to tell you that it is acceptable for publication following satisfactory revision.

ABSTRACT FIGURES: Authors are expected to use The Journal's premium BioRender account to create/redraw their Abstract Figures. Information on how to access this account is here:

<https://physoc.onlinelibrary.wiley.com/journal/14697793/biorender-access>.

REVISION CHECKLIST:

We look forward to receiving your revised submission.

Yours sincerely,

Peying Fong
Senior Editor
The Journal of Physiology

EDITOR COMMENTS

Reviewing Editor:

Both reviewers acknowledged that the re-analysis addresses concerns and note that the biological findings remain robust. The statistical reviewer highlights that the analysis confirms batch effects and recommends that the revised mixed-model analysis is included in the main text as the primary analysis. The other reviewer also makes this point.

Please revise to ensure that readers encounter the revised analysis in the main text. The original analysis can be retained in the appendix, with additional text to address the caveats.

Senior Editor:

Referee reports pertaining to your revised manuscript, "Mutations in ammonia transporter RhBG that impair NH₃/NH₄⁺ transport in patients with chronic kidney disease", are now complete and are attached herewith for your consideration. Referees 1 and 2 concur that there is potential for your study to be quite influential.

Note, however, that although both Referee 1 and the Statistics Editor (Referee 3) appreciate that the current version does apply the statistical approach requested in their previous critiques, they both also feel that the mixed effects model be presented as the primary analysis in the narrative within the manuscript, rather than in the appendix. The Reviewing Editor concurs with their assessment.

Once this modification is made, your study would set the precedent for using (and reporting) these more appropriate statistical tests, rather than communicating tacit approval of a questionable, previous practice. Presenting to readers the statistically valid framework for treating nested data within the narrative will encourage this practice.

We therefore collectively recommend that you report the original approach instead in an appendix, as recommended by the Statistics Editor (Referee 3).

Finally, while both approaches yield the same conclusion with your present data, with other data sets they may in fact may not.

Thank you for contributing this study to The Journal of Physiology. I look forward to receiving your revised manuscript soon.

REFEREE COMMENTS

Referee #1:

The authors have responded to my concerns regarding the correct statistical analysis with an additional analysis. This has been added to the Appendix. The added analysis completely addresses my concerns.

The incorrect analysis, however, has not been removed from the paper. This reviewer wonders if this should have been handled this way. Shouldn't the incorrect analysis, even if it gave the same conclusions, have been removed?

Referee #2:

no further comments.

Referee #3:

The authors have effectively addressed concerns about pseudoreplication by reanalysing their dataset with linear mixed-effects models, incorporating batch as a random effect to account for the hierarchical structure of oocytes nested within batches. Their revised analysis, with FDR correction for multiple comparisons and detailed documentation in new appendix tables and methods sections, confirms batch effects (intraclass correlation coefficients from 0 to 0.435), validating the original concern. The biological findings remain robust, with all major conclusions statistically significant and some previously masked differences now apparent.

However, I strongly recommend presenting the mixed-effects model results as the primary analysis in the main Methods and Results sections, rather than in appendices. This approach reflects the statistically valid framework for nested data, ensuring readers encounter the correct methodology in the main text. If the authors wish to retain the original analysis for comparison, it could be moved to supplementary material with appropriate caveats. I defer to the Senior Editor's judgement on whether to require this restructuring or accept the current presentation.

END OF COMMENTS

EDITOR COMMENTS

Reviewing Editor:

Both reviewers acknowledged that the re-analysis addresses concerns and note that the biological findings remain robust. The statistical reviewer highlights that the analysis confirms batch effects and recommends that the revised mixed-model analysis is included in the main text as the primary analysis. The other reviewer also makes this point.

Please revise to ensure that readers encounter the revised analysis in the main text. The original analysis can be retained in the appendix, with additional text to address the caveats.

We thank the Reviewing Editor's suggestion. We modified the manuscript to address all the points raised by the reviewers. Below is a summary of the revision conducted:

- As requested, we have revised the manuscript to set the mixed-effects models as the primary analysis. Because linear mixed-effects modeling is a less common method in biology field, we believe that reporting the observed measured mean \pm SD instead of estimates along with adjusted p-values from mixed-effects models will best maintain the data presentation flow of the manuscript and a familiar format for biological interpretation. Therefore, we kept the complete mixed-effects models output tables in the appendix (Tables A1a and A1b) and added in the main text a new table of the intra-class coefficient (ICC) results (Table 4) to demonstrate the magnitude of batch effects and justify the mixed-effects modeling approach. We also updated the summary table (now Table 5) with adjusted p-values from mixed-effects models.
- Textual changes are yellow-highlighted and can be found in Methods section (Statistics subsection), Results section (p-values were replaced with the new method's p-values), and Discussion section (modified for significant new results revealed by the new method).
- We have moved the old analysis results to the appendix (Appendix Methods and Table A2).
- We updated all bar plots with the significance indicators based on p-values from mixed-effects models.

Senior Editor:

Referee reports pertaining to your revised manuscript, "Mutations in ammonia transporter RhBG that impair NH₃/NH₄⁺ transport in patients with chronic kidney disease", are now

complete and are attached herewith for your consideration. Referees 1 and 2 concur that there is potential for your study to be quite influential.

Note, however, that although both Referee 1 and the Statistics Editor (Referee 3) appreciate that the current version does apply the statistical approach requested in their previous critiques, they both also feel that the mixed effects model be presented as the primary analysis in the narrative within the manuscript, rather than in the appendix. The Reviewing Editor concurs with their assessment.

Once this modification is made, your study would set the precedent for using (and reporting) these more appropriate statistical tests, rather than communicating tacit approval of a questionable, previous practice. Presenting to readers the statistically valid framework for treating nested data within the narrative will encourage this practice.

We therefore collectively recommend that you report the original approach instead in an appendix, as recommended by the Statistics Editor (Referee 3).

Finally, while both approaches yield the same conclusion with your present data, with other data sets they may in fact may not.

Thank you for contributing this study to The Journal of Physiology. I look forward to receiving your revised manuscript soon.

We appreciate the Senior Editor' summary and endorsement of our statistical approach. We have revised the manuscript to set the mixed-effects models as the primary analysis. Below is a summary of the revision conducted:

- Because linear mixed-effects modeling is a less common method in biology field, we believe that reporting the observed mean \pm SD instead of estimates along with adjusted p-values from mixed-effects models will best maintain the data presentation flow of the manuscript and a familiar format for biological interpretation. Therefore, we kept the complete mixed-effects models output tables in the appendix (Tables A1a and A1b) and added a new table of the intra-class coefficient (ICC) results (Table 4) to demonstrate the magnitude of batch effects and justify the mixed-effects modeling approach in the main text. We also updated the summary table (now Table 5) with adjusted p-values from mixed-effects models.
- Textual changes are yellow-highlighted and can be found in Methods section (Statistics subsection), Results section (p-values were replaced with the new

method's p-values), and Discussion section (modified for significant new results revealed by the new method).

- We have moved the old analysis results to the appendix (Appendix Methods and Table A2).
- We updated all bar plots with the significance indicators based on p-values from mixed-effects models.

REFEREE COMMENTS

Referee #1:

The authors have responded to my concerns regarding the correct statistical analysis with an additional analysis. This has been added to the Appendix. The added analysis completely addresses my concerns.

The incorrect analysis, however, has not been removed from the paper. This reviewer wonders if this should have been handled this way. Shouldn't the incorrect analysis, even if it gave the same conclusions, have been removed?

We thank the reviewer's suggestion. We have revised the manuscript to set the mixed-effects models as the primary analysis. Below is a summary of the revision conducted:

- Textual changes are yellow-highlighted and can be found in Methods section (Statistics subsection), Results section (p-values were replaced with the new method's p-values), and Discussion section (modified for significant new results revealed by the new method).
- To maintain the data presentation flow of the manuscript, we kept the complete mixed-effects models output tables in the appendix (Tables A1a and A1b) and added a new table of the intra-class coefficient (ICC) results (Table 4) in the main text and updated the p-values in the original Table 4 (now Table 5).
- We have moved the old analysis results to the appendix (Appendix Methods and Table A2).
- We updated all bar plots with the significance indicators based on p-values from mixed-effects models.

Referee #2:

no further comments.

Referee #3:

The authors have effectively addressed concerns about pseudoreplication by reanalysing their dataset with linear mixed-effects models, incorporating batch as a random effect to account for the hierarchical structure of oocytes nested within batches. Their revised analysis, with FDR correction for multiple comparisons and detailed documentation in new appendix tables and methods sections, confirms batch effects (intraclass correlation coefficients from 0 to 0.435), validating the original concern. The biological findings remain robust, with all major conclusions statistically significant and some previously masked differences now apparent.

However, I strongly recommend presenting the mixed-effects model results as the primary analysis in the main Methods and Results sections, rather than in appendices. This approach reflects the statistically valid framework for nested data, ensuring readers encounter the correct methodology in the main text. If the authors wish to retain the original analysis for comparison, it could be moved to supplementary material with appropriate caveats. I defer to the Senior Editor's judgement on whether to require this restructuring or accept the current presentation.

We thank the reviewer's recommendation. We have revised the manuscript to set the mixed-effects models as the primary analysis. Below is a summary of the revision conducted:

- Because linear mixed-effects modeling is a less common method in biology field, we believe that reporting the observed mean \pm SD instead of estimates along with adjusted p-values from mixed-effects models will best maintain the data presentation flow of the manuscript and a familiar format for biological interpretation. Therefore, we kept the complete mixed-effects models output tables in the appendix (Tables

A1a and A1b) and added a new table of the intra-class coefficient (ICC) results (Table 4) to demonstrate the magnitude of batch effects and justify the mixed-effects modeling approach in the main text. We also updated the summary table (now Table 5) with adjusted p-values from mixed-effects models.

- Textual changes are yellow-highlighted and can be found in Methods section (Statistics subsection), Results section (p-values were replaced with the new method's p-values), and Discussion section (modified for significant new results revealed by the new method).
- We have moved the old analysis results to the appendix (Appendix Methods and Table A2).
- We updated all bar plots with the significance indicators based on p-values from mixed-effects models.

END OF COMMENTS

Dear Dr Nakhoul,

Re: JP-RP-2025-288958R3 "**Mutations in ammonia transporter RhBG that impair NH₃/NH₄⁺ transport in patients with chronic kidney disease**" by Nazih L. Nakhoul, He Zhou, Abdulnour-Nakhoul Solange, and L. Lee Hamm

We are pleased to tell you that your paper has been accepted for publication in The Journal of Physiology.

Yours sincerely,

Peying Fong
Senior Editor
The Journal of Physiology

IMPORTANT POINTS TO NOTE FOLLOWING ACCEPTANCE OF YOUR PAPER:

- **IMPORTANT NOTICE ABOUT OPEN ACCESS:** To assist authors whose funding agencies mandate immediate public access to published research findings, The Journal of Physiology allows authors to pay an Open Access (OA) fee to have their papers made freely available immediately on publication.

- You can help your research get the attention it deserves! Check out Wiley's free Promotion Guide for best-practice recommendations for promoting your work at: www.wileyauthors.com/eeo/guide. You can learn more about Wiley Editing Services which offers professional video, design, and writing services to create shareable video abstracts, infographics, conference posters, lay summaries, and research news stories for your research at: www.wileyauthors.com/eeo/promotion.

- If you would like to receive our 'Research Roundup', a monthly newsletter highlighting the cutting-edge research published in The Physiological Society's family of journals (The Journal of Physiology, Experimental Physiology, Physiological Reports, The Journal of Nutritional Physiology and The Journal of Precision Medicine: Health and Disease), please click this link, fill in your name and email address and select 'Research Roundup':
<https://www.physoc.org/journals-and-media/membernews>

EDITOR COMMENTS

Senior Editor:

Review of your manuscript, "Mutations in ammonia transporter RhBG that impair NH₃/NH₄⁺ transport in patients with chronic kidney disease" is now complete. You will find the critiques of this last round of review attached. I am pleased to inform you of its acceptability for publication in The Journal of Physiology. Please accept my thanks for your continued contributions. They are highly valued.

REFEREE COMMENTS

Referee #1:

The authors have addressed the concerns regarding how to report their statistical analysis in their revised manuscript. I have a few minor comments regarding their reporting of this. First, I do not see it explicitly stated which analysis gave rise to the p-values reported in the manuscript. Second, the reporting of the group means, rather than the estimated marginal means, seems to differ from the explicit request to use only the most appropriate statistical approaches to report and analyze the data.

Referee #2:

no further comment

Referee #3:

I am happy with the outlined approach and this version of the manuscript.